# Signal-noise metrics for RNA binding protein identification reveal broad spectrum protein-RNA interaction frequencies and dynamics

JohnCarlo Kristofich[1] & Christopher V. Nicchitta ®[1] ✉

Recent efforts towards the comprehensive identification of RNA-bound proteomes have revealed a large, surprisingly diverse family of candidate RNA-binding proteins (RBPs). Quantitative metrics for characterization and validation of protein-RNA interactions and their dynamic interactions have, however, proven analytically challenging and prone to error. Here we report a method termed LEAP-RBP (Liquid-Emulsion-Assisted-Purification of RNA-Bound Protein) for the selective, quantitative recovery of UV-crosslinked RNA-protein complexes. By virtue of its high specificity and yield, LEAP-RBP distinguishes RNA-bound and RNA-free protein levels and reveals common sources of experimental noise in RNA-centric RBP enrichment methods. We introduce strategies for accurate RBP identification and signal-based metrics for quantifying protein-RNA complex enrichment, relative RNA occupancy, and method specificity. In this work, the utility of our approach is validated by comprehensive identification of RBPs whose association with mRNA is modulated in response to global mRNA translation state changes and through in-depth benchmark comparisons with current methodologies.

The essential contributions of RNA-binding proteins (RBPs) to RNA biology have fostered the development of biochemical methods for the RNA-centric capture and identication of the RNA-interacting proteome[1–6]. Through these efforts the universe of candidate RBPs has expanded dramatically, with RNA-binding functionality now attributed to a substantial fraction of the proteome, including glycolytic enzymes, regulatory kinases, and other proteins not previously implicated in RNA biology[7–9]. With the growing catalog of candidate RBPs has come the challenge of establishing quantitative criteria for RNA-binding activity, metrics for distinguishing specific (signal) from random (noise) protein–RNA interactions, and experimental approaches to the study of protein–RNA interaction dynamics[10–12]. Although substantial progress has been made on these fronts, significant methodological limitations in the study of RNA-protein interaction remain, and a systems-level understanding of RBP biology remains a frontier area of study[10,13–17].

As with protein–protein interactions, protein–RNA interactions can vary substantially in their specificities, interaction lifetimes, and apparent affinities[13]. This intrinsic biological property creates methodological hurdles to establishing biologically relevant interactions, particularly when interaction energies are weak and thus readily lost during biochemical isolation. In the case of RNA-protein complexes (RNPs), chemical- or UV cross-linking methods can capture physiologically relevant interactions though as with any cross-linking method, criteria for distinguishing specific from biologically irrelevant interactions are critical. UV cross-linking is preferred due to its high specificity, though it is also inefficient, with only a fraction of interactors forming a covalent adduct[18–20]. The generally low cross-linking efficiencies present an analytical challenge because selective, quantitative recovery of the UV-crosslinked protein–RNA complexes (clRNPs) is necessary for accurately determining RNA occupancy

[1]Department of Cell Biology, Duke University School of Medicine, Durham, NC 27710, USA. ✉e-mail: christopher.nicchitta@duke.edu

states in vivo. Here we report two different sources of non-specific free protein recovery during RNA-centric enrichment of clRNPs. To distinguish RNA-bound and free protein levels, we introduce signal-to-noise ($S/N$) metrics, where $S/N$ is a protein-specific metric representing the ratio of RNA-bound ($S$) to unbound counterparts ($N$), and where proteins without RNA-bound counterparts ($S = 0$) comprise background ($B$). Because mass spectrometry-based proteomic analysis does not distinguish RNA-bound proteins from their unbound counterparts, $S/N$-based metrics provide a valuable quantitative criterion for high confidence assignment of RNA-binding function.

In this work, we identify and validate experimental approaches for evaluating $S/N$ and use them to guide the development of a biochemical method termed LEAP-RBP (Liquid-Emulsion-Assisted-Purification of RNA-Bound Protein) for the selective isolation of total RNA-bound protein. SILAC LC–MS/MS analysis of LEAP-RBP fractions demonstrated high RNA-bound protein enrichment and through comparative analyses, revealed a key metric for evaluating method specificity for RNA-bound RBPs which we term %TP$_S$, or RNA-bound protein abundance. High %TP$_S$ is indicative of low free protein recovery and enables the accurate study of dynamic, cell state-determined changes in RBP occupancy state. Using this signal-based analytical

framework, we present perspectives and strategies for studying RNA-bound proteomes and their dynamics. The utility of this approach is established through benchmark comparisons of LEAP-RBP with current RNA-centric enrichment methods.

## Results

### RNA-bound proteins display RNase-dependent SDS-PAGE mobility

The work reported here builds on the discovery of organic phase separation methods for the selective enrichment of clRNPs[3,4,21]. Whereas acidic guanidinium thiocyanate–phenol–chloroform (AGPC) biphasic extraction (i.e., Trizol) has been widely used for the isolation of RNA and protein from biological samples[22,23], the finding that RBPs exhibit UV-dependent enrichment at the AGPC interphase revealed a utility for this method in RNA-protein interactome discovery (Supplementary Note 1)[3,4,24]. Previously, it was reported that repeated AGPC interphase extraction further enriches RBPs over non-RBPs[4,24]. Here, we assessed the ability of repeated AGPC extraction to enhance $S/N$ (RNA-bound protein/free protein) of AGPC interphase proteins using an SDS-PAGE RNase-sensitivity Assay (SRA) (Fig. 1a).

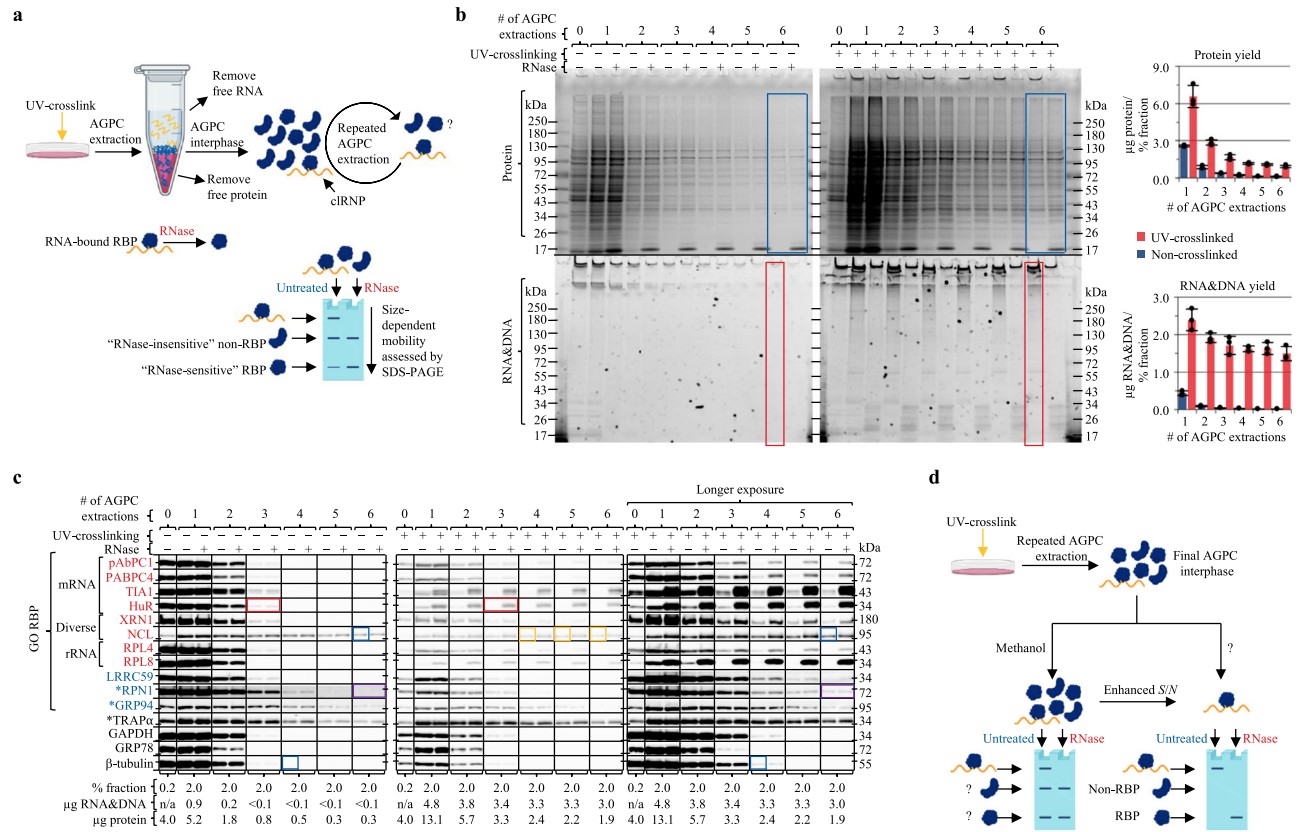

**Fig. 1 | RNase-dependent SDS-PAGE mobility provides a key diagnostic for identifying RNA-binding proteins.** AGPC acidic guanidinium thiocyanate–phenol–chloroform, SRA SRA analysis. **a** Schematic illustration of the experimental approach. Repeated AGPC extraction enriches RBPs over non-RBPs; effect of repeated AGPC extraction on $S/N$ was tested. Graphical representation of SRA: RNase-treated samples are compared to equivalent amounts of untreated samples by SDS-PAGE. RNA-bound proteins in untreated samples migrate at a higher molecular weight than their unbound counterparts. RNase digestion liberates RNA-bound RBPs allowing their mobilization into the separating gel. **b** Comparison of AGPC interphase samples isolated by methanol precipitation (95% v/v) following 1–6 AGPC extraction(s) of UV-crosslinked or non-crosslinked HeLa cells by SRA and Coomassie Blue (protein), SYBR Safe (RNA&DNA) staining; parallel gels. Protein (BCA) and RNA&DNA (UV-spectrophotometry) yields shown in the

adjoining bar charts represent the mean ± 1 SD of three biologically independent samples, pooled (equivalent % fraction) for SRA analysis. **c** Immunoblot analysis of samples from (**b**). Sample compositions, target protein RBP–RNA interactome category, and target protein GO:RBP-annotation status indicated in the figure panel. Full bots are provided in Supplementary Fig. 1a, RNA-binding domains and list of studies reporting UV-enrichment* provided in Supplementary Data 1. **d** Schematic illustration of enhanced $S/N$ output. SRA is unable to distinguish non-RBPs from RBPs with low $S/N$ in RNP fractions isolated by methanol precipitation (95% v/v) from the final AGPC interphase. SRA supports the identification of RNA enrichment methods that further enhance $S/N$, evidenced by a marked decrease in RNase-insensitive protein. Graphics (**a**, **b**) were created with BioRender.com. Experiments (**b**, **c**) were performed four times with similar results. Source data are provided as a Source Data file.

As illustrated in Fig. 1a, UV cross-linked RNA-protein complexes, which comprise signal ($= S$), migrate at a higher apparent molecular weight than their unbound counterparts (noise $= N$) in SDS-PAGE[21,25]. RNase treatment liberates RNA-bound protein, increasing the amount of observed ($= O$) protein migrating at its expected molecular weight ($\Delta O = |S|$). Consequently, a comparison of RNase-treated ($S + N$) and untreated control samples ($N$) by SDS-PAGE reveals RNase-sensitive RBPs ($|S| > 0$) and enables the determination of $S/N$. As described in Eq. (1), $\log_2$ transformation yields RNase-dependent fold-change (Supplementary Note 2).

$$\text{RNase-dependent fold-change} = \Delta\log_2(O) = \log_2(|S| + N)_{RNase} - \log_2(N)_{untreated} \tag{1}$$

We used SRA to evaluate UV-dependent enrichment and $S/N$ of proteins recovered from the AGPC interphase of UV-crosslinked (0.4 J/cm$^2$, 254 nm) and non-crosslinked HeLa cells, using sequential interphase extraction to maximize RBP enrichment, as previously reported (Fig. 1b)[24]. To track RNA enrichments and RNase digestion efficacy, parallel gels were run and stained with SYBR Safe. Interestingly, while RNA exhibited UV-enrichment in the AGPC interphase (lower gel, red boxes), proteins showed only modest AGPC interphase UV-enrichment and were largely RNase-insensitive (upper gel, blue boxes), indicating that most of the proteins recovered from the AGPC interphase fraction of UV-irradiated cells were not crosslinked to RNA. These data demonstrate that UV-irradiation-dependent enrichment at the AGPC interphase alone is insufficient evidence for RNA-binding activity. To distinguish between contributions of noise (RBPs lacking crosslinked RNA) and background (non-RBPs) to this phenomenon, we compared known GO-annotated RBPs binding different RNA species with non-RBPs by SRA and immunoblot[1]. Here, proteins lacking GO-annotation (GO:RBP) are designated as non-RBPs, but this may not be due to a lack of RNA-binding activity ("Methods"). Protein targets were selected on the basis of prior literature, GO-RBP-annotation datasets, RNA-binding domain bioinformatic analysis, and UV cross-linking/CLIP sequencing frequency, to include canonical RBPs containing known RNA-binding domains (red font) and noncanonical RBPs (blue font) which were identified in

previous studies as candidate RBPs (i.e., scored as UV-enriched) but which lack known RNA-binding domains (Fig. 1c)[8,11,26,27]. Intriguingly, all canonical RBPs exhibited some degree of UV-dependent RNase sensitivity by SRA (red box) and were recovered from the AGPC interphase, whereas noncanonical RBPs and non-RBPs were either RNase-insensitive or interphase depleted following repeat extraction. In agreement with prior findings, most glycoproteins assayed (asterisks) exhibited UV-independent enrichment at the AGPC interphase[4], although ribophorin I (RPN1) did not (purple box; Fig. 1c). The finding that known RBPs lacking crosslinked RNA, e.g., nucleolin (NCL), and non-RBPs, e.g., β-tubulin (blue boxes) can score as UV-enriched provides a clear demonstration that orthogonal assays such as SRA are necessary for validation of candidate RBPs.

Even after six AGPC extractions, the established RBP nucleolin (NCL) reached an apparent $S/N$ enrichment limit (gold box; Fig. 1c). We attributed this to intrinsic limitations in the ability of repeated AGPC extraction to enhance the $S/N$ of a given RBP (Supplementary Fig. 2a and Supplementary Notes 3 and 4). This finding suggested that a subset of RNase-insensitive proteins at the final AGPC interphase may be bona fide RBPs with intrinsically low $S/N$. Combined, these observations revealed a need for enrichment methods that are selective for RNA-bound RBPs and thereby support quantitative $S/N$ determinations (Fig. 1d).

### LEAP-RBP provides efficient recovery of RNA-bound protein

Efforts to identify methods meeting this criterion yielded two approaches, INP (Isopropanol NaCl Precipitation) and LEAP-RBP (Liquid-Emulsion-Assisted-Purification of RNA-Bound Protein). Both are RNA-centric enrichment methods that enhance $S/N$ by SRA without significant signal loss (Fig. 2a, b), but only LEAP-RBP (L) clearly distinguishes RBPs with low $S/N$ (e.g., RPN1, TRAPα) from non-RBPs (red box; Fig. 2b). LEAP-RBP is notable for its high selectivity for RNA-protein complexes, with only negligible protein quantities displaying SDS-PAGE mobility in the absence of RNase digestion (Fig. 2a). This conclusion is further supported by assays of protein and RNA yields in the different enrichment methods, where LEAP-RBP yielded a substantially higher total RNA/protein than that obtained by methanol or isopropanol/NaCl precipitation. The utility of LEAP-RBP is readily

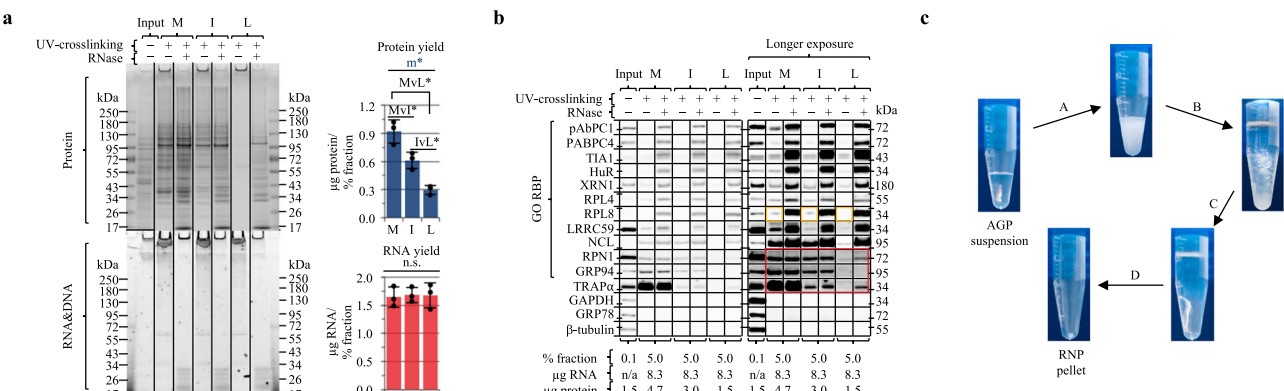

**Fig. 2 | LEAP-RBP provides rapid and efficient recovery of RNA-bound protein and identification of RBPs by SDS-PAGE RNase-sensitivity Assay (SRA).** AGPC acidic guanidinium thiocyanate–phenol–chloroform, M methanol precipitation (95% v/v), I INP, L LEAP-RBP, SRA SRA analysis. **a** Comparison of RNP fractions isolated by M, I, and L methods from final AGPC interphase suspensions of UV-crosslinked cells by SRA and Coomassie Blue (protein), SYBR Safe (RNA&DNA) staining. Protein (BCA) and RNA (UV-spectrophotometry) yields shown in the adjoining bar charts represent the mean ± 1 SD of three biologically independent samples, pooled (equivalent % fraction) for SRA analysis. Effect of isolation method on protein yield analyzed using one-way ANOVA (Fisher's PLSD post hoc test, unpaired, two-tailed, homoscedastic, no correction): significant main effect of

isolation method (m*), $F_{(2, 6)} = 41.91$, $P < 0.001$ (MvI*, $P = 0.018$; IvL*, $P = 0.001$; MvL*, $P < 0.001$). Effect of isolation method on RNA yield analyzed using a one-way ANOVA (Fisher's PLSD post hoc test, unpaired, two-tailed, homoscedastic, no correction): non-significant main effect of isolation method (n.s.), $F_{(2, 6)} = 0.04$, $P = 0.965$ (post hoc testing not applicable). **b** Immunoblot analysis of samples from **a**, sample compositions indicated in figure panel. **c** Pictorial representation of LEAP-RBP method: (A) Addition of chloroform and vortexing. (B) Addition of precipitation solution (LiCl and isopropanol), inversion and incubation for 1 minute; repeated 5+ times. (C) Vortexing (D) Centrifugation and rinsing with 95% methanol (v/v). Experiments (**a**, **b**) were performed twice with similar results. Source data are provided as a Source Data file.

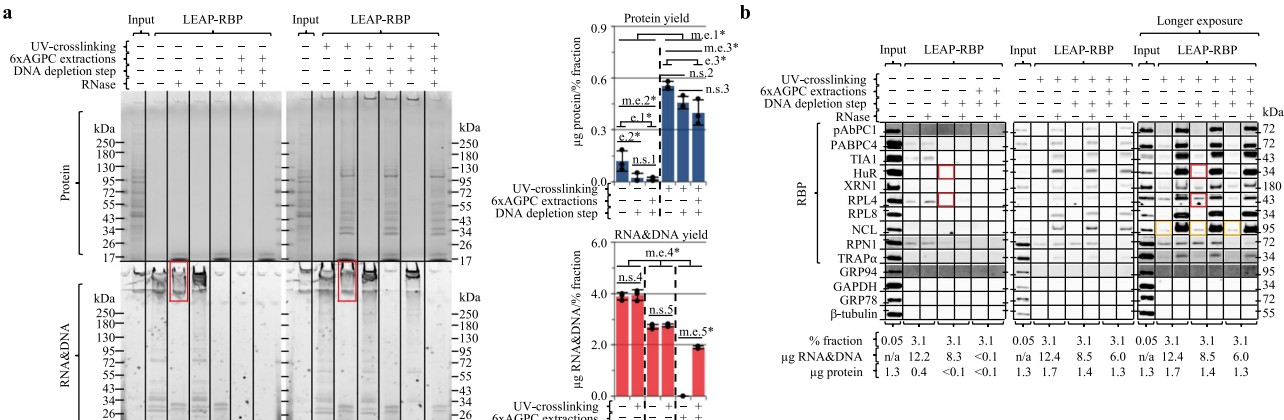

**Fig. 3 | Repeated AGPC extraction and DNA depletion step improves *S/N* and sensitivity of SRA.** AGPC acidic guanidinium thiocyanate–phenol–chloroform, L LEAP-RBP. Sample preps: s.p.1, L; s.p.2, L w/ DNA depletion step; s.p.3, repeated AGPC extraction and L w/ DNA depletion step. **a** SRA analysis of L fractions isolated from AGP input suspensions containing equivalent amounts of UV-crosslinked or non-crosslinked cells by s.p.1, s.p.2, s.p.3; parallel gels; Coomassie Blue (protein), SYBR Safe (RNA&DNA). Protein (BCA) and RNA&DNA (UV-spectrophotometry) yields shown in the adjoining bar charts represent the mean ± 1 SD of three biologically independent samples, pooled (equivalent % fraction) for SRA analysis. Effect of UV cross-linking and sample prep on protein yield of L analyzed using two-way ANOVA: significant interaction, $F_{(2, 12)} = 4.61$, $P = 0.033$. Subdivided by largest main effect (m.e.), UV cross-linking (m.e.1*, dashed line), $F_{(1, 12)} = 93.66$, $P < 0.001$. Effect of sample prep on protein yield of L for UV-crosslinked and non-crosslinked subgroups analyzed using independent one-way ANOVAs (Fisher's PLSD post hoc test, unpaired, two-tailed, homoscedastic, no correction): non-crosslinked, significant

m.e. of sample prep (m.e.2*), $F_{(2, 6)} = 6.07$, $P = 0.036$ (e.1*, $P = 0.023$; e.2*, $P = 0.023$; n.s.1, non-significant, $P = 0.998$); UV-crosslinked, significant m.e. of sample prep (m.e.3*), $F_{(2, 6)} = 6.86$, $P = 0.028$ (e.3*, $P = 0.010$; n.s.2, non-significant, $P = 0.076$; n.s.3, non-significant, $P = 0.173$). Effect of UV cross-linking and sample prep on RNA&DNA yield of L analyzed using two-way ANOVA: significant interaction, $F_{(2, 12)} = 719.45$, $P < 0.001$. Subdivided by largest m.e., sample prep (m.e.4*, dashed lines), $F_{(1, 12)} = 998.29$, $P < 0.001$. Effect of UV cross-linking on RNA&DNA yield of L for s.p.1, s.p.2, s.p.3 subgroups analyzed using independent one-way ANOVAs (Fisher's PLSD post hoc test, unpaired, two-tailed, homoscedastic, no correction): s.p.1, non-significant m.e. of UV-crosslinking (n.s.4), $F_{(1, 4)} = 0.14$, $P = 0.729$; s.p.2, non-significant m.e. of UV-crosslinking (n.s.5), $F_{(1, 4)} = 0.91$, $P = 0.393$; s.p.3, significant m.e. of UV-crosslinking (m.e.5*), $F_{(1, 4)} = 746.88$, $P < 0.001$. **b** Immunoblot analysis of samples from (**a**), sample compositions indicated in figure panel. Experiments (**a**, **b**) were performed four times with similar results. Source data are provided as a Source Data file.

apparent by SRA re-evaluation of a subset of previously identified candidate RBPs where for example the previously GO-annotated RBP and endoplasmic reticulum (ER) chaperone GRP94 scores as a false positive and the non-RBP ER integral membrane protein TRAPα as a false negative (Fig. 2b). The enhanced *S/N* of the LEAP-RBP method was achieved using a heterogeneous, lithium-supplemented solvent system which provides rapid and selective precipitation of RNA-bound proteins (Fig. 2c, Supplementary Figs. 3a–c and 4a–d, and "Methods")[28].

By virtue of high specificity, LEAP-RBP did not recover detectable protein from the final AGPC interphase suspension of non-crosslinked cells (Supplementary Fig. 5a). This finding allowed us to determine if protein recovery in the final AGPC interphase suspension of UV-crosslinked cells is restricted to protein–RNA adducts and/or can be UV-irradiation (signal)-dependent but independent of RNA cross-linking. In these experiments, LEAP-RBP was applied to AGP input suspensions containing UV cross-linked or non-cross-linked cells (Fig. 3a, b). Here the identification of contaminant DNA (red box; Fig. 3a) prompted development of a DNA depletion step utilizing a second LEAP step to further enhance *S/N* (Supplementary Fig. 5b). Recovery of protein in the AGPC interphase was primarily UV-dependent, with a small but significant difference between crosslinked samples prepared using a single LEAP and those prepared using the full protocol noted above (Fig. 3a). As expected, RNA recovery was mainly dependent on sample prep although UV-dependent following six AGPC extractions. Importantly, comparisons of the fractions prepared using the different protocols revealed similar RNase-sensitive protein profiles, thereby validating the critical assumption that all RNA-bound protein partitions to the AGPC interphase (Fig. 3a, b). Remarkably, and as a clear demonstration of the specificity and selectivity of LEAP-RBP for protein–RNA adducts, the *S/N* of most RBPs isolated by direct

LEAP-RBP treatment of AGP input suspensions were comparable to those isolated following six AGPC extractions (gold boxes; Fig. 3b). Together, these data establish the efficacy of the LEAP-RBP method and support the alternative hypothesis that recovery of noise is signal-dependent, not RNA-dependent (red boxes; Fig. 3b). In subsequent experiments we observed that RNA-free RBP recovery from AGP input suspensions was dependent on UV-dose and independent of total RNA and protein quantity (Supplementary Fig. 6a–c and Supplementary Note 4).

### Enhanced *S/N* decreases UV-enrichment* specificity

The improvements in *S/N* conferred by LEAP-RBP provided an opportunity to determine the effect of enhanced *S/N* on UV-enrichment* specificity for GO-annotated RBPs, where asterisks denote statistical significance. To this end, heavy SILAC-labeled crosslinked (CL) and light SILAC-labeled non-crosslinked (nCL) cells were pooled prior to processing to accurately quantify UV-dependent free protein recovery by LC–MS/MS and evaluate *S/N* (Fig. 4a and Supplementary Notes 5 and 6). UV-enriched* proteins were identified by generating $\log_2(CL/nCL)$ ratios with sum peptide intensities (SPI) as shown in Eq. (2) and testing against the null hypothesis that $\log_2(CL/nCL) = 0$ (Fig. 4b and Supplementary Data 2 and 3).

$$CL/nCL = SPI_{CL}/SPI_{nCL} \qquad (2)$$

For comparative purposes, INP isolation was performed on parallel samples (Supplementary Data 4 and 5). As shown in Fig. 4c, both INP and LEAP-RBP methods enriched for GO-annotated RBPs, but LEAP-RBP identified nearly twice as many UV-enriched* proteins (1794 vs 937) including 682 of the 719 UV-enriched* and 106 of the 156 non-enriched RBPs identified by INP (Fig. 4d). Notably, the LEAP-

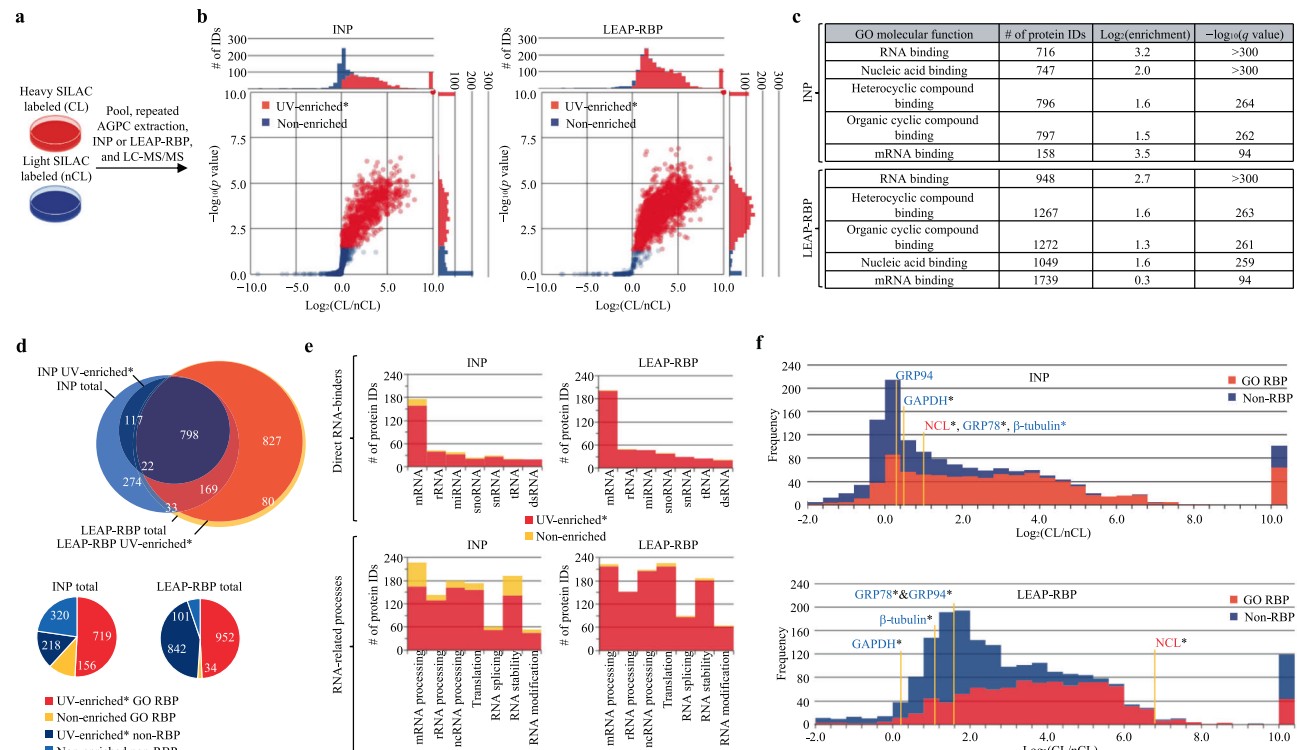

**Fig. 4 | Enhanced *S/N* decreases UV-enrichment\* specificity.** AGPC acidic guanidinium thiocyanate–phenol-chloroform, I INP, L LEAP-RBP, SILAC SILAC LC–MS/MS analysis, SRA SRA analysis, E\* significantly UV-enriched\*, NE not E\*. **a** Schematic illustration of the experimental approach. Heavy SILAC-labeled UV-crosslinked and light SILAC-labeled non-crosslinked cells were mixed prior to repeated AGPC extraction and isolation of RNP fractions by L or I methods. Graphic prepared in BioRender. **b** Volcano plots showing proteins identified as E\* (red) or NE (blue) in I or L fractions by SILAC. $\log_2(\text{CL}/\text{nCL})$ ratios were generated with $\text{SPI}_{\text{CL}}$ values and average $\text{SPI}_{\text{nCL}}$ values. Tested against the null hypothesis that protein $\log_2(\text{CL}/\text{nCL})$ ratios (*n* = 3 biologically independent samples) are equal to 0 using unpaired, upper-tailed, heteroscedastic *t* tests (RNA-binding proteins are expected to be recovered from UV-crosslinked cells in greater amounts). Correction for multiple hypothesis testing was performed using the Benjamini–Hochberg approach and a false-discovery rate of 5%. **c** GO-enrichment analysis of proteins identified as E\* in I or L fractions by SILAC (Fisher's Exact, two-tailed, correction for multiple

hypothesis testing performed using the Benjamini–Hochberg approach and a false-discovery rate of 5%). **d** Venn diagram showing overlap of total and E\* proteins identified in I and/or L fractions by SILAC. Pie charts showing the number of RBPs and non-RBPs identified as E\* or NE in I or L fractions by SILAC. For information on protein ID assignment and protein lists for each category or GO term used in this study, see Supplementary Data 6 and 7. **e** Stacked bar charts showing the number of proteins with annotated RNA-binding or RNA-related functions identified as E\* or NE in I or L fractions by SILAC. **f** Histograms showing average $\log_2(\text{CL}/\text{nCL})$ ratios of RBPs and non-RBPs identified in I or L fractions by SILAC. Font color for individual proteins reflects conclusions from SRA and immunoblot analysis of total clRNP fractions; red text: RNase-sensitive RBP; blue text: undetected or RNase-insensitive protein. Asterisks: E\*. I and L SILAC LC–MS/MS experiments (**a**) were performed once with three biologically independent samples for each SILAC label group (CL, nCL). Source data are provided as Source Data file.

RBP method identified a greater number of UV-enriched\* RBPs that bind all classes of RNA and/or regulate various RNA processes (Fig. 4e). However, only 270 of the 996 proteins exclusively UV-enriched\* by LEAP-RBP were GO-annotated RBPs, leading to a paradoxical decrease in UV-enrichment\* specificity as compared to the INP method (719/937 vs 952/1794). Exploring this further, we observed a large cluster of UV-enriched\* proteins, many lacking prior GO-annotation (i.e., non-RBPs), which were largely absent in INP fractions (Fig. 4f). We postulated that these may represent false-positives arising from low-frequency, non-specific UV cross-linking which are UV-enriched\* by the LEAP-RBP method by virtue of its high selectivity for protein–RNA complexes (Supplementary Fig. 6b and Supplementary Note 7)[29–31]. As a relevant example, the ER chaperone GRP94 was detected by immunoblot in INP fractions as an RNase-insensitive band but was exclusively UV-enriched\* by LEAP-RBP method while undergoing a 50-fold decrease in abundance. Non-RBPs (GAPDH, GRP78, and β-tubulin) were UV-enriched\* and undetectable by SRA in both fractions (Figs. 2b and 4f), highlighting the utility of this orthogonal validation method for assigning RNA-binding activity to UV-enriched\* proteins and providing a method for identifying likely low-frequency cross-linking events ("Methods").

## Enhanced *S/N* allows the detection of RBP–RNA occupancy dynamics

A critical observation from the SILAC LC–MS/MS studies noted above is that while CL/nCL ratios provide a measure of UV-dependent enrichment, *S/N* ratio determinations reveal RNA-bound protein contributions across SILAC channels. This relationship is depicted in Fig. 5a and described by Eq. (3).

$$S/N = \text{SPI}_S/\text{SPI}_N = (\text{SPI}_{\text{CL}} - \text{SPI}_{\text{nCL}})/(2 * \text{SPI}_{\text{nCL}}) \tag{3}$$

As graphically depicted in Fig. 5b, c, a threefold enrichment or $\log_2(S/N)$ ratio of 0 indicates equal amounts of RNA-bound (*S*) and unbound (*N*) counterparts (50% RNA-bound) while a $\log_2(\text{CL}/\text{nCL}) \leq 0$ indicates an absence of RNA-binding activity (when $S = 0$, $N = B$ or background); this relationship is described by Eq. (4):

$$\text{\%RNA-bound} = S/(S + N) * 100 \tag{4}$$

As is apparent in the LC–MS/MS analysis, *S/N* is inextricably linked to the ability to detect a change in observed quantity $\Delta\log_2(\text{SPI}_O)$ in

response to a change in RNA-bound quantity $\Delta\log_2(SPI_S)$. This relationship is depicted in Fig. 5d, illustrating a large decrease in statistical power for proteins displaying negative $\log_2(S/N)$ ratios; these relationships are described by Eqs. (5–7) (Fig. 5e, "Methods").

$$\Delta\log_2(O) = \log_2(S+N)_{\text{final}} - \log_2(S+N)_{\text{initial}} \quad (5)$$

$$\text{Detectable } \Delta\log_2(O) = t_{0.95} * \text{SEM of } \log_2(O) \quad (6)$$

$$\text{Detectable } \Delta\log_2(S) = \log_2(2^{\wedge}\text{detectable }\Delta\log_2(O) - N/O) - \log_2(S/O) \quad (7)$$

A key conclusion evident from this analysis is that RBPs displaying different $S/N$ ratios can be UV-enriched* but the ability to detect $\Delta\log_2(S)$ could differ substantially. These concepts are illustrated by comparing the RNase sensitivity ($S/N$) of RBPs by SRA with their SILAC LC−MS/MS-derived $\log_2(S/N)$ ratios. In principle, RNase sensitivity represents a change in RNA-bound quantity ($|S|$) when noise is constant ($N_{\text{untreated}} = N_{\text{RNase}}$); Eq. (1). This is analogous to Eq. (5) if $S_{\text{initial}} = 0$ and $N_{\text{initial}} = N_{\text{final}}$. Experimental examples of these relationships are depicted in Fig. 5f, where both the INP and LEAP-RBP methods identified NCL, and the ER membrane proteins RPN1 and TRAPα as UV-enriched*, but RNase sensitivity was more clearly evident in the LEAP-RBP fractions, which also display higher $\log_2(S/N)$ ratios by SILAC LC−MS/MS. In contrast, RBPs displaying comparable RNase sensitivity (e.g., the ER membrane protein LRRC59) have comparable $\log_2(S/N)$ ratios. Therefore, a change in RNA-bound quantity ($S$) for RBPs such as NCL, RPN1, and TRAPα will have a larger effect on their observed quantity ($O$) in LEAP-RBP fractions by LC−MS/MS (Supplementary Notes 5 and 6).

## LEAP-RBP displays high method specificity for RNA-bound RBPs

As illustrated in Fig. 1b, e, current AGPC methods are hampered by considerable noise contributions; the interphase fractions contain many RNase-insensitive (free) proteins evident by SRA and Coomassie Blue staining. In contrast, AGPC interphase fractions enriched by the LEAP-RBP protocol would be expected to be both depleted in free proteins and thus contain a higher percentage of RNA-bound protein. This hypothesis was tested on INP and LEAP-RBP fractions using the Total Protein Approach (TPA), which provides protein abundance as a percentage of the total protein in the sample (total SPI), as described by Eq. (8)[32–34].

$$\%TP = (SPI/\text{total } SPI) * 100 \quad (8)$$

By extension of Eqs. (3) and (4), TPA can be used to determine the abundance of RNA-bound (%TP$_S$) and free proteins (%TP$_N$) as a percentage of total SPI, as described by Eqs. (9–11).

$$\%TP_S = SPI_S/SPI_O * \%TP \quad (9)$$

$$\%TP_N = \%TP - \%TP_S \quad (10)$$

$$\%TP_N = \%TP_B \text{ when } S = 0 \quad (11)$$

Cumulatively, %TP$_S$ and %TP$_N$ represent the abundance of total RNA-bound (total SPI$_S$) and free protein (total SPI$_N$) in the sample. By this approach, 91% of the total protein in LEAP-RBP fractions is RNA-bound compared to 47% for INP. This is consistent with differences in RNP composition (μg protein/μg RNA), though assumes equal noise-partitioning between SILAC channels (Fig. 5g). This finding validates Eq. (3); not assuming equal noise-partitioning overestimates the amount of RNA-bound protein in INP fractions by -60% (Source Data

Fig. 5g). Remarkably, GO-annotated RBPs represent 53% of proteins identified as UV-enriched* in LEAP-RBP fractions but contribute 98.3% of total SPI$_S$ (Fig. 5h). By comparison, the %TP$_S$ of INP fractions is lower (47), but RBPs still contribute 98.6% of total SPI$_S$.

Estimating the abundance of RNA-bound proteins as a percentage of total RNA-bound protein in the sample (total SPI$_S$) can be represented by %TP$_{(S)}$, where the parenthetical text denotes the identity of the total protein population ("Methods"). While %TP$_S$ of INP fractions (47) is less than LEAP-RBP fractions (91), both methods recover near 100% of RNA-bound protein (I or L vs M, RNA yield; Fig. 2a); therefore, comparable %TP$_{(S)}$ contributions from RBPs (-98.5) and non-RBPs (-1.5) in both fractions is indicative of high UV cross-linking specificity. The disparity in %TP$_S$ between LEAP-RBP and INP fractions is not, however, readily illustrated by current analytical methods (Fig. 4b–f). To address this discrepancy, the abundance ($\log_{10}$(%TP)) of RBPs and non-RBPs identified by INP and LEAP-RBP were evaluated as a function of their $S/N$ ratios ($\log_2(S/N)$) (Fig. 6a, b). As expected, an increase in %TP$_S$ is illustrated by the enhanced enrichment efficiency ($S/N$) of both RBPs and non-RBPs in LEAP-RBP fractions as compared to INP and higher abundance (%TP) of RBPs relative to non-RBPs (Supplementary Note 8). The increase in %TP$_S$ was attributed to a marked decrease in free protein recovery by the LEAP-RBP method. Consequently, LEAP-RBP provides a lower limit of detection (%TP range), thus resulting in identification of many low-abundance proteins not observed in INP fractions (exclusive; Fig. 6c). Because most of these low-abundance proteins are not GO-annotated as RBPs and UV-enriched*, enhanced $S/N$ results in the paradoxical decrease in UV-enrichment* specificity despite a favorable increase in %TP$_S$ ("Methods").

## High $S/N$ and %TP distinguish bona fide RBPs

As noted above, we postulated that non-specific UV cross-linking, combined with the enhanced $S/N$ provided by the LEAP-RBP method, results in the UV-enrichment* of low-abundance non-RBPs. In support of this hypothesis, all non-RBPs (undetectable by SRA) were UV-enriched* but display low $S/N$ ratios and are less abundant than the majority of RNase-sensitive RBPs (Fig. 6d). Indeed, GO-annotated RBPs display significantly higher $S/N$ ratios and were significantly more abundant than non-RBPs by either method (Fig. 6e), but the latter was more apparent with LEAP-RBP (high %TP$_S$). Critically, these results demonstrate that UV-enriched* RBPs can be distinguished from UV-enriched* non-RBPs in LEAP-RBP fractions by their enrichment efficiencies ($S/N$) and abundance (%TP).

To help distinguish high and low-confidence RNA-binding proteins, we propose a ranking system based on an RBP-confidence score or RCS, where RCS = $\log_2(S/N) * \log_{10}(\%TP)$. In practice, RCS ranking prioritizes $S/N$ over protein abundance and places proteins with $S/N$ ratios <1 at lower ordinal rank (Fig. 6f). This scoring system accurately ranked proteins assayed by immunoblot (Fig. 6g) and placed the noncanonical ER RNA-binding protein LRRC59 among the top ten most confident "enigmRBPs" of the 163 detected in LEAP-RBP fractions (Fig. 6h)[8]. To facilitate rapid discovery of other noncanonical RBPs by SRA analysis of LEAP-RBP fractions, a full list of identified proteins, their RCS rank, and parameters discussed are included in the provided Source Data or in Supplementary Data 3.

## LEAP-RBP allows robust and sensitive detection of $\Delta\log_2(S)$

During comparative RBP profiling experiments, enhanced $S/N$ and high %TP$_S$ allows accurate assessment of RNA-bound protein abundance. By comparison to INP, which mirrors current AGPC methods, the LEAP-RBP method allows more sensitive detection of $\Delta\log_2(S)$, representing the fold-change in RNA-bound protein quantity ($S$) necessary to reject the null hypothesis that $\Delta\log_2(S+N) = 0$ (Supplementary Fig. 7a–c). This enables the use of stringent $S/N$-based criteria ($S/N > 3$, 75% RNA-bound) to limit detection of $\Delta\log_2(N)$ and increase statistical power by reducing multiple hypothesis testing (Supplementary Note 6b, c). In

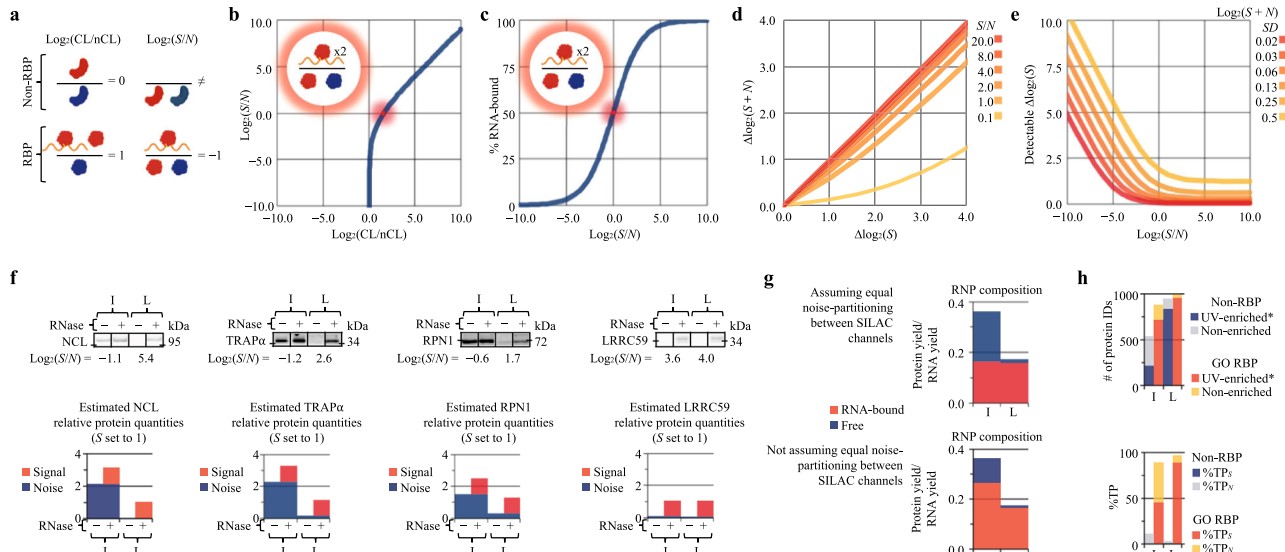

**Fig. 5 | SILAC LC−MS/MS analysis of LEAP-RBP fractions demonstrates high RNA-bound protein enrichment.** I INP, L LEAP-RBP, SILAC SILAC LC−MS/MS analysis, SRA SRA analysis, E* significantly UV-enriched*, NE not E*. **a** Predicted relationship between CL/nCL and $S/N$ ratios for a specific RBP or non-RBP identified in RNP fractions isolated from pooled UV-crosslinked (red) and non-crosslinked (blue) samples by SILAC. **b** Relationship from **a** shown as a continuous function. **c** Predicted relationship between $\log_2(S/N)$ and the percentage (%) of observed protein quantity during SILAC LC−MS/MS experiments that is RNA-bound. **d** Predicted change in observed quantity $\Delta\log_2(S + N)$ in response to a change in RNA-bound quantity $\Delta\log_2(S)$ for proteins displaying increasing $S/N$ ratios. **e** Estimated $\Delta\log_2(S)$ to successfully reject the null hypothesis that $\Delta\log_2(S + N) = 0$ increases with decreasing $S/N$ and increasing $SD$. **f** $\log_2(S/N)$ ratios quantified by SILAC were used to estimate the observed quantity of nucleolin and three non-canonical endoplasmic reticulum candidate RBPs in RNase-treated ($|S| + N$) and untreated ($N$) I and L fractions by SDS-PAGE and immunoblot. Immunoblots were selected from Fig. 2b; and performed twice with similar results. **g** Estimated amount of RNA-bound and free protein in I and L fractions by SILAC is more accurate when assuming equal noise-partitioning between SILAC channels. **h** Stacked bar charts showing the number of RBPs and non-RBPs identified as E* or NE by each method, or the estimated %TP$_S$ and %TP$_N$ contributions of RBPs and non-RBPs in each RNP fraction. Graphics prepared in BioRender (**a–c**). Source data are provided as a Source Data file.

addition, the high %TP$_S$ of LEAP-RBP fractions decreases the variability of mean-normalized samples by decreasing free protein contributions (%TP$_N$) of the most abundant proteins in the sample (Supplementary Fig. 7d–f). This allows the least computationally intensive and most accurate label-free LC−MS/MS approach for the detection of $\Delta\log_2(S)$ (Supplementary Note 4f)[35,36].

To illustrate these points, we performed a comparative LEAP-RBP experiment to examine the effect of dynamic translatome remodeling on global RBP−RNA occupancy states. Using harringtonine (HT), a selective inhibitor of translation initiation, we sought to identify RBPs whose interactions with mRNAs were either sensitive to ribosome occupancy (= translation-state-dependent interactors) or whose mRNA association was not sensitive to ribosome occupancy status (= translation-state-independent interactors). Through inhibitory interactions at the ribosomal A-site, HT induces global polyribosome runoff, to yield monosomes bearing initiation codon-locked 80 S ribosomes[37,38]. We first confirmed harringtonine efficacy by sucrose gradient density gradient polyribosome profiling (Fig. 7a and "Methods"). As expected, HT treatment resulted in the pronounced accumulation of 80 S monosome/mRNA complexes. Biological triplicate control and HT-treated cell cultures were then prepared by UV irradiation, LEAP-RBP fractions isolated, and comparisons of input (total protein) and clRNP fractions (total RNA-bound protein) were performed by LC−MS/MS analysis. Analysis of the proteomic datasets, with or without the proposed $S/N$ limit, identified 23 RBPs displaying a significant change in RNA occupancy (purple data points; Fig. 7b, c and Supplementary Data 19 and 20). Application of the $S/N$-based criteria introduced above significantly improved the specificity and sensitivity of our analysis by limiting inclusion of proteins with significant free protein contributions and revealing additional RBPs with known roles in translation and ribonucleoprotein assembly, as demonstrated by GO analysis (gold vs teal markers; Fig. 7d, e). These findings are further detailed in Fig. 7f, which depicts protein gene name, fractional RNA

occupancy, fold-change occupancy in response to HT treatment, and the presence of known RNA-binding domains and/or GO-annotated RBP status. It's noteworthy that both the observed fold-change differences as well as the fraction of RBPs whose RNA association is modulated by CDS ribosome occupancy status are generally quite modest. This suggests that CDS translation is largely determined by initiation rate frequency and regulation of ribosome processivity, with RBP-dependent regulation largely biased to 5′/3′ UTR interactions[39,40]. Subsequent comparisons of the LEAP-RBPs fractions by SRA analysis confirmed that the RBPs whose association with mRNAs was modulated by ribosome loading status were due to differences in RNA-bound abundance (Fig. 7g); comparing RNP fractions yielded similar results (Supplementary Fig. 8a, b). Notably, we identified RBPs such as RPS3, UPF1, SND1, and HDLBP which were previously reported to display decreased RNA binding following treatment with the oxidative stressor sodium arsenite[3], and include additional proteins such as ABCF3 whose RNA binding is unannotated and which was identified exclusively using the more stringent $S/N$-based criteria (Fig. 7f, g). Importantly, and as a demonstration of the utility of the signal-based analytical approach, RBPs lying near the proposed $S/N$ limit of 3, such as the translation elongation factor eEF2, displayed translation-state-dependent differences in free protein recovery (gold box; Fig. 7g and Supplementary Fig. 8a). We also note that the LEAP-RBP method does not bias towards more abundant proteins, as those displaying $S/N$ ratios >3 in LEAP-RBP fractions were found to be significantly less abundant than others in total protein (input) fractions; Kruskal–Wallis, $H(1) = 7.82$, $P = 0.005$; Source Data Fig. 7h).

As an additional demonstration of the utility of LEAP-RBP method for studying context- or cell type-dependent differences in RNA-bound proteomes, a LEAP-RBP analysis was performed on four different cell lines: human cervical cancer cells (HeLa), human embryonic kidney cells (293T), human hepatocyte-derived carcinoma cells (Huh7), and a rat pancreatic insulinoma cell line (832/13) (Supplementary Fig. 8c). Of

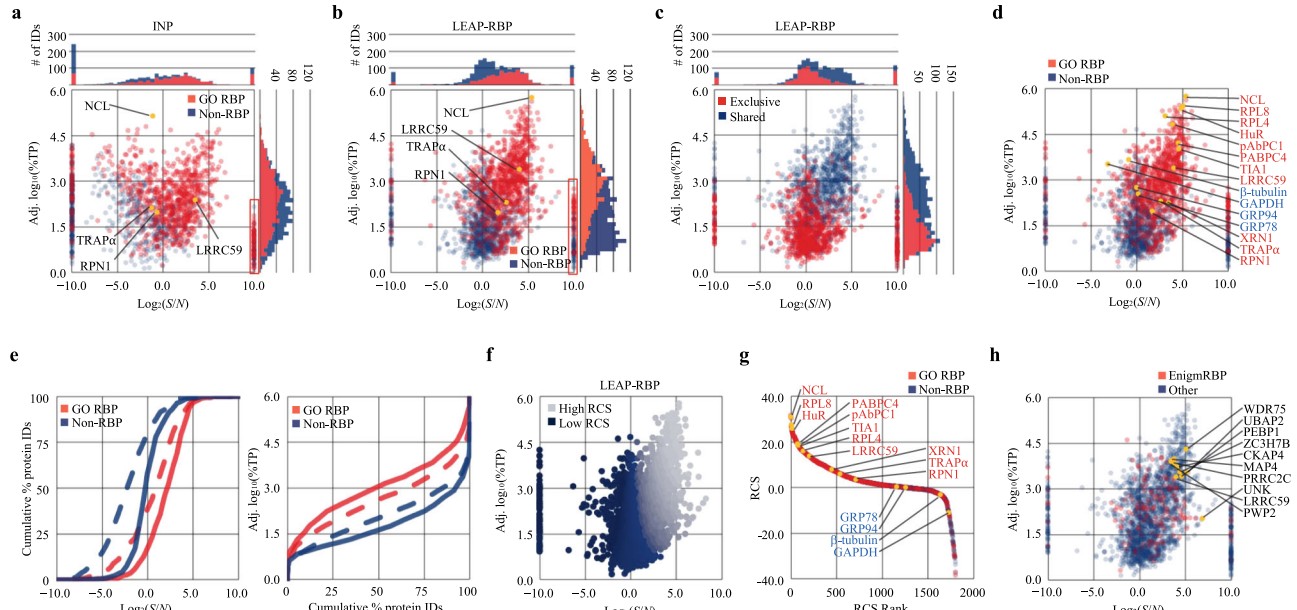

**Fig. 6 | High method specificity for RNA-bound RBPs allows accurate RCS ranking of RNA-binding proteins.** I INP, L LEAP-RBP, SILAC SILAC LC–MS/MS analysis, SRA SRA analysis. Specificity of I (**a**) and L (**b**) methods for enrichment of RNA-bound RBPs was evaluated by comparing observed abundances of RBPs and non-RBPs as a function of their $\log_2(S/N)$ ratios (SILAC). Distributions of immuno-blot targets reported in Fig. 5f demonstrates significance of vertical intercept. **c** L exclusives fall within the lower range of detection (%TP range). **d, b** With labeled immunoblot targets; red font: RNase-sensitive RBP; blue text: undetected or RNase-insensitive protein. **e** Cumulative frequency curves for RBPs (red) and non-RBPs (blue) identified in I (dashed) and L (solid) fractions as a function of $\log_2(S/N)$ or $\log_{10}(\%TP)$ (SILAC). Effect of GO-annotation (GO:RBP) on analyzed using independent Kruskal–Wallis tests: significant effect of GO-annotation on $\log_2(S/N)$ ratios of proteins identified in I and L fractions (SILAC), $H(1) = 194.63$, $P < 0.001$, $H(1) = 436.62$, $P < 0.001$, respectively; significant effect of GO-annotation on $\log_{10}(\%$ TP) of proteins identified in I and L fractions (SILAC), $H(1) = 111.11$, $P < 0.001$, $H(1) = 632.29$, $P < 0.001$, respectively. **f, b** With color overlay based on ordinal RCS rank. **g** RCS as a function of RCS rank. Protein IDs represent proteins examined by SRA and immunoblot in Fig. 2b; red font: RNase-sensitive RBP; blue font: unde-tected or RNase-insensitive proteins with positive RCS (i.e., $\log_2(S/N) > 0$) are con-sidered more representative of their RNA-bound counterparts. **h, f** With top 10 RCS ranked enigmRNPs. Source data are provided as a Source Data file.

note, input and clRNP fractions isolated from the different cell lines displayed discernible differences in total and RNA-bound proteomes by SRA analysis (Fig. 7i). Immunoblot analysis revealed more con-stitutive RNA binders (blue boxes), differentially abundant RNA bin-ders (red boxes), and dynamic RNA binders (gold boxes) whose relative RNA-bound abundance differs from their relative total abun-dance (Fig. 7j and "Methods")[41,42]. This last category includes the TIA-1a isoform displaying lower total abundance in rat 832/13 cells but com-parable RNA-bound abundance (gold asterisk) (Supplementary Note 6d).

Interestingly, integral membrane ER resident RBPs (e.g., LRRC59, RPN1, TRAPα) consistently displayed higher RNA-bound protein abundance in rat insulinoma (pancreatic β) cells (832/13) without a comparable change in total abundance (Fig. 7j). As high secretory capacity cells capable of glucose-stimulated insulin secretion, 832/13 cells have high relative translation at the ER membrane, further indi-cated by an increased abundance of the ER-luminal chaperones GRP94 and GRP78 (red asterisk)[43–45]. We speculate that their increased RNA-bound protein abundance may indicate a regulatory role in translation at the ER and/or increased local interactions with their RNA compo-nents (rRNA and mRNA)[46–49].

**Benchmarking RNA-centric methods with signal-based metrics**
Comparisons of current RNA-centric approaches include overlap (Venn) analysis of UV-enriched* proteins but lack metrics such as $S/N$ or %TP_S (Supplementary Note 7). To ascertain the broader utility of the LEAP-RBP method and $S/N$-based rubrics, we performed benchmark comparisons of LEAP-RBP to three organic phase separation methods (XRNAX, OOPs, and Ptex), one solid phase separation method (TRAPP), and one affinity-based separation method (RIC)[2–4,21,50]. Except for RIC, which selects for poly(A) RNA-binding proteins, these

methods aim to isolate total RNA-protein interactomes. RNP fractions were isolated from UV-crosslinked and non-crosslinked cells accord-ing to each of the published methods (Fig. 8a and Supplementary Note 9); $S/N$, %TP_S, and yield were evaluated by SRA analysis of RNP fractions (Fig. 8b, c) and the findings were compared with available MS data (Fig. 8d, e)[3,21,50–52]. Of note, UV-dependent enrichment of free protein is expected to increase non-specific (i.e., non-RBP) %TP_(S) contributions during non-SILAC LC–MS/MS experiments (Supple-mentary Notes 7e and 8).

By SRA analysis, XRNAX and OOPs display low to moderate UV-dependent enrichment of free protein (blue boxes; Fig. 8b) and low $S/N$ without signal loss (gold and red boxes; Fig. 8c). Ptex shows moderate UV-dependent enrichment (blue boxes; Fig. 8b), but recovered protein is not RNA-bound (gold boxes; Fig. 8c). This is consistent with prior data (Supplementary Note 8b), and available MS data (non-SILAC) indicating low %TP_S (23.2) and high non-specific %TP_(S) contributions (28.4; Fig. 8e). XRNAX utilizes partial tryptic digestion and repeated TiO_2/SiO_2 enrichment to further enrich RNA-bound peptides prior to SILAC LC–MS/MS[3]. This procedure effectively enhances $S/N$, resulting in a favorable increase in %TP_S (70.5, %TP_(S, non-RBPs) = 2.6; Fig. 8d, e); however, proteins such as β-tubulin or GRP94 remain UV-enriched* and because of the trypsinolysis step cannot be subsequently ortho-gonally validated by methods such as SRA (Supplementary Data 1). OOPs distinguish UV-enriched RBPs (gold boxes) from non-RBPs (blue boxes) at the interphase by their RNase-dependent partitioning into the organic phase (Fig. 8c), but continued partitioning of free protein decreases $S/N$[4]. It should be noted that the reported methodology for the OOPs protocol includes non-SILAC comparisons which are expec-ted to result in non-specific enrichment of free protein (purple boxes; Fig. 8c)[51,53,54]. Available MS data showing high non-specific %TP_(S) con-tributions (26.5; %TP_S = 73.5) supports this assessment (Fig. 8d, e)[51].

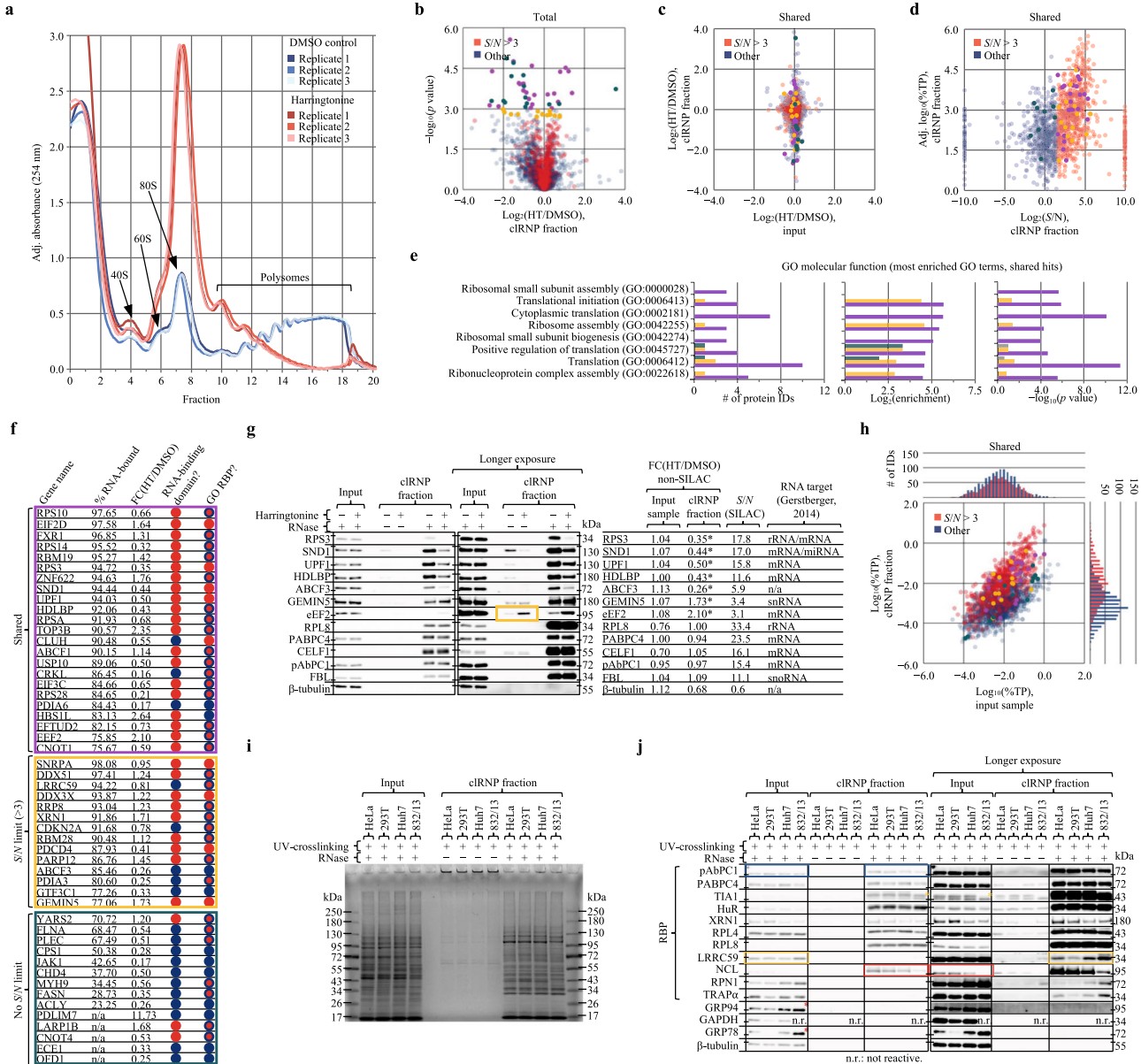

**Fig. 7 | Profiling the relationship between mRNA CDS ribosome occupancy state and RBP–RNA interactome dynamics.** SILAC SILAC LC–MS/MS analysis, non-SILAC LC–MS/MS analysis, SRA SRA analysis, clRNP in clRNP fraction, input in input sample. **a** Ribosome profiling of HeLa cells after 30-min treatment with 2 μg/mL harringtonine (HT) or DMSO control, $n = 3$ biologically independent samples. **b–f, h** Red markers: proteins which display $S/N$ ratios >3 (clRNP, SILAC) shown in Fig. 6b; blue markers: other; additional labels and color overlay ("label": color marker) for proteins displaying a significant difference in RNA occupancy (HT/DMSO, clRNP, unpaired, two-tailed, homoscedastic $t$ test; $n = 3$ biologically independent samples) after FDR-correction (Benjamini–Hochberg approach, false-discovery rate of 5%) with $S/N$ limit ("gold hits": gold marker), without $S/N$ limit ("teal hits": teal marker), or both ("purple hits": purple marker). **b** Volcano plot showing significance values of proteins before FDR-correction as a function of $\log_2$(HT/DMSO), (clRNP, non-SILAC). **c** Scatterplot comparing $\log_2$(HT/DMSO) ratios of proteins identified in both input and clRNP fractions (shared, non-SILAC). **d** Observed protein abundances (clRNP, non-SILAC) as a function of $\log_2$($S/N$)

(clRNP, SILAC). **e** GO-enrichment analysis of gold, teal, and purple hits; shown are the top ten most enriched GO terms for purple hits (Fisher's Exact, two-tailed, no correction). **f** Category-distributed comparison of hits showing % RNA-bound (clRNP, SILAC), observed fold-change (clRNP, non-SILAC), presence of RNA-binding domain (blue: no; red: yes); and RBP-annotation (GO:RBP) status (blue: no; red: yes; red with blue outline: yes, inferred from UV-enrichment* in RIC-like (non-SILAC) experiments[2,55]). **g** SRA and immunoblot of pooled (equivalent μg protein) input and clRNP fractions quantified in Supplementary Fig. 8d–g and analyzed by LC–MS/MS (non-SILAC); includes observed FC(HT/DMSO) and $S/N$ ratios (clRNP, SILAC); asterisk: significant FC(HT/DMSO) after FDR-correction with $S/N$ limit. **h** Scatterplot comparing observed protein abundances in input and clRNP fractions (shared, non-SILAC). **i** Comparison of input and clRNP fractions isolated from four different cell lines by SRA and Coomassie Blue (protein), SYBR Safe (RNA&DNA) staining. **j** Immunoblot analysis of samples from (**i**), $n = 3$ biologically independent samples, pooled (equivalent % fraction) for SRA. Experiments were performed once (**a–h**), or three times with similar results (**i, j**). Source data are provided as a Source Data file.

By SRA analysis, both TRAPP and RIC display high UV-dependent enrichment of RNase-sensitive protein (blue boxes; Fig. 8b), and signal-dependent recovery of noise (gold boxes; Source Data Fig. 5c), which are both indicative of high %TP$_S$ (Supplementary Note 8a). Comparisons using a higher percentage of RNP fractions isolated from UV-crosslinked cells suggest LEAP-RBP achieves higher $S/N$, %TP$_S$, and yield than TRAPP or RIC methods (goldbox; Fig. 9a, b). Available MS data from RIC (non-SILAC) and TRAPP (SILAC) experiments support these observations and indicates that the LEAP-RBP method provides more sensitive detection of $\Delta\log_2(S)$ for a greater number of proteins

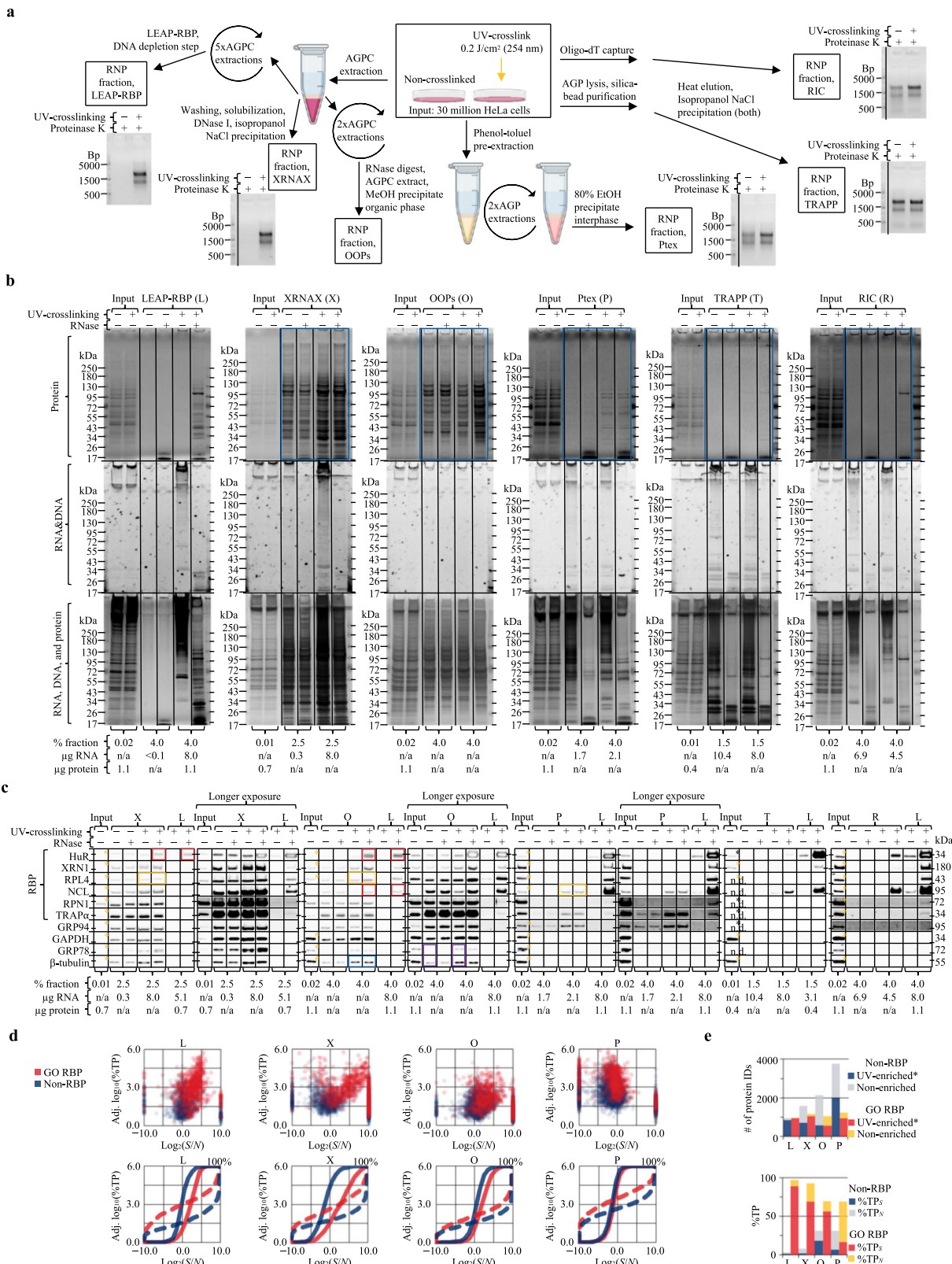

(Fig. 9c–g). It's important to note that available MS data for TRAPP is from yeast; therefore, non-specific $\%TP_{(S)}$ contributions are due to incomplete RBP annotations (GO:RBP) for ribosomal proteins (Supplementary Note 8b). Curiously, TRAPP shows efficient recovery of RNA-bound ribosomal proteins (blue boxes) but relatively poor recovery of others (red boxes; Fig. 9b), consistent with available MS data (Fig. 9h, i). We speculate that this RBP-specific signal loss may

occur because of the stringent denaturing washes employed in the purification process. The optimal amount of UV-energy (254 nm) for TRAPP was discussed by these authors, based on GO analysis of UV-enriched* proteins, with "lower doses being potentially less noisy but at the cost of recovering fewer proteins with annotated functions in RNA biology"[50]. From an $S/N$ perspective, higher UV doses increase $\%TP_S$ but decrease UV-enrichment* specificity (Fig. 9j).

**Fig. 8 | Benchmark comparisons illustrate utility of LEAP-RBP and $S/N$, $\%TP_S$ metrics.** N/AGPC neutral/acidic guanidinium thiocyanate–phenol-chloroform, AGP acidic guanidinium thiocyanate–phenol, L LEAP-RBP, X XRNAX, O OOPs, P Ptex, T TRAPP, R RIC, SILAC, SILAC LC–MS/MS analysis, SRA SRA analysis, E* significantly UV-enriched*, NE not E*. **a** Experimental flow outlining the main steps of LEAP-RBP and five referenced RNA-centric methods. TBE gel analysis was performed on 1 μg of RNA isolated by NGPC extraction of proteinase K-treated RNP fractions isolated from UV-crosslinked cells and an equivalent % fraction of their corresponding non-crosslinked samples. UV-dependence of protein and RNA recovery as well as $S/N$ were evaluated by SRA with SYBR Safe (RNA&DNA), Coomassie Blue (protein), and silver stain (RNA, DNA, and protein) staining (**b**) or immunoblot (**c**). Sample compositions and/or normalization values are indicated in figure panels **b, c**. Immunoblot targets found UV-enriched* in referenced studies (Supplementary Data 1) were marked with gold asterisks in **c**; black asterisk (T): no yeast homolog, hence n/a; n.d.: not detected. **d** Specificity and selectivity of the different methods for RNA-bound RBPs was evaluated by comparing observed abundances of RBPs and non-RBPs as a function of their $\log_2(S/N)$ ratios (SILAC and non-SILAC). Differences in corresponding frequency curves plotted as a function of $\log_2(S/N)$ (solid) or $\log_{10}(\%TP)$ (dashed) indicate differences in protein–RNA adduct enrichment efficiency ($S/N$) or abundance ($\%TP$), respectively. **e** Stacked bar charts showing the number of RBPs and non-RBPs identified as E* or NE by each method, or the estimated $\%TP_S$ and $\%TP_N$ contributions of RBPs and non-RBPs in each RNP fraction. Experiments were performed once (**a**–**c**); $n = 1$ (X, O, P, T, R); $n = 3$ biologically independent samples (L), pooled (equivalent % fraction) for SRA. Input samples isolated from AGP input suspensions containing UV-crosslinked and/or non-crosslinked HeLa cells were used as inter-run controls during SRA analysis, $n = 1$. Complete MS datasets and parameters discussed in this study for each of the referenced studies are provided as individual Excel files (Supplementary Data 10–20). Source data are provided as a Source Data file.

As expected, RIC recovers RNA-bound mRNA binders more efficiently than TRAPP (red boxes; Fig. 9b), but still recovers RNA-bound ribosomal proteins (blue box) and rRNA (Fig. 8a). More stringent protocols utilizing LNA probe capture (eRIC) have been developed to address these concerns (Fig. 9k, l)[52]. However, unless rRNA is entirely removed from the sample, UV-enrichment* of rRNA-bound protein is likely to occur. Indeed, ribosomal proteins are less abundant in eRIC fractions compared to RIC (1.2 vs 4.1) but are detected and UV-enriched* (Fig. 9m). A similar trend is observed for exclusive DNA binders which are less abundant in LEAP-RBP fractions compared to other methods but were also UV-enriched* in greater numbers (Fig. 9n). Surprisingly, while RIC and eRIC are aimed at selective recovery of mRNA binders, LEAP-RBP identified a greater number of mRNA binders and they are more abundant in LEAP-RBP fractions (Fig. 9i, m). Nonetheless, ribosomal proteins are less abundant in RIC (4.1) and eRIC (1.2) fractions than LEAP-RBP fractions (7.9). These data indicate that eRIC—and to a lesser extent, RIC—are selective for mRNA binders, while LEAP-RBP provides a comprehensive assessment of the RNA-bound proteome.

## Discussion

We report LEAP-RBP as a highly selective and cost-efficient method for the purification of RNA-bound protein from biological samples. We identify $S/N$ and $\%TP_S$ (RNA-bound protein abundance) as key metrics for evaluating RNA-bound protein enrichment and method specificity for RNA-bound RBPs. Importantly, we present practical, experimentally accessible strategies for the accurate determination of in vivo RNA-binding activity and for robust profiling of RNA-bound proteomes at steady-state and following dynamic cell state transitions.

A $S/N$-based comparative analysis of RBP profiling data generated by LEAP-RBP and other RNA-centric methods revealed the complexity and challenges inherent in accurate identification of direct RNA binders based on their UV-enrichment* and assessment of RNA-binding activity based on protein recovery alone[2,3,21,26,42,50–52,55,56]. These method-intrinsic challenges can be compounded by low method specificity and/or non-SILAC comparisons, both of which result in apparent UV-dependent enrichment of free protein. While RBP enrichment methods utilizing SILAC LC–MS/MS and stringent sample washes achieve higher $\%TP_S$, the benchmark comparisons performed here reveal both reduced yields and biases in signal recovery which were previously unrecognized. These observations provide insights into why non-poly(A) RNA binders such as ribosomal proteins can represent a large fraction of MS spectra[52]. The high selectivity of LEAP-RBP achieves high $\%TP_S$ without the need for high-stringency washes, and thus provides a more specific, selective portrait RNA interactomes.

RNA-binding proteins containing well-established canonical RNA-binding domains display higher $S/N$ ratios and RNA-bound abundance, which greatly simplifies study of their RNA interactome dynamics, largely independent of limitations in existing methods. A primary challenge in the field however is the study of candidate RBPs lacking canonical RNA-binding domains, known functions in RNA biology, relatively low UV cross-linking frequencies, and/or significant free protein contributions in phase separation-based RNA-centric methods, all of which can hinder interpretation as well as meta-analysis of RNA interactomes and their dynamic regulation (Supplementary Notes 4–6). The signal-based analytical framework described here addresses these limitations and provides experimental avenues for the discovery and study of potentially novel RNA interactors with previously unknown roles in RNA biology. We highlight the noncanonical integral membrane RBP candidates LRRC59, TRAPα, and RPN1, all of which are resident proteins of the endoplasmic reticulum and which may function in mRNA and/or ribosome localization to the ER, as a representative example of the utility of LEAP-RBP[49]. We also note a primary conclusion from our study of the effect of selective reduction in CDS ribosome occupancy status on RNA interactome composition, where global inhibition of translation initiation and ribosome runoff elicited RNA occupancy changes in only a small fraction of the RNA interactome. For those RBPs whose RNA interactions were sensitive to global translation initiation inhibition, differences in RNA-bound protein abundances were relatively modest, suggesting that for the supermajority of the RNA interactome, regulatory RBP–RNA interactions are biased to interactions at the 5' and 3' UTRs[39,40,57]. The successful application of this approach to identify and validate the dynamic responses of bona fide RBPs involved in translation initiation and uncover additional RNA interactors with previously unknown roles in RNA regulation provides strong experimental evidence of its utility for biological discovery.

The results presented herein suggest that the number of RNA-binding proteins currently thought to comprise the RNA interactome (~4925 human RBPs)[54] and/or those with GO-RBP annotations (~1693) is an overestimation. This perspective is consistent with findings in a recent CLIP-based RNA-protein cross-linking frequency analysis, where a crosslink frequency threshold that distinguishes bona fide and low-significant protein–RNA interactions was reported[11]. Importantly, LEAP-RBP combined with quantitative proteomic and SRA analysis provides direct experimental evidence of RNA binding and orthogonal validation of RBP activity[15,41,58]. The principle behind SRA has been previously reported and performed in combination with silica-based RBP purification methods[21,25]. Yet, biases in RBP–RNA adduct recovery or low sensitivity ($|S|/$μg RNA) and/or low $S/N$ can confound detection of many bona fide RBPs by SRA analysis alone (e.g., pAbPC1 and XRN1) (T; Fig. 9b). To that end, high $\%TP_S$ and efficient, unbiased recovery of RNA-bound protein is critical to the accurate identification of RNA interactomes and their state change dynamics, and is a goal largely met by LEAP-RBP (Supplementary Note 3). The inability to validate metabolic enzymes such as GAPDH which are frequently identified as candidate RBPs by other RBP profiling methods such as RIC and eRIC may indicate

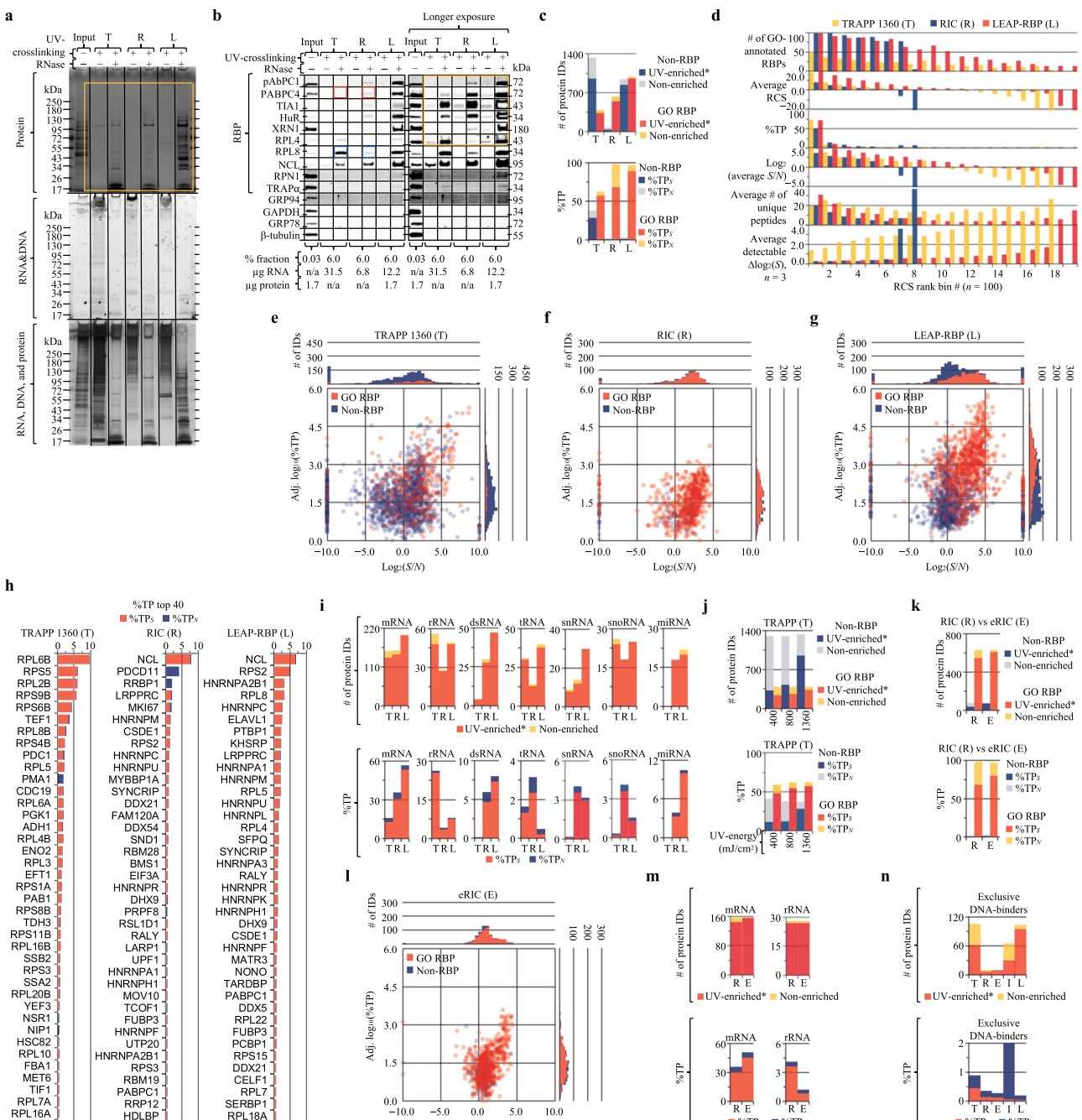

**Fig. 9 | LEAP-RBP provides a high-stringency, comprehensive portrait of the RNA-bound proteome.** L: LEAP-RBP; TRAPP; R: RIC; E: eRIC; SILAC: SILAC LC–MS/MS analysis; SRA: SRA analysis; E*: significantly UV-enriched*; NE: not E*. **a, b** UV-dependent enrichment and $S/N$ were evaluated by SRA with SYBR Safe (RNA&DNA), Coomassie Blue (protein), and Silver Stain (RNA, DNA, and protein) staining (**a**) or immunoblot (**b**). Sample compositions and/or normalization values included in figure panel **b**. **c** Stacked bar charts showing the number of RBPs and non-RBPs identified as E* or NE by each method, or the estimated %TP$_S$ and %TP$_N$ contributions of RBPs and non-RBPs in each RNP fraction. **d** Evaluation of method performance by RCS rank analysis. Proteins are ranked by their RBP-confidence scores and binned ($n = 100$ per bin). The number (#) of GO-annotated RBPs, RBP-confidence scores (RCS), %TP contributions, log$_2$($S/N$) ratios, # of unique peptides (i.e., coverage), and detectable $\Delta$log$_2$($S$) are compared. For methods with high performance and/or %TP$_S$, RCS rank predicts log$_2$($S/N$), %TP, coverage, and detectable $\Delta$log$_2$($S$). **e–g** Method specificity for RNA-bound RBPs was evaluated by

comparing observed abundances of RBPs and non-RBPs as a function of their log$_2$($S/N$) ratios (SILAC and non-SILAC). **h** Comparison of the 40 most abundant (%TP) proteins identified by each method with their estimated %TP$_S$ and %TP$_N$ contributions in each RNP fraction. **i** Stacked bar charts showing the number of proteins for different classes of RNA binders and their estimated %TP$_S$ and %TP$_N$ contributions in each RNP fraction. **j** Comparison of TRAPP experiments performed at differing UV cross-linking energies as described in (**c**). **k** Comparison of RIC (R) and eRIC (E) as described in **c** shows eRIC achieves a higher %TP$_S$ than RIC. **l** Evaluation of eRIC specificity as done in (**e–g**). **m** Comparison of mRNA and rRNA binders identified by RIC (R) and eRIC (E) as described in (**i**). **n** Comparison of exclusive DNA binders identified by TRAPP (T), RIC (R), eRIC (E), INP (I), and LEAP-RBP (L) as described in (**i**). Experiments were performed once (**a, b**); $n = 1$ (T, R); $n = 3$ biologically independent samples (L), pooled (equivalent % fraction) for SRA. Source data are provided as a Source Data file.

a limitation of our methodology for validation of low-frequency RNA interactors (Supplementary Note 7d). In these scenarios, validation of RNA-binding function in situ using complementary and/or orthogonal methods may be preferable. The high specificity and selectivity of the LEAP-RBP method for RNA-bound protein allows efficient capture of broad-spectrum RNA interactors from biological samples. Potential applications beyond those demonstrated here, including PAR and chemical cross-linking approaches, are reasonable to consider using the provided strategies (Supplementary Figs. 9–42, Supplementary Notes 1–8, and "Methods").

## Methods

### Methodical and analytical framework

A description of sample types, terminologies, quantitative metrics, and analytical approaches are provided in the Supplementary Methods. Analytical approaches: evaluating UV-dependent enrichment and $S/N$ by SDS-PAGE RNase-sensitivity Assay (SRA); estimating RBP-specific UV cross-linking efficiencies and $S/N$ ratios by SDS-PAGE and immunoblot; evaluating total protein and total RNA-bound protein abundance by SDS-PAGE; MS data analysis; RCS rank analysis.

### Criteria for assignment of RNA-binding activity

Protein displaying CL/nCL ratios >0 in LEAP-RBP fractions by SILAC LC–MS/MS were considered high or low-confidence RBPs based on their observed enrichment efficiency ($S/N$) and abundance (%TP). However, only those displaying discernible RNase sensitivity by SRA and immunoblot were considered bona fide RNA-binding proteins. Proteins which remained RNase-insensitive by SRA or undetectable were not considered bona fide RBPs regardless of GO-annotation (e.g., GRP94, a GO-annotated RBP). However, because the inability to detect a protein by SRA and immunoblot could be due to their low RNA-bound protein abundance, negative data were not considered formal confirmation of an absence of RNA-binding activity. To this point, validation of RNA-binding activity with LEAP-RBP and SRA requires that RNA-protein interactors are susceptible to UV cross-linking.

### LEAP-RBP optimization and quality control

The ability of LEAP-RBP to rapidly (< 5′) recover total RNA-bound protein from AGP suspensions with near 100% recovery is supported by a lack of quantifiable RNA and RNase-sensitive bands in the unprecipitated fraction by SRA and Coomassie Blue (protein) staining (Supplementary Fig. 3a). Total RNA-bound protein recovery was further validated by performing repeated LEAP steps without a significant, discernible decrease in protein-bound RNA yield (Supplementary Fig. 3b and Supplementary Note 1b). The importance of the liquid-liquid interphase during the LEAP step was evidenced by a significant decrease in RNA recovery (30–50%) when the solvents were quickly mixed (Supplementary Fig. 3c). RNA-dependence of LEAP-RBP was validated by performing LEAP-RBP on RNase-treated clRNP fractions resulting in a loss of detectable protein by SDS-PAGE and Coomassie Blue (protein) staining. In addition, performing LEAP-RBP on proteinase K-treated clRNP fractions resulted in recovery of RNA, but not protein (determined by Coomassie Blue staining), thereby demonstrating RNA-centricity (Supplementary Fig. 3d). Efficiency of DNA depletion and signal recovery during MS sample prep steps were validated by qPCR and SRA analysis (Supplementary Fig. 5b).

An RNA-seq analysis of small RNA composition was performed to determine if small RNA species are recovered by LEAP-RBP from final AGPC interphase suspensions of UV-crosslinked cells. RNA samples were found to be of high integrity (RIN > 9) and contained diverse sRNA species displaying broad genome distributions (Supplementary Fig. 4a–d). As noted by others, small RNA species were expected to be depleted following repeated AGPC extraction relative to other larger RNA species due to lower UV cross-linking efficiencies and depletion of free RNA[3,4]. Therefore, assessing the abundance of different RNA biotypes in clRNP fractions relative to their abundance in total RNA samples was considered uninformative. Nonetheless, SDS-PAGE of LEAP-RBP fractions isolated from AGP input suspensions demonstrate recovery of 60–100 bp RNA species visible as RNase-sensitive bands by SYBR Safe (RNA&DNA) staining migrating between 17 and 30 kD (nCL, w/o repeated AGPC extractions; Fig. 3a). For additional supporting information on the LEAP-RBP method, see Supplementary Notes 1–3.

### Cell line and culture conditions

HeLa, 293T, and Huh7 cells were maintained in Dulbecco's Modified Eagle's Medium (D6428, Sigma) supplemented with 10% FBS (35-010-CV, Corning) at 37 °C, 5% $CO_2$. 832/13 cells were maintained in RPMI1640 (11875-093, Invitrogen) supplemented with 2 mM L-glutamine (25030-081, Invitrogen), 1 mM Na-pyruvate (11360-070, Invitrogen), 10 mM HEPES (15630-080, Invitrogen), 0.05 mM 2-mercaptoethanol (M722, Sigma), and 10% FBS at 37 °C, 5% $CO_2$. SILAC-labeling was done using the Pierce SILAC-protein quantitation kit (1863108, Thermo), supplemented with 2 mM L-glutamine (02-0131-0200, VWR), and 10 μg/mL L-proline (88211, Thermo). Cells were passaged at least five times in their respective SILAC-labeled media (>10 doublings). For the comparative LEAP-RBP experiment, HeLa cells were maintained as described above and treated with DMSO (negative control) or 2 μg/mL Harringtonine (15361, Cayman Chemical Company) for 30 min at 37 °C, 5% $CO_2$; Harringtonine (HT) was prepared as a 1000× stock in DMSO.

### Sucrose density gradient polysome profiling

HeLa cells were cultured in 150 mm dishes until 80–90% confluent and treated with DMSO or harringtonine as described above, were washed twice with ice-cold 1× PBS and harvested on ice with 3 mL fresh ice-cold DDM lysis buffer (200 mM KOAc, 25 mM K-HEPES pH 7.2, 15 mM Mg(OAc)$_2$, 1 mM DTT, 50 μg/mL CHX, 1× protease inhibitor cocktail (11836153001, Roche), 40 U/mL RNase OUT(10777019, Thermo), and 2% dodecylmaltoside (DDM) (w/v)). DDM Lysates were centrifuged at 5000 × $g$ for 5 min at 4 °C and 1 mL of the clarified supernatants were resolved on a 10 mL sucrose gradients (15–40% w/v) containing DDM lysis buffer components noted above via centrifugation at 35,000 × $g$ for 3 h at 4 °C. Gradients were fractionated on a Teledyne Isco Lincoln (NE) gradient fractionator with continuous $A_{254}$ sampling[59].

### UV cross-linking and cell harvesting

Cells were cultured in 100- or 150-mm dishes until 60–90% confluent, washed twice with ice-cold 1× PBS, and UV-crosslinked on ice with 100–800 mJ/cm² at 254 nm. Cells were lysed on plate, scraped, and transferred to a 2.0 mL microcentrifuge tube using two 400 μL aliquots of guanidinium thiocyanate (w/o phenol) buffer[22,23]. Guanidinium thiocyanate (GT) buffer (4 M GT, 25 mM sodium citrate pH 7.0, 0.5% N-lauryl sarcosine, 5 mM EDTA pH 8.0, and 0.1 M 2-mercaptoethanol) was prepared with the following stock solutions prepared in DEPC-treated DI water: 5 M guanidinium thiocyanate (00522, Chemimpex), 750 mM sodium citrate pH 7.0 (BDH-9288, VWR; C-0759, Sigma), 10% N-lauryl sarcosine (L9150, Sigma), 0.5 M EDTA pH 8.0 (0105, VWR). Stock solutions were filtered (0.2 μm) to remove insoluble particulates that accumulate at the AGPC interphase: GT was filtered twice using Whatman paper (1001-150, Whatman) or by standing incubation overnight and transferring of the clarified portion; sodium citrate and EDTA stock solutions were filtered using 0.2-μm syringe filters (28145-477, VWR).

### Acidic guanidinium thiocyanate–phenol-chloroform extraction

400 μL of acidic phenol (0981, VWR) were added to 800 μL GT cell extracts. Alternatively, cells were lysed in 1.2 mL Trizol reagent (15596026, Invitrogen) and transferred to a 2 ml microcentrifuge tube. Cell lysates were prepared by passage through a 19 ga 1–1/2" needle fifteen times (305187, BD). For AGPC extraction, 240 μL chloroform (CX-1060-1, Millipore) or ~3/5th vol of phenol were added to samples

and vigorously vortexed for 10 s. Samples were centrifuged at 10,000 × g for 10 min at 4 °C with slow brake setting and ~80% (v/v) of the aqueous and organic phases were removed. For repeated AGPC extraction, 800 μL of fresh acidic guanidinium thiocyanate–phenol (2:1) buffer and 160 μL chloroform were added to the AGPC interphase and the process was repeated. The final AGPC interphase was resuspended in 1.0–1.5 mL fresh acid guanidinium thiocyanate–phenol (2:1) buffer. If AGP suspensions appeared cloudy, an additional AGPC extraction was performed. Additional protocol information is included in the Supplementary Methods.

## Precipitation of RNA from aqueous phase samples

Sodium chloride (5 M) was added to aqueous phase samples to a final concentration of 0.6 M and mixed by brief vortexing. One part isopropanol was added to a final concentration of 50% and samples were mixed by brief vortexing. Samples were incubated on a rotator for 15 min at 4 °C and centrifuged at 18,000 × g for 15 min at 4 °C with slow brake setting. Following removal of the supernatant, pellets were washed three times with ice-cold 75% ethanol (twice the volume of precipitation mixture), incubated for 5 min on ice with occasional agitation, and centrifuged at 18,000 × g for 5 min at 4 °C with slow brake setting. Pellets were air-dried and resuspended at the desired volume with DEPC-treated water or TE buffer. For long-term storage, precipitates were stored in 75% ethanol at −80 °C. Final working sample concentrations ranged from 0.2–2.0 μg of RNA/μL.

## Methanol precipitation (95% v/v)

Samples were mixed with 19 parts room temperature (RT) 100% methanol, incubated on a rotator for 1 h at RT, and centrifuged at 20,000 × g for 10 min at 20 °C with slow brake setting. Following removal of the supernatant fraction, precipitates were washed twice with 1.0 mL RT 95% methanol (for up to 100 μg protein). For each wash, samples were vortexed for 5 sec, incubated on a rotator for 10 min at RT, and centrifuged at 20,000 × g for 10 min at 20 °C with slow brake setting. Three 400 μL aliquots of RT 95% methanol were used to recover precipitates adhering to the sides of the tubes and combined in a 1.5 mL microcentrifuge tube. The tubes were then stored vertically at 4 °C overnight or at RT for 30 min to allow precipitates to settle at the bottom of the tube. Samples were centrifuged at 20,000 × g for 10 min at 20 °C with slow brake setting and supernatants were removed. Pellets were air-dried and resuspended at the desired concentration with 1% lithium dodecyl sulfate (LiDS) (J32816, Thermo) in TE. For long-term storage, samples were stored as precipitates in 95% methanol or as 1% LiDS TE suspensions at −80 °C. Working concentrations of methanol precipitated samples in 1% LiDS TE ranged from 0.1–5.0 μg protein/μL, 0.1–8.0 μg of RNA/μL, or 0.1–2.0 μg of protein-bound RNA/μL.

## Isolation of RNP fractions by INP

Final AGPC interphase suspensions were split between 2 mL microcentrifuge tubes (160 μL each). AGP suspensions were either stored at −80 °C or used immediately for precipitation. For precipitation, the following reagents were added to each AGP suspension in order while mixing by brief vortexing (5 sec) after each addition: 3 μL of GlycoBlue (AM9515, Invitrogen), 640 μL of 1% LiDS TE, 96 μL 5.0 M NaCl, and 899 μL isopropanol. Samples were vortexed for 5 sec and incubated on a rotator for 15 min at 4 °C. Samples were centrifuged at 14,000 × g for 15 min at 4 °C with slow brake setting. Following removal of the supernatant fraction, samples were washed three times with 1 mL ice-cold 75% ethanol, incubated for 5 min on ice, and centrifuged at 14,000 × g for 5 min at 4 °C with slow brake setting. Samples were then washed twice with 1 mL RT 95% methanol, incubated on a rotator for 10 min at RT and centrifuged at 20,000 × g for 10 min at 20 °C with slow brake setting. Supernatants were removed. Precipitates were air-dried and resuspended at the desired concentration in 1% LiDS in TE.

For long-term storage, precipitates were stored in 95% methanol or as 1% LiDS TE suspensions at −80 °C. Working concentrations of INP-precipitated RNPs ranged from 1.0 to 3.0 μg of protein-bound RNA/μL.

## Isolation of RNP fractions by LEAP-RBP

AGP input suspensions or final AGPC interphase suspensions were aliquoted (200 μL) across 1.5 mL microcentrifuge tube and stored at −80 °C or used immediately for precipitation. Chloroform was added to a final concentration of ~7% v/v and the sample was mixed by vortex to form an emulsion (after step A; Fig. 2c). Four parts of a precipitation solution containing 3.75 M LiCl (10515, VWR) and 50% isopropanol (v/v) were layered onto the AGPC mixtures, and the tubes were closed. Samples were slowly inverted to 90 degrees and/or until the AGPC mixture was displaced from the bottom of the tube, and then returned to an upright position followed by incubation for 1 min. This process was repeated at least four times, switching the direction of inversion, increasing the angle, and increasing the speed during reversion. Samples were then homogenized by vigorous vortexing, centrifuged at 14,000 × g for 5 min at 20 °C, and supernatants were removed. RNP pellets were rinsed twice with 1 mL RT 95% methanol by inverting the tube multiple times and removing the supernatant. RNP pellets were then washed with 1 mL RT 95% methanol by incubating for 5 min at RT with occasional inversion. Following removal of the final 95% methanol wash, pellets were air-dried and resuspended at the desired concentration with 1% LiDS TE. Additional protocol information is included in the Supplementary Methods. Working concentrations of LEAP-RBP isolated RNP fractions ranged from 0.1 to 4.0 μg of protein-bound RNA/μL.

## LEAP-RBP DNA depletion step

DNA digestion was performed using the Turbo DNase kit (Thermo, AM2238). RNP pellets containing <55 μg RNA&DNA were fully resuspended in 15 μL of TE buffer and 5 μL of a master mix containing 10× Turbo DNase buffer, TE buffer, and Turbo DNase were added to a final concentration of 1× Turbo DNase buffer and 1 μL of Turbo DNase/10 μg DNA. Samples were incubated at 37 °C for 15 min and nine parts (180 μL) fresh acid guanidinium thiocyanate–phenol (2:1) buffer were added. Samples were precipitated according to the LEAP-RBP protocol using 14 μL of chloroform and resuspended in 1% LiDS TE at the desired concentration. Additional protocol information is included as part of the Supplementary Methods.

## RNA and protein quantitation

Samples containing more than 1.5 μg RNA/μL were diluted 1:5 in their respective buffers for RNA quantitation by UV-spectrophotometry (Thermo Scientific, Nanodrop ND-1000). For samples where DNA contamination is expected to impact RNA quantitation by more than 10%, "RNA&DNA" was used in place of "RNA" for figure panels. Protein concentrations were determined by BCA protein assay (23225, Thermo) using a microplate 96-well format and BSA as a protein standard. Typically, 1% LiDS TE sample suspensions were clarified prior to protein quantitation: sample suspensions were incubated at 55 °C for 20 s, mixed by brief vortex, centrifuged at 3000 × g for 20 s at 20 °C, and clarified supernatants (~90% v/v) were transferred to a new tube. Two 2 μL aliquots of the clarified sample suspensions, typically containing between 0.1 and 1.0 μg protein were added to separate wells and mixed with 200 μL working reagent (Pierce BCA kit, 50:1 A:B) for BCA quantitation.

## RNase digestion for SDS-PAGE RNase-sensitivity Assay (SRA)

RNase digestions were performed in separate 0.2 mL thermocycler tubes (10–12 μL reactions) using a maximum of 5 μL of 1% LiDS TE sample suspensions containing <4.0 μg RNA/μL. RNase Cocktail (AM2286, Invitrogen), 10× RNase digest buffer (100 mM Tris-HCl pH 7.5, 1 M NaCl, and 10 mM EDTA), and 25× protease inhibitors

(11836153001, Roche) were added at the same time to a final concentration of 2 μL RNase Cocktail/15 μg RNA (Supplementary Fig. 12a), 1× RNase digestion buffer, and 1× protease inhibitors. A minimum of 0.2 μL RNase Cocktail were added regardless of RNA concentration. Samples were mixed by brief vortexing followed by a brief spin in a mini centrifuge (Supplementary Note 2a). Untreated control samples were prepared without RNase Cocktail, and both were incubated for 2 h at 37 °C in a thermocycler with a heated lid (98 °C) unless indicated otherwise in the provided Source Data (e.g., Supplementary Fig. 6b, c). Input samples suspended in 1% LiDS TE were set up as untreated control reactions for SDS-PAGE and were not incubated at 37 °C unless indicated otherwise in the provided Source Data (e.g., Fig. 7i, j and Supplementary Fig. 6b).

### SDS-PAGE, SYBR safe, Coomassie blue, silver stain staining

Sample loading buffer was prepared as a 5× stock (10% SDS, 50% glycerol, 312.5 mM Tris-HCl pH 6.8, and 0.1% (m/v) bromophenol blue (B8026, Sigma)) and diluted 3:1 with β-mercaptoethanol (v/v) for a working stock (LB WS). LB WS was added to samples to a final detergent concentration of 2% and denatured by incubating for 15 min at 65 °C. Samples were separated on a 0.75 mm, 15-well, 4–12% gradient polyacrylamide gels (6, 8, 10, 12% (1:1:1:1) resolver, 4% stacker) at constant voltage (80 V) for 1.5 h at RT (Supplementary Note 2c). SYBR Safe (S33102, Invitrogen), Coomassie Blue (1610406, Biorad), and Silver Stain (PROTSIL2, Sigma) staining of polyacrylamide gels was performed on an orbital shaker. Imaging was performed using an Amersham Imager 600 (see corresponding Source Data). Additional protocol information is included as part of the Supplementary Methods.

### Immunoblot

Following separation by SDS-PAGE, samples were transferred to nitrocellulose membranes using Bjerrum and Schafer-Nielsen transfer buffer (48 mM Tris and 39 mM glycine supplemented with 10% methanol and 0.03% SDS) and a Trans-Blot SD semi-dry electrophoretic transfer cell (170-3940, Bio-Rad). Alternatively, samples were wet transferred to nitrocellulose membranes using wet-transfer buffer (25 mM Tris, 96 mM glycine, 0.05% SDS, and 20% methanol) and a Bio-Rad Mini-Protean II system. Blocking and blotting conditions were performed as follows: anti-pAbPC1 antibody (ABclonal, A14872, lot 1160820101, rabbit polyclonal, diluted 1:2000 in 1× TBST + 5.0% milk, blocked with 1× TBST + 5.0% milk for 1 h at RT), anti-PABPC4 antibody (ABclonal, A5948, lot 1150980101, rabbit polyclonal, diluted 1:2000 in 1× TBST + 5.0% milk, blocked with 1× TBST + 5.0% milk for 1 h at RT), anti-TIA1 antibody (ABclonal, A6237, lot 1150860101, rabbit polyclonal, diluted 1:2000 in 1× TBST + 5.0% milk, blocked with 1× TBST + 5.0% milk for 1 h at RT), anti-HuR antibody (Santa Cruz Biotechnology, Sc-5261, clone 3A2, lot n/a, mouse monoclonal, diluted 1:2000 in 1× TBST + 5.0% milk, blocked with 1× TBST + 0.3% casein for 15 min at RT), anti-XRN1 antibody (Bethyl Laboratories, A300-443A, lot A300-443A-3, rabbit polyclonal, diluted 1:2000 in 1× TBST + 5.0% milk, blocked with 1× TBST + 0.3% casein for 15 min at RT), anti-RPL4 antibody (Santa Cruz Biotechnology, Sc-100838, clone RQ-7, lot n/a, mouse monoclonal, diluted 1:500 in 1× TBST + 5.0% milk, blocked with 1× TBST + 0.1% casein for 15 min at RT), anti-RPL8 antibody (ABclonal, A10042, lot 0051990201, rabbit polyclonal, diluted 1:2000 in 1× TBST + 5.0% milk, blocked with 1× TBST + 5.0% milk for 1 h at RT), anti-LRRC59 antibody (Bethyl Laboratories, A305-076A, lot A305-076A-1, rabbit polyclonal, diluted 1:1000 in 1× TBST + 0.2% milk, blocked with 1× TBST + 0.3% casein for 15 min at RT), anti-NCL antibody (ABclonal, A5904, lot 0015360101, rabbit polyclonal, diluted 1:2000 in 1× TBST + 5.0% milk, blocked with 1× TBST + 0.2% casein for 15 min at RT), anti-RPN1 antibody (Nicchitta, αP3, lot bleed 1990/08/04, rabbit polyclonal, diluted 1:5000 in 1× TBST + 5.0% milk, blocked with 1× TBST + 5.0% milk for 15 min at RT), anti-TRAPα antibody (Nicchitta,

TRAPα, lot bleed 7, rabbit polyclonal, diluted 1:5000 in 1× TBST + 5.0% milk, blocked with 1× TBST + 5.0% milk for 15 min at RT), anti-GRP94 antibody (Nicchitta, DU120, lot bleed 1998/11/11, rabbit polyclonal, diluted 1:5000 in 1× TBST + 5.0% milk, blocked with 1× TBST + 5.0% milk for 15 min at RT), anti-GAPDH antibody (DSHB, DSHB-hGAPDH-2G7, clone 2G7, lot n/a, mouse monoclonal, diluted 1:250 in 1× TBST + 5.0% milk, blocked with 1× TBST + 0.1% casein for 15 min at RT), anti-GRP78 antibody (Santa Cruz Biotechnology, Sc-376768, clone A-10, lot n/a, mouse monoclonal, diluted 1:100 in 1× TBST + 5.0% milk, blocked with 1× TBST + 0.1% casein for 15 min at RT), anti-β-tubulin antibody (DSHB, E7-s, clone E7, lot n/a, mouse monoclonal, diluted 1:250 in 1× TBST + 5.0% milk, blocked with 1× TBST + 0.1% casein for 15 min at RT), anti-RPS3 antibody (ABclonal, A4872, clone ARC0302, lot 4000000302, rabbit monoclonal, diluted 1:2000 in 1× TBST + 5.0% milk, blocked with 1× TBST + 5.0% milk for 1 h at RT), anti-SND1 antibody (ABclonal, A5874, lot 0029220201, rabbit polyclonal, diluted 1:2000 in 1× TBST + 5.0% milk, blocked with 1× TBST + 5.0% milk for 1 h at RT), anti-UPF1/RENT1 antibody (ABclonal, A5071, clone ARC1268, lot 4000001268, rabbit monoclonal, diluted 1:2000 in 1× TBST + 5.0% milk, blocked with 1× TBST + 5.0% milk for 1 h at RT), anti-HDLBP antibody (ABclonal, A20896, clone ARC2855, lot 4000002855, rabbit monoclonal, diluted 1:2000 in 1× TBST + 5.0% milk, blocked with 1× TBST + 5.0% milk for 1 h at RT), anti-ABCF3 antibody (ABclonal, A15168, lot 0127370101, rabbit polyclonal, diluted 1:2000 in 1× TBST + 5.0% milk, blocked with 1× TBST + 5.0% milk for 1 h at RT), anti-GEMIN5 antibody (ABclonal, A17125, lot 0111800101, rabbit polyclonal, diluted 1:2000 in 1× TBST + 5.0% milk, blocked with 1× TBST + 5.0% milk for 1 h at RT), anti-eEF2 antibody (ABclonal, A9721, clone ARC1717, lot 4000001717, rabbit monoclonal, diluted 1:2000 in 1× TBST + 5.0% milk, blocked with 1× TBST + 5.0% milk for 1 h at RT), anti-CELF1 antibody (ABclonal, A5958, lot 0202600301, rabbit polyclonal, diluted 1:2000 in 1× TBST + 5.0% milk, blocked with 1× TBST + 5.0% milk for 1 h at RT), anti-Fibrillarin/U3 RNP antibody (ABclonal, A1136, lot 0002110201, rabbit polyclonal, diluted 1:2000 in 1× TBST + 5.0% milk, blocked with 1× TBST + 5.0% milk for 1 h at RT). Signal detection was performed using WesternBright ECL HRP substrate (K-12045, Advansta) and an Amersham Imager 600 (see corresponding Source Data). Additional protocol information is included as part of the Supplementary Methods.

### Proteinase K digestion

For proteinase K digestion, samples were diluted 1:2 with 2× proteinase K buffer (100 mM Tris-HCl pH 7.5, 20 mM EDTA pH 8.0, 300 mM NaCl, 2% SDS), mixed with 2 μL proteinase K stock (20 mg/mL proteinase K (BIO-37037, Bioline), 20 mM Tris-HCl pH 7.5, 1 mM CaCl$_2$, 50% glycerol v/v) per 10 μg of protein, and incubated at 55 °C for 15 min. For isolation of RNA and/or DNA samples were mixed with four parts neutral guanidinium thiocyanate–phenol (2:1) buffer (J75829, Affymetrix) and 1 part chloroform, vigorously vortexed for 10 s, and centrifuged at 10,000 × $g$ for 10 min at 4 °C with slow brake setting. Aqueous phase samples were precipitated as outlined above (Precipitation of RNA from aqueous phase samples).

### TBE gel analysis of RNA samples

RNA samples suspended in DEPC-treated water were mixed with 6× gel loading buffer (R0611, Thermo), incubated at 65 °C for 2 min, and chilled for 2 min on ice before being loaded on a 1.0% or 1.5% agarose TBE gel containing 0.5–1× SYBR Safe stain. Samples were separated under constant voltage at 140 V or 140 V for 20–40 min and visualized using an Amersham Imager 600 (see corresponding Source Data for specific experimental conditions).

### qPCR analysis

qPCR was performed using the Luna Universal qPCR Master Mix (NEB, M3003) on a Bio-RAD Cfx96 real-time PCR system using a 96-well

format and 20 μL reactions. DNA contamination was quantified using primers targeting the coding region of GRP78:
F-primer: 5′-CTTGGTATTGAAACTGTGGGAGGT-3′
R-primer: 5′-AGATCTGAGACTTCTTGGTAGGCA-3′

## Sample preparation for MS proteomic analysis

Digestion and depletion of RNA and/or DNA from input samples and RNP fractions were necessary prior to MS-based proteomic analysis (Supplementary Figs. 5b and 9a–c). For SILAC LC–MS/MS experiments, RNP suspensions were normalized to 3 μg/μL protein-bound RNA in 1% LiDS TE, and 33 μL were used per 100 μL reaction containing 13.3 μL RNase Cocktail, 1× RNase digest buffer (10 mM Tris-HCl pH 7.5, 100 mM NaCl, and 1 mM EDTA pH 8.0), and 1× protease inhibitors (11836153001, Roche). For the comparative LEAP-RBP experiment, 10 μL of input samples containing 2.0 μg protein/μL were used per 25 μL reaction containing 0.6 μL RNase Cocktail, 1× RNase digest buffer, and 1× protease inhibitors; 20 μL of clRNP fractions containing 0.2 μg RNA-bound protein/μL were used per 50 μL reaction containing 3.1 μL RNase Cocktail, 1× RNase digest buffer, and 1× protease inhibitors. RNase digests were incubated for 2 h at 37 °C and then precipitated with 95% methanol v/v as described above. Each input sample was suspended in TE buffer and DNA was digested using 2.5 μL Turbo DNase (Thermo, AM2238) and a final concentration of 1× Turbo DNase buffer in a 50 μL reaction (15 min, 37 °C) and precipitated with 95% methanol v/v as described above. Pellets were air-dried and submitted for proteomics analysis. For long-term storage, we recommend storing precipitates in 95% methanol at −80 °C.

## SILAC LC–MS/MS analysis of LEAP-RBP and INP fractions

Prior to LC–MS/MS analysis, samples were supplemented with 50 μL 8.0 M urea in 50 mM ammonium bicarbonate and subjected to two rounds of probe sonication. Next, samples were spiked with either a total of 120 or 240 fmol of bovine casein, supplemented with 15 μL 20% SDS, reduced with 10 mM dithiothreitol for 30 min at 45 °C and alkylated with 20 mM iodoacetamide for 45 min at RT. Then, samples were supplemented with a final concentration of 1.2% phosphoric acid and 543 μL of S-Trap (Protifi) binding buffer (90% methanol/100.0 mM TEAB). Proteins were collected on the S-Trap, digested using 20 ng/μL sequencing grade trypsin (Promega) for 1 h at 47 °C, and eluted using 50 mM TEAB, followed by 0.2% FA, and lastly using 50% ACN/0.2% FA. All samples were then lyophilized to dryness and resuspended in 12 μL 1% TFA/2% acetonitrile containing 12.5 fmol/μL yeast alcohol dehydrogenase (ADH_YEAST).

Quantitative LC–MS/MS was performed on 1 μg of each sample, using a nanoAcquity UPLC system (Waters Corp) coupled to a Thermo Orbitrap Fusion Lumos high-resolution accurate mass tandem mass spectrometer (Thermo) equipped with a FAIMSPro device via a nanoelectrospray ionization source. Briefly, peptides were trapped on a Symmetry C18 20 mm × 180 μm trapping column (5 μL/min at 99.9/0.1 v/v water/acetonitrile), after which the analytical separation was performed using a 1.8 μm Acquity HSS T3 C18 75 μM × 250 mm column (Waters Corp.) with a 90-min linear gradient of 5 to 30% acetonitrile with 0.1% formic acid at a flow rate of 400 nanoliters/minute (nL/min) with a column temperature of 55 °C. Data collection on the Fusion Lumos mass spectrometer was performed for three difference compensation voltages (40 V, 60 V, 80 V). Within each CV, a data-dependent acquisition (DDA) mode of acquisition with a $r = 120,000$ ($m/z$ 200) full MS scan from $m/z$ 375 to 1500 with a target AGC value of 4e5 ions was performed. MS/MS scans were acquired in the ion trap in rapid mode from $m/z$ 100 with a target AGC value of 2e4 and max fill time of 100 ms. The total cycle time for each CV was 1 s, with total cycle times of 3 s between like full MS scans. A 45 s dynamic exclusion was employed to increase depth of coverage. The total analysis cycle time for each fraction injection was ~2 h.

Data were imported into Proteome Discoverer 2.5 (Thermo Scientific Inc.) and all LC–MS/MS runs were aligned based on the accurate mass retention time of detected ions ("features") which contained MS/MS spectra using Minora Feature Detector algorithm in Proteome Discoverer. Relative peptide abundance was calculated based on area-under-the-curve (AUC) of the selected ion chromatograms of the aligned features across all runs. A filter was applied which required each peptide to be measured in at least 2 unique samples and in at least 50% of at least one of the unique biological groups. The MS/MS data was searched against the SwissProt *H. sapiens* database (downloaded 11/2019) and an equal number of reversed sequence "decoys" for false-discovery rate determination. Mascot Distiller and Mascot Server (v 2.5, Matrix Sciences) were utilized to produce fragment ion spectra and to perform database searches. Database search parameters included fixed modification on Cys (carbamidomethyl) and variable modifications on Meth (+16, oxidation) and Arg/Lys (+10/ +8 for heavy SILAC residues K + 8, R + 10). Peptide Validator and Protein FDR Validator nodes in Proteome Discoverer were used to annotate the data at a maximum 1% protein false-discovery rate based on $q$ value calculations. Note that peptide homology was addressed using razor rules in which a peptide matched to multiple different proteins was exclusively assigned to the protein that has more identified peptides. Protein homology was addressed by grouping proteins that had the same set of peptides to account for their identification. Following database searching and peptide scoring using Proteome discoverer validation, the data was annotated at a 1% protein false-discovery rate.

## SILAC LC–MS/MS data processing and analysis

Initial data processing for the identification of UV-enriched proteins and generation of sum peptide intensities were done separately for each method (INP vs LEAP-RBP). Peptide intensities of common contaminants and spike-ins (human keratins, BSA, porcine trypsin, yeast alcohol dehydrogenase) were manually curated from protein lists. The remaining peptide intensities were sorted by SILAC label and used to generate sum peptide intensities (SPI). Proteins not detected in all three UV-crosslinked samples were excluded from downstream sample normalization procedures and data analysis. Replicate samples were mean-normalized to total SPI and $SPI_{nCL}$ values equal to 0 were replaced with the average non-zero $SPI_{nCL}$ value of the same protein ID. Proteins only detected in UV-crosslinked samples were scored as UV-enriched*, ommitted from statistical analysis, and given the following pseudo-value: $-\log_{10}(P \text{ value}) = 10$, $\log_{2}(CL/nCL) = 10$. For the remaining proteins, $\log_{2}(CL/nCL)$, ratios were generated with $SPI_{CL}$ values and average $SPI_{nCL}$ values according to Eq. (2). UV-enriched* proteins were identified by testing against the null hypothesis that the average $\log_{2}(CL/nCL)$ ratio equals zero using a heteroscedastic upper-tailed $t$ test. Correction for multiple hypothesis testing was performed using the Benjamini–Hochberg approach and a false-discovery rate of 5%. A summary of statistical testing for INP and LEAP-RBP is provided in Supplementary Data 2 and 4, respectively.

## Data processing and analysis of referenced MS datasets

Maxquant output files (.txt) for XRNAX[3], OOPs[51], Ptex[21], and TRAPP[50] were downloaded from the ProteomeXchange using the identifiers PXD010520, PXD026716, PXD009571, and PXD011071 respectively. Protein identifiers, unique peptide counts, and sum peptide intensities were obtained from their respective proteingroup.txt file; proteins marked as potential contaminants were removed. MS datasets for RIC and eRIC, including protein identifiers, unique peptide counts, and sum peptide intensities, were obtained from[52]; Supplementary Data 1 ("full dataset" tab). Protein identifiers (Uniport IDs and gene names) starting with "Majority protein IDs" were used to generate primary Uniprot IDs for comparative analyses. A list of extracted protein IDs, gene names, and assigned primary Uniprot IDs are available in

Supplementary Data 6. For RIC and eRIC, a pseudo-third replicate was added by averaging non-zero SPI values of replicates 1 and 2 (Supplementary Data 14 and 15)[52]. Because XRNAX was performed without replicates, samples were first mean-normalized and the average non-zero SPI values of 12 different samples were used for MS data analyses (Supplementary Data 10)[3]; MCF7, HEK293, and HeLa; half-confluent and confluent; 15 min and 30 min partial digestion prior to silica purification ($3 \times 2 \times 2 = 12$ different samples). For the remaining MS datasets, proteins not detected in all UV-crosslinked samples were excluded from downstream sample normalization procedures and data analysis. Replicate samples were mean-normalized to total SPI and $SPI_{nCL}$ values equal to 0 were replaced with the average non-zero $SPI_{nCL}$ value of the same protein ID.

### LC−MS/MS analysis (comparative LEAP-RBP experiment)

Prior to LC−MS/MS analysis, samples were supplemented with 50 μL 8.0 M and subjected to two rounds of probe sonication. Next, samples were spiked with either a total of 120 or 240 fmol of bovine casein, supplemented with 7.9 μL 20% SDS, reduced with 10 mM dithiolthreitol for 30 min at 32 °C and alkylated with 20 mM iodoacetamide for 45 min at RT. Then, samples were supplemented with a final concentration of 1.2% phosphoric acid and 472 μL of S-Trap (Protifi) binding buffer (90% methanol/100.0 mM TEAB). Proteins were collected on the S-Trap, digested using 4 or 20 ng/μL (for clRNP fractions containing 4 μg protein or input samples containing 20 μg protein, respectively) sequencing grade trypsin (Promega) for 1 h at 47 °C, and eluted using 50 mM TEAB, followed by 0.2% FA, and lastly using 50% ACN/0.2% FA. All samples were then lyophilized to dryness and resuspended in 12 or 60 μL (for clRNP fractions or input samples, respectively) of 1% TFA/2% acetonitrile containing 12.5 fmol/μL yeast alcohol dehydrogenase (ADH_YEAST).

Quantitative LC−MS/MS was performed on 3 μL (1 μg) of each sample, using a nanoAcquity UPLC system (Waters Corp) coupled to a Thermo Orbitrap Fusion Lumos high-resolution accurate mass tandem mass spectrometer (Thermo) equipped with a FAIMSPro device via a nanoelectrospray ionization source. Briefly, peptides were trapped on a Symmetry C18 20 mm × 180 μm trapping column (5 μL/min at 99.9/0.1 v/v water/acetonitrile), after which the analytical separation was performed using a 1.8 μm Acquity HSS T3 C18 75 μM × 250 mm column (Waters Corp.) with a 90-min linear gradient of 5 to 30% acetonitrile with 0.1% formic acid at a flow rate of 400 nanoliters/minute (nL/min) with a column temperature of 55 °C. Data collection on the Fusion Lumos mass spectrometer was performed for three different compensation voltages (40 V, 60 V, 80 V). Within each CV, a data-dependent acquisition (DDA) mode of acquisition with a $r = 120,000$ ($m/z$ 200) full MS scan from $m/z$ 375 to 1500 with a target AGC value of 4e5 ions was performed. MS/MS scans were acquired in the Orbitrap at r = 50,000 ($m/z$ 200) from $m/z$ 100 with target AGC value of 1e5 and max fill time of 35 ms. The total cycle time for each CV was 1 s, with total cycle times of 3 sec between like full MS scans. A 45 s dynamic exclusion was employed to increase depth of coverage. The total analysis cycle time for each fraction injection was approximately 2 h.

Following 15 total UPLC-MS/MS analyses (excluding conditioning runs, but including three replicate SPQC samples), data were imported into Proteome Discoverer 3.0 (Thermo Scientific Inc.), and individual LC−MS data files were aligned based on the accurate mass retention time of detected precursor ions ("features") using Minora Feature Detector algorithm in Proteome Discoverer. Relative peptide abundance was measured based on peak intensities of the selected ion chromatograms of the aligned features across all runs. The MS/MS data was searched against the SwissProt *H. sapiens* database (downloaded 08/2022), a common contaminant/spiked protein database (bovine albumin, bovine casein, yeast ADH, etc.), and an equal number of reversed sequence "decoys" for false-discovery rate determination. Sequest was utilized to produce fragment ion spectra and to perform the database searches. Database search parameters included fixed modification on Cys (carbamidomethyl) and variable modification on Met (oxidation). Search tolerances were 2 ppm and 0.8 Da product ion with full trypsin enzyme rules. Peptide Validator and Protein FDR Validator nodes in Proteome Discoverer were used to annotate the data at a maximum 1% protein false-discovery rate based on $q$ value calculations. Note that peptide homology was addressed using razor rules in which a peptide matched to multiple different proteins was exclusively assigned to the protein that has more identified peptides. Protein homology was addressed by grouping proteins that had the same set of peptides to account for their identification. A master protein within a group was assigned based on % coverage.

### LC−MS/MS data processing (comparative LEAP-RBP experiment)

Initial data processing and generation of sum peptide intensities were done separately for each fraction and each sample group (input or clRNP and DMSO or HT). Peptide intensities of common contaminants and spike-ins (human keratins, BSA, porcine trypsin, yeast alcohol dehydrogenase) were manually curated from protein lists. Proteins not detected in all three replicates of both sample groups (DMSO and HT) for a given fraction (input or clRNP) and containing at least two unique peptide matches were excluded from downstream sample normalization and data analysis. Samples were mean-normalized to total SPI and $\log_2$ normalized SPI values were used to test for differences in protein recovery between samples groups (DMSO vs HT) for each fraction (input or clRNP) using independent two-tailed homoscedastic $t$ tests (Supplementary Data 8 and 9). Correction for multiple hypothesis testing was performed with the Benjamini–Hochberg approach and a false-discovery rate of 5% on total protein IDs (no $S/N$ limit) or only those which displayed $S/N$ ratios >3 in LEAP-RBP (clRNP) fractions by SILAC LC−MS/MS analysis.

### Gene-Ontology (GO) enrichment analysis

GO-enrichment analyses were performed for UV-enriched* proteins identified by INP and LEAP-RBP using PANTHER V17.0[60]. For additional analyses of GO-annotated groups, lists were exported and assigned primary Uniprot IDs as described in Supplementary Data 6. The resulting GO-annotated protein lists were used to sort protein IDs (e.g., RBP vs non-RBP) for downstream analyses and can be found in Supplementary Data 7.

### Sample preparation for sRNA-seq and data analysis

Two independent samples (HeLa) were UV-crosslinked with 0.4 J/cm$^2$ (254 nm). clRNPs were isolated from the final (6th) AGPC interphase suspension by LEAP-RBP and resuspended in TE buffer. Ca. 6 μg of protein-bound RNA was treated with Turbo DNase as outlined above (LEAP-RBP DNA depletion step) without performing the second LEAP step. Then, 20 μL of 2× proteinase K buffer and 3 μL proteinase K stock (20 mg/mL) were added, and samples were processed as described above (Proteinase K digestion).

### Library construction, quality control, and sRNA sequencing

For sRNA library construction, 3' and 5' adapters were ligated to 3' and 5' ends of small RNAs, respectively. First-strand cDNA was synthesized after hybridization with a reverse transcription primer and double-stranded cDNA libraries generated via PCR enrichment. After purification and size selection, libraries with insertions between 18−40 bp were selected. Library concentrations and QC was performed via Qubit and real-time PCR for quantitation and Bioanalyzer for size distribution analysis. Quantified libraries were pooled and sequenced on Illumina platforms in SE50 mode.

## Data analysis (sRNA-seq)

Raw data (raw reads) in fastq format were processed through custom (Novogene) perl and python scripts to remove read sequences containing poly-N, 5′ adapter contaminants, lacking 3′ adapter or the insert tag, containing polyA, T, G or C, and low quality reads. Small RNA read data were mapped to reference sequence using Bowtie version 0.12.9[61], without mismatch. Mapped small RNA tags were examined for known miRNA homologies using miRDeep2 version 0.0.5[62]. To remove tags originating from protein-coding genes, repeat sequences, rRNA, tRNA, snRNA, and snoRNA, small RNA tags were mapped with RepeatMasker version 4.0.3 and Rfam version 11.0[63,64]. Novel miRNA predictions were performed using miRDeep2 version 0.0.5 modified with miREvo version 1.1 and ViennaRNA version 2.1.1 through exploration of secondary structure, Dicer cleavage sites, and the minimum free energy of the small RNA tags unannotated in the former steps[62,65,66]. For alignment and annotations, some small RNA tags may map to more than one category. To ensure that small RNAs mapped to only one annotation, the following priority rules were used: known miRNA > rRNA > tRNA > snRNA > snoRNA > repeat > gene > NAT-siRNA > gene > novel miRNA > ta-siRNA. miRNA expression levels were estimated by TPM (transcript per million) through the following criteria[67]: Normalization formula: Normalized expression = mapped reads * 1,000,000.

## Statistical analysis

All statistical analyses were performed using JMP Pro 14.0, exported test results included as part of the provided Source Data.

## Reporting summary

Further information on research design is available in the Nature Portfolio Reporting Summary linked to this article.

## Data availability

Raw data and Protein Discoverer results files from LEAPR-RBP and INP SILAC, and non-SILAC LC−MS/MS experiments are available on the MassIVE repository, accession record MSV000088005. Small RNA sequencing data are available at NCBI GEO, series record GSE235647. Maxquant output files for XRNAX[3], OOPs[51], Ptex[21], and TRAPP[50] were downloaded from the ProteomeXchange using the following accession codes; XRNAX: PXD010520 (proteinGroups.txt file located in the txt_ihRBP.zip file); OOPs: PXD021169 (proteinGroups.txt file located in the txt.zip file); Ptex: PXD009571 (proteinGroups.txt file located in the txt_Human.zip file); TRAPP: PXD011071 (Maxquant_proteinGroups.txt files located in the TRAPP_cerevisiae_400.zip, TRAPP_cerevisiae_800.zip, and TRAPP_cerevisiae_1360.zip files). MS datasets for RIC and eRIC, including protein identifiers, unique peptide counts, and sum peptide intensities, were obtained from https://doi.org/10.1038/s41467-018-06557-8 (Supplementary Data 1, "full dataset" tab)[52]. Complete MS datasets and parameters discussed in this study for LEAP-RBP, INP, and each of the referenced studies are provided as individual Excel files: LEAP-RBP (Supplementary Data 3), INP (Supplementary Data 5), XRNAX (Supplementary Data 10), OOPs (Supplementary Data 11), Ptex 1.5 (Supplementary Data 12), TRAPP 1360 (Supplementary Data 13), RIC (Supplementary Data 14), eRIC (Supplementary Data 15), TRAPP 400 (Supplementary Data 16), TRAPP 800 (Supplementary Data 17), Ptex 0.015 (Supplementary Data 18), Ptex 0.15 (Supplementary Data 19), and OOPs LFQ (Supplementary Data 20). Data sources for all referenced MS datasets can be found in Supplementary Data 6. A summary of statistical tests used for the identification of UV-enriched* protein IDs in LEAP-RBP and INP fractions by SILAC LC−MS/MS is provided in Supplementary Data 2, 4, respectively. Summary of statistical tests used for the identification of proteins displaying a significant difference in total (input samples) and/or RNA-bound abundance (clRNP fractions) in response to

harringtonine treatment are provided in Supplementary Data 8, 9, respectively. The data supporting the findings of this study are available within the main Manuscript and Supplementary Information, or in the Source data provided with this paper. Specific P values are included within the Source Data file as well. Source data are provided with this paper.

## Code availability

Custom scripts used during the small RNA sequencing experiment to clean reads are propriety scripts of Novogene. The remaining software used for small RNA sequencing analysis is publicly available: Bowtie version 0.12.9 [https://sourceforge.net/projects/bowtie-bio/files/bowtie/0.12.9/]; RepeatMasker version 4.0.3 [https://www.repeatmasker.org/]; Rfam version 11.0 [http://xfam.org/]; miRDeep2 version 0.0.5 [https://github.com/rajewsky-lab/mirdeep2]; miREvo version 1.1 [https://github.com/akahanaton/miREvo]; ViennaRNA version 2.1.1 [https://www.tbi.univie.ac.at/RNA/#download].

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

## Acknowledgements

We thank members of the Nicchitta lab for scientific input and insightful
critiques, and to Dr. Qiang Chen for expert help with polyribosome
profiling studies. We thank the Duke University School of Medicine for
the use of the Proteomics and Metabolomics Core Facility, which pro-
vided mass spectrometry services critical to this study and thank Erik
Soderblom, PhD, Director, Proteomics and Metabolomics Core, and Matt
Foster, PhD, Proteomics and Metabolomics Core, for invaluable scien-
tific advice and support. Research reported in this publication was
supported in part by the Office of The Director, National Institutes of
Health of the National Institutes of Health under Award Number
S10OD024999. The content is solely the responsibility of the authors
and does not necessarily represent the official views of the National
Institutes of Health. Funding for this study was provided by a grant from
the NIH to CVN (GM139480).

## Author contributions

J.K. performed conceptualization, methodology, investigation, visuali-
zation, data curation, writing—original draft preparation. C.V.N. per-
formed supervision, writing—reviewing and editing, and funding
acquisition.

## Competing interests

The authors declare no competing interests.

## Additional information

**Supplementary information** The online version contains
supplementary material available at

Christopher V. Nicchitta.

**Peer review information** *Nature Communications* thanks Ana M. Matia-
González, and the other, anonymous, reviewers for their contribution to
the peer review of this work. A peer review file is available.

