## [Peer Review File · Nature Communications]

REVIEWER COMMENTS

Reviewer #1 (Remarks to the Author):

This manuscript reports a novel method to isolate RNA binding proteins. Given the importance of the posttranscriptional regulation for the cell fitness, this objective has been pursued by numerous research groups worldwide and several methodologies have been developed with this aim in the last decade. However, one of the major concerns of these recent methods was the specificity. Their workflow builds on the previously published organic phase separation methods, which enrich RBPs according to their physiochemical properties and are deemed to provide improved specificity and efficiency. In addition, these methods, as opposed to RIC, enable the identification of all proteins associated with any class of RNA instead just messenger RNAs, resulting in a more complete picture of the RBPome.

The main strength of this manuscript is the development of a new method to specifically isolate RBPs and a way (signal to noise ratio) to distinguish real binders to background. As the manuscript mentioned, UV-CL is not very efficient and one of the main criticisms for techniques to isolate RBPs has always been the 'noise', so I think that the authors have found a very elegant method to identify real RNA-binders. For the most part, this manuscript is clearly presented, concise, and the interpretations are not overstated. The method is extremely well described and every step of it is justified.

Even though I consider this manuscript original and doubtless will be of interest to the RNA community I have three major concerns:

1. The authors evaluate throughout the manuscript GO-annotated RBPs and non-RBPs however they do not describe how did they selected these proteins and would be interesting to know whether these proteins have any known feature like the RNA binding domains, the type of RNA that they bind, or the number of targets – this may help to the interpretation of the data.

In this regard, they considered within the GO-annotated RBPs two subgroups, depending on the presence of RNA binding domains. From their observations when applying LEAP-RBP they concluded that 'only LEAP-RBP clearly distinguishes RBPs with low S/N (e.g., RPN1, TRAPa) from non-RBPs like GRP94 (red box; Fig. 1f). These results suggest GRP94 (a GO-annotated RBP) is a false positive and indicates TRAPa (non-RBP) is a false negative'. Could this be due to the number of targets of the non-canonical RBP – GRP94? Or could be just that these other proteins are less sensitive to the UV-CL as they are performing other activities within the cell and their RNA binding activity is minimal? My point is, this method is clearly great for the canonical RBPs but is it too restrictive for proteins that bind just a few targets or targets not highly expressed?

2. When the authors make the comparison with other published methods (OOPs, RIC, TRAPP, etc...) I can clearly see the differences across the methods, but it is hard to believe that proteins like GAPDH show up as RNA-binder when applying some techniques and no other and the positive results have to be considered background or false positive. In this regard, it has been described that glycolytic enzymes do bind RNA (recently reviewed in Wegener M & Dietz KJ. The mutual interaction of glycolytic enzymes and RNA in post-transcriptional regulation. *RNA*. 2022 doi: 10.1261/rna.079210.122), and for some of them the specific targets have been identified (the most recent example is the enolase Huppertz I, et al. Riboregulation of Enolase 1 activity controls glycolysis and embryonic stem cell differentiation. *Mol Cell*. 2022 Jul 21;82(14):2666-2680.e11. doi: 10.1016/j.molcel.2022.05.019.). As I mentioned before, would be very helpful to monitor in these western blots RBPs with different features, RBPs that bind mRNAs, tRNAs, miRNAs, circRNAs, etc... to make sure that this method is not restrictive to proteins that bind a certain type of RNA or have highly expressed targets. Indeed, the RNase Cocktail (AM2286, Invitrogen) used by the authors, would not be helpful for those RBPs that bind circular RNAs (not sensitive to the RNases) or even those proteins that bind DNA-RNA hybrids (treatment with RNase H1 would be helpful).

3. One last comment is about the overlaps showed in the supplementary data (Suppl. Fig 14). One of the main criticisms of RIC that I mentioned in the beginning of the revision has always been the specificity, and the application of the LEAP-RBP is deemed to provide improved specificity and efficiency, how is it possible then that the overlap between RIC and LEAP-RBP is ~89%? One might expect that RIC would give more false positives (or background) than a 10%, however it seems that all mRNA-binding proteins identified by RIC are also identified by LEAP-RBP – what is the advantage then? Of course with LEAP you do get other proteins that bind other RNAs rather than mRNAs but still is very surprising. If you then look at the Venn diagram with other organic phase separation methods, then the overlaps are not so great as compared to the overlap with RIC. Could that be just due to the technique itself? Looking at the panel a LEAP-RBP has a good overlap with all of the other techniques (709 proteins) although has little overlaps with other techniques individually but still, 286 (~16%) have not been identified in other techniques, are these real binders or is it the same background we find in the comparison with the RIC? What should be considered real binders – those proteins identified by all methods?

Reviewer #2 (Remarks to the Author):

RNA binding proteins (RBPs) from RNA nucleoprotein complexes in biological systems regulate every aspect of RNA biology. Recent efforts in the identification of the proteins that bind to the RNAs are largely based on the use of proteomics technique followed by the RNA-dependent enrichment from the various biological systems. Such studies revealed that there are hundreds of novel RBPs whose RNA-related functions and mechanisms for RNA-protein interactions remained largely unknown.

In this study, the authors present the improved method and criteria for the determination of such unconventional RBPs. It is rather clear that the method developed by the authors, LEAP-RBP (Liquid-Emulsion-Assisted-Purification of RNA-Bound Protein), achieve highly specific enrichment of the proteins that are cross-linked to the RNAs based on the data presented throughout the manuscript. The authors first show that while AGPC based RBP enrichment have been widely utilized, conventional approach is highly prone to the enrichment of the proteins that are free from the RNA and thus migrate to their respective size even without the RNase treatment in SDS-PAGE. Authors thus introduced the additional phase separation and RNA-dependent precipitation step, LEAP-RBP, to effectively deplete such RNA free proteins at the interphase of AGPC. The method was further developed through the introduction of the DNA removal step for the depletion of the DNA binding proteins which have also been repeatedly found as the major contaminants in the RBP profiling studies. Authors then further demonstrate that LEAP-RBP have advantages for RBP identification in terms of sensitivity and specificity over the previously developed RBP profiling methods, especially in comparison to the similarly designed OOPS and XRNAX methods.

However, the criteria that authors argue to be newly defined does not seem to clearly distinguish themselves from the statistical analysis that have been performed in previous RBP profiling studies. It is also fair to say that previous studies have also provided a number of means, including RNase and no crosslinking control, to exclude such RNA crosslinking independently enriched proteins. Actually it appears that LEAP-RBP is an improved version or an analogue of the OOPS or XRNAX method, in which the core improvement in the LEAP-RBP is the additional phase separation and RNA-dependent precipitation steps. Therefore, this paper seems more suited to a specialized journal in terms of methodology. I would recommend the paper is able to be further considered for publication in Nature Communications as long as the authors could validate their claims on the LEAP-RBP method for RBP identification based on the orthogonal experimental approaches or solidly demonstrate the biological utility of LEAP-RBP method for biological discovery in RNA-protein interaction via e.g., a condition-dependent RBP profiling experiments. Few of the detailed comments and the suggestions are provided below.

Comments

1. Page 3. 51 As the authors also noted here, condition dependent changes in the profile of RNA bound proteome are critical for the deeper understanding of RNA biology. Authors may also want to consider the profiling of condition dependent changes in the landscape of RNP complex formation based on the LEAP-RBP method. The result would also serve as the orthogonal validation for the method's accuracy and the ability to detect RNA-protein interactions that are occurring in vivo.

2. Page 7. 181 Orthogonal validation for the specificity of the distinct methods to the "RBPs" and "non-RBPs" are largely based on SRA assay in the manuscript. As I noted above, inclusion of the new data based on other orthogonal validation approaches would be useful for the manuscript. The result and analysis may also add some biological insight to the manuscript. It is also important to

note that GAPDH protein's RNA related function and RNA binding activity is relatively well known (1,2).

3. Fig2g It would be helpful if the total number of RBPs identified by each method are included in the figure. 1794 proteins identified via the LEAP-RBP method likely include some of the novel RBPs whose RNA binding activity has not been reported. The authors claimed that LEAP-RBP has the unique ability to identify bona fide RBPs. Such novel RBPs would be good candidates for the orthogonal experimental validations.

4. Fig3a While the authors present more detailed and clearer explanation for the SILAC based S/N ratio estimation in the supplementary Fig9 and the other parts of the manuscript, Fig3a does not seem to be the accurate representation of the S/N that they determined and thus somewhat misleading. For example, a +CL sample may contain more than 1 part of the non RNA-cross-linked proteins, due to the enrichment of the secondary protein that is cross-linked to the RNA-cross-linked protein for instance. Even if the authors provide evidence that suggests equal partitioning of the noise in general (Fig3g), the possibility still remains. It is also important to note that equal partitioning of the noise may be true for some proteins but not for the others, for the same reason provided above. To my understanding, the figure represents the authors' logic behind the determination of S/N. Nonetheless, it would be more appropriate to revise the figure or provide clearer explanation on the figure legend.

References

- 1) Singh, R. and M. R. Green (1993). "Sequence-Specific Binding of Transfer RNA by Glyceraldehyde-3-Phosphate Dehydrogenase." *Science* 259(5093): 365-368.
- 2) Castello A, Hentze MW, Preiss T. Metabolic Enzymes Enjoying New Partnerships as RNA-Binding Proteins. *Trends Endocrinol Metab.* 2015 Dec;26(12):746-757. doi: 10.1016/j.tem.2015.09.012. Epub 2015 Oct 28. PMID: 26520658; PMCID: PMC4671484.

Response to Referees

Reviewer #1:

Summary: *This manuscript reports a novel method to isolate RNA binding proteins. Given the importance of the posttranscriptional regulation for the cell fitness, this objective has been pursued by numerous research groups worldwide and several methodologies have been developed with this aim in the last decade. However, one of the major concerns of these recent methods was the specificity. Their workflow builds on the previously published organic phase separation methods, which enrich RBPs according to their physiochemical properties and are deemed to provide improved specificity and efficiency. In addition, these methods, as opposed to RIC, enable the identification of all proteins associated with any class of RNA instead just messenger RNAs, resulting in a more complete picture of the RBPome.*

The main strength of this manuscript is the development of a new method to specifically isolate RBPs and a way (signal to noise ratio) to distinguish real binders to background. As the manuscript mentioned, UV-CL is not very efficient and one of the main criticisms for techniques to isolate RBPs has always been the 'noise', so I think that the authors have found a very elegant method to identify real RNA-binders. For the most part, this manuscript is clearly presented, concise, and the interpretations are not overstated. The method is extremely well described and every step of it is justified.

The reviewer has neatly summarized the broad impact and significance of the study; we are grateful for the supportive comments.

We agree with the reviewer that an essential question in efforts to define RNA interactomes is that of specificity. To help clarify on that point, we now include additional documentation of the criteria used for assignment of RNA-binding activity (see below, *Methods. Criteria for assignment of RNA-binding activity*).

2) *Even though I consider this manuscript original and doubtless will be of interest to the RNA community I have three major concerns:*

Concern #1) *The authors evaluate throughout the manuscript GO-annotated RBPs and non-RBPs however they do not describe how did they selected these proteins and would be interesting to know whether these proteins have any known feature like the RNA binding domains, the type of RNA that they bind, or the number of targets – this may help to the interpretation of the data.*

In this regard, they considered within the GO-annotated RBPs two subgroups, depending on the presence of RNA binding domains. From their observations when applying LEAP-RBP they concluded that 'only LEAP-RBP clearly distinguishes RBPs with low S/N (e.g., RPN1, TRAP α) from non-RBPs like GRP94 (red box; Fig. 1f). These results suggest

GRP94 (a GO-annotated RBP) is a false positive and indicates TRAP α (non-RBP) is a false negative'. Could this be due to the number of targets of the non-canonical RBP – GRP94? Or could be just that these other proteins are less sensitive to the UV-CL as they are performing other activities within the cell and their RNA binding activity is minimal? My point is, this method is clearly great for the canonical RBPs but is it too restrictive for proteins that bind just a few targets or targets not highly expressed?

Regarding the question of how proteins were selected, proteins identified by SILAC LC-MS/MS analysis of LEAP-RBP fractions as significantly UV-enriched were selected for orthogonal validation of RNA binding activity by SRA and immunoblot. In the analysis depicted in **Fig.1**, proteins lacking prior GO-RBP annotation (i.e., β -tubulin) as well as GO-annotated RBPs containing (e.g., HuR) or lacking (e.g., GRP94) established RNA-binding domains were selected, an approach similar to that suggested by the reviewer. GAPDH is particularly noteworthy as a high interest non-canonical GO-RBP lacking a known RNA binding motif [1, 2]. For all MS-based analyses, proteins were binned as RBPs or non-RBPs according to their GO-annotation status (**Methods**) [3]. The same approach was used when evaluating samples by SRA and immunoblot initially (**Fig. 1c, f**). This was considered appropriate given that GRP94 (a GO-annotated RBP) was still detectable in INP fractions, albeit RNase-insensitive and non-enriched by SILAC LC-MS/MS (**Fig. 1f, 2i**), raising the possibility that it may be a bona fide RBP with low S/N. However, all proteins assayed by immunoblot were identified as significantly UV-enriched in LEAP-RBP fractions by SILAC LC-MS/MS, and according to their S/N ratios, would be predicted to display RNase-sensitivity by SRA (**Fig. 2i, 3i**). In support of this, nucleolin (NCL) displays a lower \log_2 (S/N) ratio in INP fractions than GRP94, β -tubulin, and GRP78 in LEAP-RBP by SILAC LC-MS/MS but is RNase-sensitive by SRA and immunoblot (NCL, “I”; **Fig. 2f**). However, only β -tubulin was detected in LEAP-RBP fractions by SRA and immunoblot and is RNase-insensitive (**Extended Data Fig. 1**). We suspect UV-enrichment of β -tubulin may in part reflect low abundance, non-specific UV-crosslinking to RNA-bound proteins (**Supplementary Note 4e**). We also considered that UV-crosslinking of non-RNA nucleotide substrates (e.g., nicotinamide- and flavin-dinucleotide cofactors of dehydrogenases, such as GAPDH, or nucleotide substrates of nucleotide binding proteins such as GRP94 and β -tubulin) to proteins could result in significant, albeit low efficiency UV-enrichment by SILAC LC-MS/MS, with such proteins also scoring as RNase-insensitive by SRA/immunoblot (**Supplementary Note 7c**; [4, 5]). Therefore, it was only considered tenable to bin proteins displaying discernible RNase-sensitivity in LEAP-RBP fractions as RNA-binding proteins.

We agree with the reviewer that proteins which are inefficiently UV-crosslinked to RNA are expected to be less abundant in LEAP-RBP fractions and may fall below the detection limit of SRA/immunoblotting. Therefore, we revised sections of the manuscript to explicitly state this possibility (see below). It is clear, though, that LEAP-RBP achieves higher sensitivity and specificity as compared to other current methods (**Fig. 5, 6**). Therefore, this concern, valid as it is, is a more general limitation rather than an exclusive limitation of the presented methodology. As evidence of this, all RBPs with well-defined RNA-binding domains were successfully validated by SRA/immunoblot, with many displaying

in vivo UV-crosslinking efficiencies between 10–20% (**Extended Data Fig. 1a**). Importantly, we were able to detect and validate RNA-binding of non-canonical RBPs displaying UV-crosslinking efficiencies <0.3% by SRA and immunoblot (e.g., RPN1) and so proteins as the reviewer describes, should they exist, are expected to have quite low UV crosslinking efficiencies. Such proteins do require considerable study to further validate apparent RNA binding functions. RPN1, for example, resides in close physical proximity to endoplasmic reticulum-bound ribosomes and so the observed RNA binding may represent low frequency crosslinking to rRNA [6].

Revisions in response to concern (edits are in italicized blue font)

Methods. Criteria for assignment of RNA-binding activity

Protein displaying +CL/–CL ratio >0 in LEAP-RBP fractions by SILAC LC-MS/MS were considered high or low confidence RBPs based on their observed enrichment efficiency (S/N) and abundance (%TP). However, only those displaying discernible RNase-sensitivity by SRA and immunoblot were considered bona fide RNA-binding proteins. Proteins which remained RNase-insensitive or undetectable were considered non-RBPs regardless of GO-annotation status (e.g., GRP94, a GO-annotated RBP). However, because the inability to detect a protein by SRA and immunoblot could be due to their low RNA-bound protein abundance, it was not considered confirmation that it lacks RNA-binding activity (Supplementary Methods).

Supplementary Methods. Description of terminologies.

Compared to SILAC LC-MS/MS, evaluating S/N by SRA was considered more accurate because non-specific UV-crosslinking does not contribute to the displayed S/N of proteins (i.e., S/N of observable protein quantities). Therefore, proteins which appeared “**RNase-sensitive**” (ISI > 0) by SRA analysis were considered “**RNase-sensitive RBPs**” or “**bona fide RBPs**” while proteins displaying “**positive S/N ratios**” by SILAC LC-MS/MS analysis were considered “**UV-enriched**” (i.e., +CL/–CL ratios >1). The “**S/N of RBPs**” refers to the S/N of GO-annotated RBPs or RNase-sensitive RBPs, while the “**S/N of non-RBPs**” refers to the S/N of proteins without prior GO-annotations (GO:RBP). During MS data analysis, S/N ratios for non-RBPs represented RNA-bound protein enrichment (SILAC and non-SILAC) despite the premise that non-specific UV-crosslinking to non-RNA substrates and/or UV-dependent enrichment of free protein *can result in* their apparent “UV-enrichment” (i.e., +CL/–CL ratios >1 and S/N ratios >0).

Supplementary Methods. Evaluating UV-dependent enrichment and S/N by SDS-PAGE RNase-sensitivity Assay (SRA).

“**RNase-sensitivity**” refers to the fold-change in observed protein quantity (denoted by “**ΔObs.**” or “**Δlog₂(S+N)**”) between RNase-treated and untreated samples as shown in equation (1). Proteins were considered “**RNase-sensitive**” if they displayed discernible RNase-sensitivity and “**RNase-insensitive**” if they did not. All RNase-sensitive proteins were considered RNase-sensitive RBPs or bona fide RBPs while “**RNase-insensitive proteins**” were considered non-RBPs or RBPs with low S/N. *UV-enriched proteins*

displaying +CL/-CL ratios >1 by SILAC LC-MS/MS and which remained undetectable by SRA and immunoblot were not considered bona fide RBPs regardless of GO-annotation status (e.g., GRP94, a GO-annotated RBP). However, because this could be due to their low RNA-bound abundance, it was not considered confirmation that a protein lacks RNA-binding activity.

Concern #2) When the authors make the comparison with other published methods (OOPs, RIC, TRAPP, etc...) I can clearly see the differences across the methods, but it hard to believe that proteins like GAPDH show up as RNA-binder when applying some techniques and no other and the positive results have to be considered background or false positive. In this regard, it has been described that glycolytic enzymes do bind RNA (recently reviewed in Wegener M & Dietz KJ. The mutual interaction of glycolytic enzymes and RNA in post-transcriptional regulation. RNA. 2022 doi: 10.1261/rna.079210.122), and for some of them the specific targets have been identified (the most recent example is the enolase Huppertz I, et al. Riboregulation of Enolase 1 activity controls glycolysis and embryonic stem cell differentiation. Mol Cell. 2022 Jul 21;82(14):2666-2680.e11. doi: 10.1016/j.molcel.2022.05.019.). As I mentioned before, would be very helpful to monitor in these western blots RBPs with different features, RBPs that bind mRNAs, tRNAs, miRNAs, circRNAs, etc. to make sure that this method is not restrictive to proteins that bind a certain type of RNA or have highly expressed targets. Indeed, the RNase Cocktail (AM2286, Invitrogen) used by the authors, would not be helpful for those RBPs that bind circular RNAs (not sensitive to the RNases) or even those proteins that bind DNA-RNA hybrids (treatment with RNase H1 would be helpful).

The reviewer raises the question of the RNA binding activity of glycolytic enzymes and in the case of GAPDH, how to justify negative results and apparent discordance with conclusions from previous studies. First, and as the reviewer notes, a major concern surrounding the RNA interactome methods has been their specificity [7-12]. We provide substantial data demonstrating that this concern is well-justified. For example, we demonstrate how non-specific UV-enrichment can result in false assignment of RNA-binding activity based on the conventional criteria (**Supplementary Note 4d, 7e, d**) [13-23]. To this point, the RNA-binding activity of many GO-annotated RBPs has been inferred using this convention (**Supplementary Fig. 33h, Supplementary Note 8b**). This concern is also evident in the recent enolase 1 study noted by the reviewer (Huppertz et al.). We identified enolase 1 as significantly UV-enriched in LEAP-RBP fractions by SILAC LC-MS/MS but with a $\log_2(S/N)$ ratio of -0.8 we consider it a “low confidence” RBP. Our findings are also consistent with the data reported in **Fig. 1B** of Huppertz et al., where these authors note “*only a small fraction of ENO1 is bound by RNA*”. Indeed, proteins which remained undetected in LEAP-RBP fractions would be expected to display, at most, UV-crosslinking efficiencies orders of magnitude lower than what we observe for bona fide RBPs (which can be as high as 20%). These dramatic differences in crosslinking efficiencies illustrate the need for quantitative signal-based metrics for assessing RNA binding function and highlight the challenges of establishing physiologically relevant interactions with RNAs when the protein of interest displays very low crosslinking efficiency.

To expand on this perspective, we considered it appropriate to bin UV-enriched proteins identified by SILAC LC-MS/MS of LEAP-RBP fractions as low or high confidence RBPs based on their observed enrichment efficiency (S/N) and abundance (%TP). However, we considered it untenable to bin UV-enriched proteins which are RNase-insensitive or undetectable by SRA/immunoblot assay as bona fide RBPs (as per response above). As we note, the principles behind SRA and the ability to provide direct evidence that proteins are UV-crosslinked to sizable RNA populations has been accepted by many experts in the field [5, 9, 24]. However, and despite widespread application of UV crosslinking to identify RNA interactomes, it is hampered by low method specificity and biases in signal recovery, as evidenced by direct benchmark comparisons (**Figs. 5,6**). Returning to the glycolytic enzyme question and specifically enolase 1, the Huppertz et al. study provides CLIP-seq data showing UV-enrichment of many hundreds of sizable RNA-substrates ([25]; **Figure 1c**) and so one would expect Eno1 to be validated in our system, assuming sufficient UV-crosslinking efficiency. Huppertz and colleagues note that “*ENO1 is among the top 100 most abundant proteins in the cell.*” [26] and “*For the enzyme-RNA interaction to play a role, a considerable fraction of ENO1 needs to be associated with RNA*” [25]. Yet, the authors provide clear evidence to the contrary on this point (their **Fig. 1B**), and our data are consistent with their findings. We also want to strongly emphasize that low abundance RNA-binders are not viewed dichotomously (i.e., true vs false) from an S/N-based perspective. As described in the manuscript (**Methods, Supplementary Note 7d**), the proposed metrics for evaluating “RBP confidence” assumes that higher UV-crosslinking efficiency is indicative of physiological relevance. Given the efficiency and unbiased signal recovery of the LEAP-RBP method (**Supplementary Note 3**), the inability to detect RNA-bound proteins by SRA and immunoblot is indicative, at the very least, of the challenges in establishing the physiological significance of such weak protein-RNA interactions, as noted above (**Supplementary Note 6d**).

The reviewer is concerned that the RNase-treatment conditions utilized in the SDS-PAGE RNase-sensitivity Assay (SRA) may be limited to proteins that bind restricted classes of RNAs. While this may be a valid concern, to our knowledge there isn't a clear example of an RNA-binding protein which exclusively binds cRNAs or DNA:RNA hybrids in vivo. Importantly, although cRNAs are resistant to exonucleases they are sensitive to endonucleases and so the RNase cocktail used in the present study, comprised of RNase A and RNase T1, both of which are endonucleases, would be expected to digest RBP-cRNA complexes [27]. Nonetheless, to directly address this concern, we provide new data demonstrating efficient RNase-digestion using the recommended conditions (**Supplementary Fig. 4a**).

Concern #3) *One last comment is about the overlaps showed in the supplementary data (Suppl. Fig 14). One of the main criticisms of RIC that I mentioned in the beginning of the revision has always been the specificity, and the application of the LEAP-RBP is deemed to provide improved specificity and efficiency, how is it possible then that the overlap between RIC and LEAP-RBP is ~89%? One might expect that RIC would give more false*

positives (or background) than a 10%, however it seems that all mRNA-binding proteins identified by RIC are also identified by LEAP-RBP – what is the advantage then? Of course with LEAP you do get other proteins that bind other RNAs rather than mRNAs but still is very surprising. If you then look at the Venn diagram with other organic phase separation methods, then the overlaps are not so great as compared to the overlap with RIC. Could that be just due to the technique itself? Looking at the panel a LEAP-RBP has a good overlap with all of the other techniques (709 proteins) although has little overlaps with other techniques individually but still, 286 (~16%) have not been identified in other techniques, are these real binders or is it the same background we find in the comparison with the RIC? What should be considered real binders – those proteins identified by all methods?

We appreciate the reviewers thoughtful query and focus on rigorous identification of RNA-binding proteins. To this point, the reviewer is surprised by the large overlap between the LEAP-RBP and RIC MS datasets. In particular, the reviewer notes that one might expect the lower specificity of the RIC method to result in a higher false positivity rate. As addressed in **Supplementary Note 8b**, the RIC studies utilize a non-SILAC proteomics approach and therefore cannot distinguish UV-dependent enrichment of RNA-free protein, a finding we highlight in the current study. Furthermore, comparisons of RIC eluants isolated from UV-crosslinked and non-crosslinked cells by SDS-PAGE and immunoblot are typically done after RNase-treatment and without the inclusion of untreated (–RNase) controls, which are necessary to demonstrate that UV-enriched protein is RNA-bound ([15]; see **Fig. 1d, e**; [18]; see **Fig. 2c, d**). That said, RIC fractions isolated from UV-crosslinked cells show a general lack of RNase-insensitive proteins by SRA and Coomassie Blue staining, indicating a low potential for such false positives (our **Fig. 6a**). In essence, RIC studies utilizing a SILAC LC-MS/MS approach would provide more accurate quantitation of free protein recovery which is useful for comparative experiments aimed at identifying differences in RNA-bound protein abundance (**Supplementary Note 4d**).

The reviewer asks whether the poor overlap of proteins identified as significantly UV-enriched by the LEAP-RBP method and other organic phase separation methods is due to differences in the techniques themselves. Indeed, because free proteins exhibit both UV- and method-dependent enrichment, non-SILAC based comparisons utilizing different methods with low specificity for RNA-bound protein are expected to result in significant UV-enrichment of different background proteins (i.e., profiles) lacking RNA-binding activity (**Supplementary Note 4d**). In support of this view, a non-SILAC comparison utilizing methods such as Ptex, that utilizes different RNA-centric enrichment conditions than other organic phase separation methods (e.g., OOPs) yields exclusive, significant UV-enrichment of many proteins lacking prior GO-RBP annotation (1075 proteins; **Supplementary Fig. 14a**).

The lower number of proteins identified by the RIC method can be attributed to a higher limit of detection (i.e., lower sensitivity). As addressed more thoroughly in

Supplementary Note 8, the presence of one or a few RBPs recovered at higher efficiency than the majority of other RBPs can lead to a diminished or higher limit of detection. This is evidenced by the large difference in %TP contributions of the most abundant proteins in RIC fractions as compared to LEAP-RBP fractions (RIC: 7.85, 4.22, 2.1, and 1.96 vs LEAP-RBP: 6.78, 4.95, 3.34, and 3.29; **Fig. 6h, Supplementary Note 8a, b**). The difficulty of achieving high sampling depth during MS-based analysis of proteomic samples containing one or a few highly abundant proteins is well documented [28].

The reviewer asks whether proteins identified as UV-enriched exclusively by the LEAP-RBP method are real RNA-binders or if they should be considered background. Because these proteins were identified in LEAP-RBP fractions using a SILAC LC-MS/MS approach, their UV-enrichment is likely real (i.e., true background would display a mean $\log_2(+CL/-CL)$ distribution of 0; **Supplementary Note 7e**). However, these are considered low-confidence RBPs as their RNA-bound protein abundance is orders of magnitude below the abundance of RNase-sensitive RBPs with well-defined RNA-binding domains and they display lower enrichment efficiency (**Supplementary Fig. 17b**). As discussed above, we consider it untenable to conclude that their UV-enrichment is solely due to covalent coupling to RNA interactors without further orthogonal validation (e.g., SRA and immunoblot (**Supplementary Note 7c, d**)). As for their exclusive UV-enrichment status, we attribute this to the markedly lower free protein recovery by the LEAP-RBP method, as compared to related methods, the consequence of which is enhanced enrichment (S/N) and detection of low-abundance RNA-bound proteins (**Supplementary Fig. 17d, Supplementary Note 7b, c**). However, as we state in the **Methods**: “During MS data analysis, S/N ratios for non-RBPs represented RNA-bound protein enrichment (SILAC and non-SILAC) despite the premise that non-specific UV-crosslinking to non-RNA substrates and/or UV-dependent enrichment of free protein can result in their apparent “UV-enrichment” (i.e., +CL/-CL ratios >1 and S/N ratios >0)”.

As an overarching comment, the reviewer asks whether real binders should be limited to those proteins identified by all methods. This query relates back to the main premise of the current study: *UV-enrichment alone is an insufficient criterion for assigning RNA-binding activity*. As discussed in **Supplementary Note 6d**, β -tubulin was identified as significantly UV-enriched by LEAP-RBP, INP, XRNAX, OOPs, Ptex, TRAPP, RIC, and eRIC methods (**Supplementary Table 1**). However, SRA and immunoblot analysis of LEAP-RBP fractions suggests this is a widely observed false positive (**Extended Data Fig. 1b**). Two other observations complicate meaningful meta-analysis: only some methods display sufficient specificity for RNA-bound protein to achieve significant UV-enrichment of low abundance RNA-bound proteins, and RBP profiling studies utilizing non-SILAC comparisons can result in method-specific, significant UV-enrichment of different background proteins (**Supplementary Note 4d, 7e, 8a, b**). For example, comparison of OOPs fractions isolated from UV-crosslinked and non-crosslinked cells shows how non-SILAC comparisons, as previously reported [16, 17, 29], can yield UV-enrichment of RNase-insensitive (RNA-free) β -tubulin (**Fig. 6c**). Indeed, and relating to the reviewers’ concerns, the complexity and challenges inherent in rigorous identification

of RNA-binders based solely on their significant UV-enrichment has been exacerbated by the growing number of RNA interactome ID methods, with only limited commentary to the limitations of the techniques. Nonetheless, RNA-bound protein abundance *is* highly conserved regardless of method specificity, if signal recovery is comparable (**Supplementary Note 4f**). As discussed in **Supplementary Note 7c**, LEAP-RBP and SRA is a widely accessible method to acquire the necessary orthogonal evidence of direct RNA-binding for UV-enriched proteins.

Reviewer #2:

Summary: "...In this study, the authors present the improved method and criteria for the determination of such unconventional RBPs. It is rather clear that the method developed by the authors, LEAP-RBP (Liquid-Emulsion-Assisted-Purification of RNA-Bound Protein), achieve highly specific enrichment of the proteins that are cross-linked to the RNAs based on the data presented throughout the manuscript. The authors first show that while AGPC based RBP enrichment have been widely utilized, conventional approach is highly prone to the enrichment of the proteins that are free from the RNA and thus migrate to their respective size even without the RNase treatment in SDS-PAGE. Authors thus introduced the additional phase separation and RNA-dependent precipitation step, LEAP-RBP, to effectively deplete such RNA free proteins at the interphase of AGPC. The method was further developed through the introduction of the DNA removal step for the depletion of the DNA binding proteins which have also been repeatedly found as the major contaminants in the RBP profiling studies. Authors then further demonstrate that LEAP-RBP have advantages for RBP identification in terms of sensitivity and specificity over the previously developed RBP profiling methods, especially in comparison to the similarly designed OOPS and XRNAX methods."

We thank the reviewer for highlighting the value and broad utility of the research presented in this submission.

Concern #1) ...the criteria that authors argue to be newly defined does not seem to clearly distinguish themselves from the statistical analysis that have been performed in previous RBP profiling studies.

We appreciate the reviewer's concern. To this point though, we define two new, key metrics; protein-specific S/N ratios and %TP_s (percent of total protein that is RNA-bound) (**Supplementary Methods**). Importantly, we use these metrics to perform many analyses lacking in prior RBP profiling studies. This is exemplified by examining how prior RNA-centric methods have been applied to identify global, condition-dependent changes in RNA-binding activity. In principle, a change in the amount of a protein that is UV-crosslinked to RNA (i.e., RNA-bound) *in vivo* without a proportional change in its total abundance is likely due to modulation of protein-RNA interactions/RNA binding [24]. However, without considering protein-specific S/N ratios (**Supplementary Note 6b**), UV-dependent enrichment of free protein (**Supplementary Note 4d, 7e**), and method specificity for RNA-bound protein (**Supplementary Note 4f**), the significance of observed

differences in protein (or RBP) recovery in the condition-dependent studies is uncertain. To address this question methodologically, we propose setting S/N limits for comparative experiments to exclude proteins whose observed abundance in RNP fractions is not representative of RNA-bound protein abundance (**Supplementary Note 6c**). Assuming robust and efficient recovery of RNA-bound protein, the S/N-based criteria presented here allows the investigator to identify differences in observed protein abundance which are more likely to reflect a difference in RNA-bound protein abundance, and conversely less likely to reflect a difference in, for example, free protein recovery. This is especially critical in scenarios involving glycolytic enzymes where only a very small fraction of the total protein is potentially complexed with RNA, and based on their S/N ratios, recovered protein is expected to be more representative of free protein than RNA-bound (**Supplementary Note 6c**). To provide experimental support for the novelty and scientific utility of our analytical approach, we append a list of RBPs which were identified as having significant condition-dependent changes in RNA-binding in previous RBP profiling studies (which utilized the XRNAX or TRAPP methods). We included their estimated fold-change in RNA-binding and $\log_2(S/N)$ ratios generated with their reported $\log_2(+CL/-CL)$ ratios. These studies were chosen because both utilize a SILAC-based approach, allowing more accurate assessment of S/N (**Supplementary Note 6a**). In the XRNAX study, 633 proteins were found to display a significant change in RNA binding during arsenite stress. However, RNA-binding is a bit of a misnomer, as the reported fold-change is calculated using observed protein abundance (S+N) rather than the RNA-bound fraction ([22]; **Fig. 5A**). For example, the protein CASC3 is listed as having a 1.35X fold-change in RNA-binding following a 30 min arsenite treatment. From an S/N-based perspective, a 1.35X fold-change in observed quantity of CASC3, which displays a $\log_2(S/N)$ ratio of -1.23 in XRNAX fractions by SILAC LC-MS/MS, would require a 2.17X fold-change in RNA-bound quantity, assuming noise is constant. However, from the SILAC data only about 1/3rd of the observed CASC3 peptide intensity originates from RNA-bound CASC3 and so the observed fold-change in CASC3 abundance upon arsenite stress is likely skewed by unmeaningful changes in free protein recovery (**Supplementary Note 6c**). Based on the proposed S/N-based criteria, 499 proteins of the 633 analyzed (~73%) would be considered representative of RNA-bound protein (i.e., display S/N ratios >3 and >75% RNA-bound; **Supplementary note 6b, c**). Demonstrating the importance of this selection criteria, ~95% of the 499 proteins displaying S/N ratios >3 are GO-annotated RBPs (~95%), whereas ~66% of the 134 of the proteins displaying S/N ratios <3 are GO-annotated RBPs. Importantly, GO-annotated RBPs in the latter group include many of the suspected false positives (e.g., plectin and histone H1.0) whose RNA-binding activities have been inferred by their UV-enrichment status in prior RIC experiments (non-SILAC) and have not been directly validated (**Supplementary Fig. 33h, Supplementary Note 8b**). Similarly, in the TRAPP study, 289 proteins were reported to display a significant change in RNA binding during sorbic acid stress. However, like the XRNAX study, the reported fold-change in RNA binding represents observed protein abundance ([20]; their **Fig. 5a**), not RNA-bound protein abundance. In the TRAPP study, 57% of the 289 proteins with reported changes in RNA-binding display S/N ratios >3. Critically, only 3 of the top 30 proteins which are reported to exhibit the largest change in RNA binding display S/N

ratios >3 (~10%). In these scenarios, evaluating isolation fractions by methods such as SRA and immunoblot analysis are critical for demonstrating that observed differences in protein recovery reflect a change in RNase-sensitive or RNA bound protein.

Many RBP profiling methods referenced in the current manuscript have reported global changes in RNA-binding activity during cellular stress-responses [18-20, 22]. However, none of the observed differences in what formally are protein recovery data have been orthogonally validated. Because of this, we consider the comparative experiment shown in **Fig.4** of current manuscript to be a more rigorous, albeit small-scale, validation that the presented methodology allows assessment of both RNA-bound and total protein abundance. That is, we demonstrate that the observed protein abundance of RBPs in LEAP-RBP fractions and assayed by SDS-PAGE and immunoblot are representative of RNA-bound abundance based on their displayed S/N ratios (**Supplementary Note 6c, 6d**). Importantly, and as revealed by the LEAP-RBP method, many proteins don't exhibit UV dose-sensitive crosslinking efficiencies (e.g., TIA1 and HuR; **Extended Data Fig. 2d, Supplementary Note 3c**). Because this may impact the correlation between RNA-binding activity and observed RNA-bound protein abundance in an RBP-specific manner (i.e., linear range), it was not considered appropriate to make claims regarding changes in RNA-binding activity. Nonetheless, we believe that the presented methodical and analytical framework provides the tools necessary for understanding the relationship between observed differences in RNA-bound protein abundance and in vivo RNA-binding activity.

Concern #2) It is also fair to say that previous studies have also provided a number of means, including RNase and no crosslinking control, to exclude such RNA crosslinking independently enriched proteins.

The reviewer notes that previous studies have provided several means to exclude UV-enrichment of RNA crosslinking independently enriched (i.e., free) proteins. Indeed, as noted in the current manuscript, some previous studies utilized a SILAC-based approach to control for UV-dependent enrichment of free protein. However, it is quite clear that non-SILAC comparisons using methods with low specificity for RNA-bound protein have resulted in false-assignment of RNA-binding activity based on the conventional criteria (i.e., UV-enrichment). Our response to this primary concern is addressed above, in the responses to reviewer 1.

Concern #3) Actually it appears that LEAP-RBP is an improved version or an analogue of the OOPS or XRNAX method, in which the core improvement in the LEAP-RBP is the additional phase separation and RNA-dependent precipitation steps. Therefore, this paper seems more suited to a specialized journal in terms of methodology.

We appreciate the reviewers' comments and concern though feel that separating the methodology from the analytical framework substantially diminishes the impact and value of the study. Indeed, the reviewer acknowledges the improved specificity of the LEAP-

RBP method for RNA-bound protein but does not note the conceptual advancements gained by the introduction of a new S/N-based perspective and signal-based metrics (e.g., protein-specific S/N ratios and %TPs) which allow one to reach conclusions regarding the specificity of LEAP-RBP. We feel this is a critical element and so included comparisons between the INP and LEAP-RBP methods to directly demonstrate the paradoxical decrease in UV-enrichment specificity for GO-annotated RBPs despite a favorable increase in method specificity for RNA-bound protein (**Supplementary Note 8a**).

Concern #4) I would recommend the paper is able to be further considered for publication in Nature Communications as long as the authors could validate their claims on the LEAP-RBP method for RBP identification based on the orthogonal experimental approaches or solidly demonstrate the biological utility of LEAP-RBP method for biological discovery in RNA-protein interaction via e.g., a condition-dependent RBP profiling experiments. Few of the detailed comments and the suggestions are provided below.

The reviewer suggests that the current manuscript should be considered for publication in *Nature Communications* if the utility of the LEAP-RBP method for biological discovery is clearly demonstrated by performing a condition-dependent RBP profiling experiment. We direct the reviewer to the experiment depicted in **Fig. 4**, demonstrating the ability of the LEAP-RBP method to identify differences in RNA-bound protein abundance between four cell lines (**Fig. 4h**). As noted above, conventional RBP profiling evaluates the observed abundance of proteins in isolated RNP fractions without assessing the contributions of RNA-bound vs unbound counterparts. Therefore, we consider the comparative experiment shown in **Fig. 4** to be a more rigorous demonstration that the LEAP-RBP method allows assessment of both RNA-bound and total protein abundance. Relating to the utility of the LEAP-RBP methods for biological discovery, the biological significance of this experiment was clearly noted in the accompanying results section titled “*LEAP-RBP allows robust assessment of RNA-bound protein abundance and sensitive detection of $\Delta\log_2(S)$* ”. Here, we observed higher RNA-bound protein abundance for three non-canonical ER RBPs in rat insulinoma (pancreatic β) cells (832/13) without a comparable change in their total abundance (RPN1, TRAP α , and LRRC59; **Fig. 4i**). Notably, such RBP profiling experiments have not been provided in some prior studies (e.g., Ptex) published in *Nature Communications* [23].

Concern #5) As the authors also noted here, condition dependent changes in the profile of RNA bound proteome are critical for the deeper understanding of RNA biology. Authors may also want to consider the profiling of condition dependent changes in the landscape of RNP complex formation based on the LEAP-RBP method. The result would also serve as the orthogonal validation for the method’s accuracy and the ability to detect RNA-protein interactions that are occurring in vivo.

Condition-dependent changes in the profile of the RNA-bound proteome is critical for deeper understanding of RNA-protein interaction occurring *in vivo*. To that end, high

%TPs and efficient, unbiased recovery of RNA-bound protein is necessary and is a goal largely met by LEAP-RBP (**Supplementary Note 4f**). Supporting evidence demonstrating the critical importance of the S/N-based criteria presented in this manuscript is provided in the first response to reviewer 2. A scientifically meaningful advance in the field would be to demonstrate that condition-dependent changes in RBP occupancy are biologically relevant. That objective is non-trivial and is a study unto itself.

Concern #6) Orthogonal validation for the specificity of the distinct methods to the “RBPs” and “non-RBPs” are largely based on SRA assay in the manuscript. As I noted above, inclusion of the new data based on other orthogonal validation approaches would be useful for the manuscript. The result and analysis may also add some biological insight to the manuscript. It is also important to note that GAPDH protein’s RNA related function and RNA binding activity is relatively well known (1,2).

The reviewer suggests the inclusion of new data based on other orthogonal validation approaches noted above. However, it’s not clear what orthogonal validation approaches are being referenced. Regarding GAPDH, we address the discordance between our results and prior studies in **Supplementary Note 7d**. Importantly, we have not, and do not, consider the inability to validate a protein by SRA and immunoblot to be indicative of a lack of RNA-binding. As stated in the text: “*Because many well-established RBPs were found to have UV-crosslinking efficiencies between 1-20% (0.4 J/cm², 254 nm), the physiological relevance of proteins in LEAP-RBP fractions undetected by SRA and immunoblot is, in our view, questionable*”. Importantly, when examined by the new metrics we describe herein, glycolytic enzymes in general display distinctly low UV crosslinking efficiencies as well as very low RNA occupancy levels. We feel these quantitative findings are critical to evaluating proposed functions for these proteins in RNA biology. When compared to bona fide RBPs, glycolytic enzymes display very modest interactions with RNA. The physiological significance of this remains in our view uncertain. We recognize that our perspective challenges a growing orthodoxy; by providing new quantitative metrics for the study of RBPs, we expect that the research reported here will further encourage rigorous, quantitative hypothesis testing.

Concern #7) Fig2g It would be helpful if the total number of RBPs identified by each method are included in the figure. 1794 proteins identified via the LEAP-RBP method likely include some of the novel RBPs whose RNA binding activity has not been reported. The authors claimed that LEAP-RBP has the unique ability to identify bona fide RBPs. Such novel RBPs would be good candidates for the orthogonal experimental validations.

The reviewer suggests including the “total number of RBPs” identified by each method in **Fig. 2g**. Because the number of GO-annotated RNA-binding proteins identified as UV-enriched is included in the figure, it’s likely that the reviewer is referring to the total number of proteins identified as UV-enriched. However, and as discussed in detail above, we do not consider UV-enrichment to be sufficient criteria for assigning RNA-binding activity. It is important to emphasize that we are not claiming that LEAP-RBP has the unique ability

to identify bona fide RBPs, as suggested by the reviewer. We do, however, demonstrate that the enhanced S/N, %TPs, and yield of the LEAP-RBP method allows more sensitive and accurate validation of RBPs by SRA and immunoblot than other RNA-centric methods (**Fig. 5c, 6b**). As noted in the discussion and previous responses, the principles behind SRA and its ability to provide direct evidence that proteins are UV-crosslinked to RNA substrate has been noted by many experts in the field [5, 9, 24]. However, we suspect widespread utilization of this approach is hampered by low method specificity, biases in signal recovery, and poor yield as evidenced by the provided benchmark comparisons (see XRN1 or pAbPC1, TRAPP or “T”; **Fig. 6a**). Regarding the reviewer’s last point on validation of novel RBPs, we were able to identify and validate three non-canonical RBPs using our approach: LRRC59, TRAP α , and RPN1; and, as noted in the previous response, identify cell type-dependent differences in their RNA-bound abundance (**Fig. 4i**).

Concern #8) Fig3a While the authors present more detailed and clearer explanation for the SILAC based S/N ratio estimation in the supplementary Fig9 and the other parts of the manuscript, Fig3a does not seem to be the accurate representation of the S/N that they determined and thus somewhat misleading. For example, a +CL sample may contain more than 1 part of the non RNA-cross-linked proteins, due to the enrichment of the secondary protein that is cross-linked to the RNA-cross-linked protein for instance. Even if the authors provide evidence that suggests equal partitioning of the noise in general (Fig3g), the possibility still remains. It is also important to note that equal partitioning of the noise may be true for some proteins but not for the others, for the same reason provided above. To my understanding, the figure represents the authors’ logic behind the determination of S/N. Nonetheless, it would be more appropriate to revise the figure or provide clearer explanation on the figure legend.

We appreciate the reviewers concern regarding the accuracy of the graphical representations and can address this by providing additional clarification. The reviewer is concerned that the graphical representation of S/N shown in **Fig. 3a** is inaccurate given that non-specific UV-crosslinking of proteins to RNA-bound proteins (i.e., UV-enrichment of RNA-protein-protein adducts). Indeed, the potential for these non-specific UV-crosslinking events are discussed in the main text and elaborated on in **Supplementary Note 7c**: “RNase-dependent mobility shifts to the molecular weight of the unbound counterpart during SDS-PAGE is only expected for proteins bound to sizable RNA substrates through a single UV-crosslinking event (RNA-protein). Although the probability of multiple UV-crosslinked events is presumably lower than single UV-crosslinking events, rigorous testing of this assumption in vivo is lacking and likely depends on the RBP of interest.”. It’s important to note that the relationship in **Fig.3a** “assumes equal noise partitioning between SILAC channels” (**Source Data Fig. 3a, Methods**). However, UV-crosslinking is expected to decrease the starting amount of noise contributed from UV-crosslinked cells relative to non-crosslinked cells. In support of this, comparing equivalent amounts of total protein isolated from UV-crosslinked and non-crosslinked cells by SDS-PAGE and immunoblot shows a UV-dependent decrease in observed

protein migrating at their expected weight for RBPs with estimated UV-crosslinking efficiencies between 10-20% (e.g., HuR and RPL8; **Extended Data Fig. 1a, b**). Therefore, assuming equal noise partitioning likely overestimates free protein contributions towards observed quantities in the UV-crosslinked SILAC channel, making it a conservative estimate. The figure legend in question states “*For additional information on MS data analysis and graphical representations, see Source Data and Methods.*”. There, all the underlying assumptions are clearly stated and the relationship between +CL/-CL ratios and S/N ratios is mathematically established (**Source Data Fig. 3a**).

Literature Cited:

1. Castello, A., M.W. Hentze, and T. Preiss, Metabolic Enzymes Enjoying New Partnerships as RNA-Binding Proteins. *Trends Endocrinol Metab*, 2015. 26(12): p. 746-757.
2. Garcin, E.D., GAPDH as a model non-canonical AU-rich RNA binding protein. *Semin Cell Dev Biol*, 2019. 86: p. 162-173.
3. Thomas, P.D., et al., PANTHER: a library of protein families and subfamilies indexed by function. *Genome Res*, 2003. 13(9): p. 2129-41.
4. Kroning, N., et al., ATP binding to the KTN/RCK subunit KtrA from the K⁺-uptake system KtrAB of *Vibrio alginolyticus*: its role in the formation of the KtrAB complex and its requirement in vivo. *J Biol Chem*, 2007. 282(19): p. 14018-27.
5. Tawk, C., et al., A systematic analysis of the RNA-targeting potential of secreted bacterial effector proteins. *Sci Rep*, 2017. 7(1): p. 9328.
6. Hannigan, M.M., et al., Quantitative Proteomics Links the LRRC59 Interactome to mRNA Translation on the ER Membrane. *Mol Cell Proteomics*, 2020. 19(11): p. 1826-1849.
7. Jankowsky, E. and M.E. Harris, Specificity and nonspecificity in RNA-protein interactions. *Nat Rev Mol Cell Biol*, 2015. 16(9): p. 533-44.
8. Smith, J.M., J.J. Sandow, and A.I. Webb, The search for RNA-binding proteins: a technical and interdisciplinary challenge. *Biochem Soc Trans*, 2021. 49(1): p. 393-403.
9. Vaishali, et al., Validation and classification of RNA binding proteins identified by mRNA interactome capture. *RNA*, 2021. 27(10): p. 1173-1185.
10. Van Ende, R., S. Balzarini, and K. Geuten, Single and Combined Methods to Specifically or Bulk-Purify RNA-Protein Complexes. *Biomolecules*, 2020. 10(8).
11. Vieira-Vieira, C.H. and M. Selbach, Opportunities and Challenges in Global Quantification of RNA-Protein Interaction via UV Cross-Linking. *Front Mol Biosci*, 2021. 8: p. 669939.
12. Wheeler, E.C., E.L. Van Nostrand, and G.W. Yeo, Advances and challenges in the detection of transcriptome-wide protein-RNA interactions. *Wiley Interdiscip Rev RNA*, 2018. 9(1).
13. Baltz, A.G., et al., The mRNA-bound proteome and its global occupancy profile on protein-coding transcripts. *Mol Cell*, 2012. 46(5): p. 674-90.
14. Bao, X., et al., Capturing the interactome of newly transcribed RNA. *Nat Methods*, 2018. 15(3): p. 213-220.
15. Castello, A., et al., Insights into RNA biology from an atlas of mammalian mRNA-binding proteins. *Cell*, 2012. 149(6): p. 1393-406.
16. Hoefig, K.P., et al., Defining the RBPome of primary T helper cells to elucidate higher-order Roquin-mediated mRNA regulation. *Nat Commun*, 2021. 12(1): p. 5208.
17. Kalesh, K., et al., Transcriptome-Wide Identification of Coding and Noncoding RNA-Binding Proteins Defines the Comprehensive RNA Interactome of *Leishmania mexicana*. *Microbiol Spectr*, 2022. 10(1): p. e0242221.

18. Perez-Perri, J.I., et al., Discovery of RNA-binding proteins and characterization of their dynamic responses by enhanced RNA interactome capture. *Nat Commun*, 2018. 9(1): p. 4408.
19. Queiroz, R.M.L., et al., Comprehensive identification of RNA-protein interactions in any organism using orthogonal organic phase separation (OOPS). *Nat Biotechnol*, 2019. 37(2): p. 169-178.
20. Shchepachev, V., et al., Defining the RNA interactome by total RNA-associated protein purification. *Mol Syst Biol*, 2019. 15(4): p. e8689.
21. Sysoev, V.O., et al., Global changes of the RNA-bound proteome during the maternal-to-zygotic transition in *Drosophila*. *Nat Commun*, 2016. 7: p. 12128.
22. Trendel, J., et al., The Human RNA-Binding Proteome and Its Dynamics during Translational Arrest. *Cell*, 2019. 176(1-2): p. 391-403 e19.
23. Urdaneta, E.C., et al., Purification of cross-linked RNA-protein complexes by phenol-toluol extraction. *Nat Commun*, 2019. 10(1): p. 990.
24. Hentze, M.W., et al., A brave new world of RNA-binding proteins. *Nat Rev Mol Cell Biol*, 2018. 19(5): p. 327-341.
25. Huppertz, I., et al., Riboregulation of Enolase 1 activity controls glycolysis and embryonic stem cell differentiation. *Mol Cell*, 2022. 82(14): p. 2666-2680 e11.
26. Nagaraj, N., et al., Deep proteome and transcriptome mapping of a human cancer cell line. *Mol Syst Biol*, 2011. 7: p. 548.
27. Holdt, L.M., A. Kohlmaier, and D. Teupser, Circular RNAs as Therapeutic Agents and Targets. *Front Physiol*, 2018. 9: p. 1262.
28. Jaros, J.A., et al., Affinity depletion of plasma and serum for mass spectrometry-based proteome analysis. *Methods Mol Biol*, 2013. 1002: p. 1-11.
29. Qin, W., et al., Spatiotemporally-resolved mapping of RNA binding proteins via functional proximity labeling reveals a mitochondrial mRNA anchor promoting stress recovery. *Nat Commun*, 2021. 12(1): p. 4980.

REVIEWER COMMENTS

Reviewer #1 (Remarks to the Author):

I acknowledge the authors for their response and the effort they made in addressing all my concerns, however I am not fully satisfied with the answers. Probably I did not express myself clearly in some of my points, so I would kindly ask them to clarify the following issues.

Comment #1: I thank the authors the modifications, I consider they are useful and help to interpret the data, however my comment was more focused on the proteins selected for the validations performed across the manuscript.

I think is clear how proteins were selected in general, but I still miss the criteria for the selection of proteins displayed in the different panels. In the analysis depicted in Fig.1, proteins lacking prior GO-RBP annotation (i.e., β -tubulin) as well as GO-annotated RBPs containing (e.g., HuR) or lacking (e.g., GRP94) established RNA binding domains were selected, but why within the proteins lacking prior GO-RBP annotation, authors selected β -tubulin, or GRP94 among proteins lacking an established RBD? My point is, are these proteins representative of what we could find within that groups? For me would be helpful to include in this westerns proteins with different features other than just harboring RBDs or being annotated as RBPs. It would be very illustrating that within GO-annotated RBPs containing RBDs author would include proteins with different number of known targets, or proteins for instance that would bind mRNA and other type of RNA, such as rRNA, or tRNA, etc... Same applies to the other 2 groups, what are the differences one might expect from GRP94, RPN1, LRRRC59? I mean, looking at the literature, is there any CLIP data from these proteins?

Just to sum up, I would like to know whether the candidates selected for the different panels truly reflect the different possibilities.

Comment #2: Many thanks for addressing the RNase concern. Regarding the glycolytic enzymes, I just pointed out 2 recent articles about this topic, but as authors may know this is a hot topic which has gained a lot of attention in the last 10 years and there are many more research articles describing additional functions in RNA binding carried out by glycolytic enzymes. For instance, the specific case of GAPDH has been described not only in mammalian cells but also in other organisms, so for me is very intriguing that this novel method cannot identify this protein as an RNA binder. Is clear that this method is very specific and can distinguish real binders but what about those proteins which are on the edge due to low RNA association- should they be considered as non-real binders? I know is hard to set a threshold and I think the authors made a great effort to establish a novel method to select real RBPs however I still think that conclusions cannot be so restrictive and consider that this method can also have some limitations.

Comment #3: I thank the authors again for addressing my comment. I have a question about this though. I understand that the introduction of a quantitative method such as SILAC can distinguish UV-dependent enrichment of RNA-free protein but I am still surprised that RIC recovered the same mRNA-binding proteins as the method presented by the authors. RIC proteomics do not apply SILAC, so as the authors claim this method cannot distinguish UV-dependent enrichment of RNA-free protein, so I would expect to get more than 11% proteins exclusively present in RIC, since one should get more unspecific proteins but looking at the result, my conclusion is that RIC is pretty specific for mRBPs. If anybody wants to analyse only mRBPs I still wonder, why we should go for a protocol which is more complex and long if the result (and I am not talking about the quantification but the identification of proteins) is very similar (technical replicates of RIC would show this 10% difference).

I think this new methodology is great, is clever but I think is very useful when you are comparing conditions or you want to identify the entire RBPome, however, when looking at mRBPs I would go for RIC – quicker and less laborious.

Reviewer #2 (Remarks to the Author):

I appreciate the author's effort to address the potential concerns, that are also similarly shared by the other reviewer, and their responses have allowed to me to better understand their findings. I also find the manuscript, as it is, to be logically sound and can be of value to the future RNA binding protein profiling studies.

Nonetheless, as it was stated in the previous comment, to me, the value and applicability of the method and the criteria for the experimental determination of the RBPs, which have significantly higher S/N ratio, appear to be especially relevant for the OOPS/XRNAX and analogous techniques which, as they stand, apparently include substantially larger amount of noise, RNA-free, proteins in their analyses compared to the other methods. On the other hand, potential gain from the application of the newly defined criteria for the experiment such as RIC seem to be rather limited, despite the authors' claim that exceptional utility of the LEAP-RBP is established through comparative analysis. Related comments are also provided below. It is also important to note that while 'complete' implementation of the method and criteria presented in this manuscript require SILAC-MS analysis approach. SILAC-MS analysis requires relatively high amount of effort/cost and limit the applicability of the method to the cultured cell lines and model organisms.

Given that, based on the authors' response, I now agree that additional experiment/validation, and thus further revision of the manuscript, may not be especially relevant for the scope of this study. Although, I do have some comments regarding their claims which are provided below. Given that, I

do believe that it is up to the editors' decision on the merits of the author's finding to the general scientific community.

Comments

Fig 5. and 6.

In general, it appears that compared to the other approaches, RIC/eRIC enriched the RBPs with relatively high S/N, even compared to the LEAP-RBP, and most of the identified proteins are "GO RBP". Potential gain from the utilization of the newly defined criteria for the method such as RIC thus seem to be rather limited. While, in principle, stricter RBP filtering criteria ought to increase the accuracy of RBP profiling, as it is true for all of the experimental method and analysis, there is risk of losing potentially real interactors. It is also important to note that use of SILAC in LEAP-RBP limit the method's applicability compared to the non-SILAC based RIC/eRIC.

It also seems rather premature to conclude that LEAP-RBP have distinct advantage over the RIC, based on the fact that it identified greater number of annotated 'mRNA binders'. Such known mRNA interacting RBPs may also interact with the ribosomal RNAs in cell. Actual interaction between LEAP-RBP exclusive 'mRNA binders' and mRNA may have been relatively, and thus absent from the RIC defined RBPs. It is thus unclear and rather premature to say that LEAP-RBP can replace, or have distinct advantage over, the RIC methods in the profiling of mRNA interactome.

Also it is not clear to me that how the authors performed similar analysis, defining S/N, on the RIC and eRIC experiments both of which were not done with the SILAC. Authors may want to elaborate more on how they derived such S/N and comment on whether it is appropriate, and how, to apply the same criteria/analysis platform that authors devised to the none SILAC experiments.

Author's Response to Concern #4)

Here authors claim that LEAP-RBP on four different cell lines, 3 of which are human origin and another of rat origin. As the authors noted in other comments, demonstration of the condition dependent RBP profiling may indeed be out of scope for this study and it does not seem to be necessary for the demonstration of the value of this study.

Nonetheless, experiment and data presented in Figure 4 does not seem to constitute a rigorous test for LEAP-RBP's ability to detect biologically relevant change in RNA interactome.

1. Experimental conditions within three different human cell lines are drastically different, in terms of nucleus to cytoplasm ratio etc. It is thus not surprising that three RBP profile determined via LEAP-RBP are distinct from each other. It is therefore unclear to me how one would properly assess the actual significance of the data to reach the conclusion that LEAP-RBP has the ability to accurately determine the biologically relevant changes in RNA-protein interactions.

2. In line with the above comment, I also appreciate that authors performed the additional experiment on rat insulinoma and their claim that increase in the abundance of distinct ER RBPs can

be explained based on the distinct characteristics of the 832/13 cell line. On the other hand, authors compared the human and rat cell lines to reach to that conclusion which does seem to be rather inappropriate. LEAP-RBP abundance may have changed due to the number of other factors apart from the explanation provided by the authors. Strictly speaking, it remains unclear whether the LEAP-RBP detected context dependent change in RNA-protein interactions or simple differences between the human and rat proteins, including but not limited to the RNA binding affinity and protein characteristics.

Author's Response to Concern #6)

In terms of the orthogonal validation approach, I was referring to the RNA centric approaches, that have been repeatedly utilized in the RIC studies. Representative example and related information can be found in Beckmann et al. 2015 Nature communications (Figure 1. E). On the other hand, based on the authors' response I now agree that additional validation may not be necessary and out of scope for this manuscript.

Point-by-point Response:

Reviewer #1:

Summary: I acknowledge the authors for their response and the effort they made in addressing all my concerns, however I am not fully satisfied with the answers. Probably I did not express myself clearly in some of my points, so I would kindly ask them to clarify the following issues.

Response: We regret not being able to adequately address the reviewer's concerns. To rectify this, we completed a substantial editing of the manuscript, with further edits reflecting responses to the reviewer's comments below.

Comment #1: I thank the authors the modifications, I consider they are useful and help to interpret the data, however my comment was more focused on the proteins selected for the validations performed across the manuscript. I think is clear how proteins were selected in general, but I still miss the criteria for the selection of proteins displayed in the different panels. In the analysis depicted in Fig.1, proteins lacking prior GO-RBP annotation (i.e., β -tubulin) as well as GO-annotated RBPs containing (e.g., HuR) or lacking (e.g., GRP94) established RNA binding domains were selected, but why within the proteins lacking prior GO-RBP annotation, authors selected β -tubulin, or GRP94 among proteins lacking an established RBD? My point is, are these proteins representative of what we could find within that groups? For me would be helpful to include in this westerns proteins with different features other that just harboring RBDs or being annotated as RBPs. It would be very illustrating that within GO-annotated RBPs containing RBDs author would include proteins with different number of known targets, or proteins for instance that would bind mRNA and other type of RNA, such as rRNA, or tRNA, etc... Same applies to the other 2 groups, what are the differences one might expect from GRP94, RPN1, LRRRC59? I mean, looking at the literature, is there any CLIP data from these proteins? Just to sum up, I would like to know whether the candidates selected for the different panels truly reflect the different possibilities.

Response: We greatly appreciate that candid dialog of the reviewer and to further address the concerns noted above we made additional modifications to **Fig. 1c**, to show the current consensus RNA target of GO-annotated RBPs, which we draw from reference [1](citations included as the final page of this point-by-point), and in **Fig. 4g** where we introduce an additional ten proteins for validation of newly included data on translation-dependent RBP remodeling. Additionally, we have added a curated list of known RNA-binding domains for the proteins we investigated. Of note, there is evidence that LRRRC59 interacts with ribosomes and translational machinery localized to the ER membrane [2, 3]. However, as the reviewer correctly points out, CLIP data is lacking for LRRRC59, and many other UV-enriched proteins which were selected for validation (e.g., GRP94, β -tubulin). Although access to CLIP datasets for every protein studied is preferable, we didn't consider the lack of such data as a reason to exclude a given UV-enriched protein from being validated in our system. From a glass half full perspective, and given recent easyCLIP data revealing spurious, low-frequency UV-crosslinking events that are commonly scored as a demonstration of RNA binding [4], we considered the absence of CLIP data to be a good reason to directly validate these targets. Because LEAP-RBP recovers close to if not 100% of RNA-bound protein, the inability to detect them in our system is likely indicative of a very low UV-crosslinking efficiency (**Extended Data Fig. 1a**). In sum and including the additional targets reported in this revised manuscript, we have now directly, orthogonally validated RNA-binding activity for twenty-five UV-enriched proteins (i.e., putative RBPs) and include these data in the revised manuscript.

To further address RBP/protein selection rationale, in particular whether the spectrum of RNA interactions was examined, the new proteins we examined for validation of the comparative LEAP-RBP experiments now includes GEMIN5 which binds snRNA, EEF2 which binds tRNA, and FBL which binds snoRNA (see **Fig. 4g**). Based on reported frequency of RBPs which bind different RNA biotypes [1], we now feel that we have provided a fully representative analysis, and demonstrated the utility of LEAP-RBP in each scenario (**Supplementary Table 1**).

Comment #2: Many thanks for addressing the RNase concern. Regarding the glycolytic enzymes, I just pointed out 2 recent articles about this topic, but as authors may know this is a hot topic which has gained a lot of attention in the last 10 years and there are many more research articles describing additional functions in RNA binding carried out by glycolytic enzymes. For instance, the specific case of GAPDH has been described not

only in mammalian cells but also in other organisms, so for me is very intriguing that this novel method cannot identify this protein as an RNA binder. Is clear that this method is very specific and can distinguish real binders but what about those proteins which are on the edge due to low RNA association- should they be considered as non-real binders? I know is hard to set a threshold and I think the authors made a great effort to establish a novel method to select real RBPs however I still think that conclusions cannot be so restrictive and consider that this method can also have some limitations.

Response: We share the reviewer's perspective and scientific curiosity on these points. On the one hand there is a need to be open to the possibility that there could be many cellular proteins of known function that also have moonlighting functions as RBPs, but on the other hand there is also a need for quantitative analyses and common metrics for establishing bona fide vs. biologically irrelevant (random) interactions. The Khavari lab recently proposed a CLIP-based approach to this question in their Nature Communications paper (doi.org/10.1038/s41467-021-21623-4). They present the scientific challenge very clearly: "*Addressing this question for such proteins, and for additional potentially novel RBPs, has been hindered by the lack of a test that quantitates RNA interaction events per protein molecule to provide a global cutoff level of RNA binding to nominate a protein as an RBP.*" As noted in the prior submission, we've revised the manuscript to explicitly state the possibility that proteins with low-frequency UV-crosslinking may fall below the limit of detection in our system. We have also revised the manuscript title and the primary conclusions of the paper to make it clear that we are not suggesting a strict, binary criteria for assigning RNA-binding activity. We anticipate that the experimental systems we report here will catalyze interest in rigorous RBP characterization, and through such approaches the question of the biological significance of very low frequency RNA crosslinking to proteins can be more meaningfully studied.

A scientific point to consider – if the question is can GAPDH be UV crosslinked to RNA, the answer is yes. That said, the accounting matters; if a known RBP such as HuR can be crosslinked with high (e.g. > 30% efficiency), displays high UV-dependent partitioning to the AGPC interphase as validated by SILAC, and has a high S/N in the SRA assay, and a candidate RBP such as GAPDH, displays an RNA crosslinking efficiency of < 0.01%, very low UV-dependent partitioning to the AGPC interphase via SILAC, and a very low S/N by SRA we feel that it is unlikely that the observed RNA crosslinking is biologically relevant. Others in the field will likely disagree, but our method and analytical approach provides a common quantitative framework to debate the interpretation of these data and catalyze further hypothesis testing.

Comment #3: I thank the authors again for addressing my comment. I have a question about this though. I understand that the introduction of a quantitative method such as SILAC can distinguish UV-dependent enrichment of RNA-free protein but I am still surprised that RIC recovered the same mRNA-binding proteins as the method presented by the authors. RIC proteomics do not apply SILAC, so as the authors claim this method cannot distinguish UV-dependent enrichment of RNA-free protein, so I would expect to get more than 11% proteins exclusively present in RIC, since one should get more unspecific proteins but looking at the result, my conclusion is that RIC is pretty specific for mRBPs. If anybody wants to analyze only mRBPs I still wonder, why we should go for a protocol which is more complex and long if the result (and I am not talking about the quantification but the identification of proteins) is very similar (technical replicates of RIC would show this 10% difference). I think this new methodology is great, is clever but I think is very useful when you are comparing conditions or you want to identify the entire RBPome, however, when looking at mRBPs I would go for RIC – quicker and less laborious.

Response: This a very useful, perceptive commentary and we are grateful for this scientific dialog on this point. Regarding the expected percentage of protein exclusively identified by the RIC method, it's not clear whether there is an accurate way to predict what percentage of proteins would be exclusively identified by the RIC method. We are not, though, suggesting these represent false positives simply because they were not identified in LEAP-RBP fractions. Regarding the reviewer preference for the RIC method over the LEAP-RP method, we respect the reviewer's opinion and think either method can provide valuable information when applied appropriately. However, most current GO-annotations for RNA-binding (GO:0003723) or mRNA binding (GO:0003729) for proteins are based on their significant UV-enrichment status alone, and in only a few RIC-like (non-SILAC) experiments (see red circle with blue outline; **Fig. 4f**) [5-7]. While the GO-analyses presented in **Supplementary Figure 15** is typically considered the gold standard when evaluating UV-enrichment

specificity of RNA-centric methods, it's also important to note it will always favor RIC-like methods. As the reviewer correctly notes, it's not necessary to use the LEAP-RBP method when studying well-established, canonical mRBPs. However, based on our analysis, there isn't a clear reason to not use LEAP-RBP in place of RIC. In our experience, the LEAP-RBP method is much more time and resource efficient and considerably more cost-effective: RNP fractions which cost US\$150 to isolate with the RIC method costs less than US\$1 to isolate with the LEAP-RBP method. From a lab resource perspective this alone makes LEAP-RBP more scientifically egalitarian and accessible.

Reviewer #2:

Summary: I appreciate the author's effort to address the potential concerns, that are also similarly shared by the other reviewer, and their responses have allowed to me to better understand their findings. I also find the manuscript, as it is, to be logically sound and can be of value to the future RNA binding protein profiling studies. Nonetheless, as it was stated in the previous comment, to me, the value and applicability of the method and the criteria for the experimental determination of the RBPs, which have significantly higher S/N ratio, appear to be especially relevant for the OOPS/XRNAX and analogous techniques which, as they stand, apparently include substantially larger amount of noise, RNA-free, proteins in their analyses compared to the other methods. On the other hand, potential gain from the application of the newly defined criteria for the experiment such as RIC seem to be rather limited, despite the authors' claim that exceptional utility of the LEAP-RBP is established through comparative analysis. Related comments are also provided below...

Response: We want to again thank the reviewer for highlighting the value and utility of the findings presented in this submission. We believe that the additional comparative LEAP-RBP experiment we include in this revision, where we examine how global alterations in CDS ribosome occupancy alter RBP/mRNA interactions, notably using a non-SILAC approach, further establishes the value and versatility of the method as well as the analytical framework we have developed. Recognizing the essence of the reviewer's concern, we have edited the manuscript extensively, making every effort to limit conclusions to scientifically objective statements that are well supported by experimental data. Responses to individual concerns and comments are provided below.

Concern #1: ... It is also important to note that while 'complete' implementation of the method and criteria presented in this manuscript require SILAC-MS analysis approach. SILAC-MS analysis requires relatively high amount of effort/cost and limit the applicability of the method to the cultured cell lines and model organisms. Given that, based on the authors' response, I now agree that additional experiment/validation, and thus further revision of the manuscript, may not be especially relevant for the scope of this study. Although, I do have some comments regarding their claims which are provided below. Given that, I do believe that it is up to the editors' decision on the merits of the author's finding to the general scientific community.

Response: We appreciate the reviewers concern and addressed this in the revised manuscript by providing the comparative LEAP-RBP experiment (non-SILAC) noted above and demonstrating the ability of our method to identify RBPs displaying condition-dependent differences in RNA-occupancy. Also, we now explicitly state the experimental limitation noted by the reviewer in the **Methods** section of the revised manuscript as shown below. We do note that the Hentze lab has recently established that applications of UV crosslinking to mammalian organs: doi.org/10.1038/s41467-023-37494-w.

Revisions in response to concern (new edits are in italicized blue font)

Methods. Criteria for assignment of RNA-binding activity:

Protein displaying +CL/-CL ratio >0 in LEAP-RBP fractions by SILAC LC-MS/MS were considered high or low confidence RBPs based on their observed enrichment efficiency (S/N) and abundance (%TP). However, only those displaying discernible RNase-sensitivity by SRA and immunoblot were considered bona fide RNA-binding proteins. Proteins which remained RNase-insensitive or undetectable were considered non-RBPs regardless of GO-annotation status (e.g., GRP94, a GO-annotated RBP). However, because the inability to detect a protein by SRA and immunoblot could be due to their low RNA-bound protein abundance, it was not considered confirmation that it lacks RNA-binding activity (Supplementary Methods). *To this point, validation of RNA-binding activity with LEAP-RBP and SRA requires that RNA-protein interactors are susceptible to UV-crosslinking.*

Therefore, current applications are limited to cultured cell lines and model organisms that are capable of being UV-irradiated.

Comment #1: In general, it appears that compared to the other approaches, RIC/eRIC enriched the RBPs with relatively high S/N, even compared to the LEAP-RBP, and most of the identified proteins are “GO RBP”. Potential gain from the utilization of the newly defined criteria for the method such as RIC thus seem to be rather limited

Response: As noted in the responses to reviewer 1, most of the current GO-annotations for RNA-binding (GO:0003723) or mRNA binding (GO:0003729) of proteins are based on their significant UV-enrichment status in a limited number of RIC-like (non-SILAC) experiments [5-7]. Therefore, most proteins identified by the referenced RIC study are now GO-annotated as RNA- and/or mRNA- binders. This highlights the need for caution in the use of GO annotation alone in referencing RNA binding function.

Concern #1: While, in principle, stricter RBP filtering criteria ought to increase the accuracy of RBP profiling, as it is true for all of the experimental method and analysis, there is risk of losing potentially real interactors. It is also important to note that use of SILAC in LEAP-RBP limit the method’s applicability compared to the non-SILAC based RIC/eRIC.

Response: We agree that this is an important concern and have included a non-SILAC comparative LEAP-RBP experiment to address this concern (**Fig. 4a-h**). In support of the reviewers’ comments, utilizing a stringent S/N-based cutoff was found to improve the specificity of our analysis; it was also found to improve the sensitivity and detection of RBPs exhibiting condition-dependent differences in RNA-occupancy by reducing multiple hypotheses testing (discussed in the revised manuscript). Nonetheless, we do want to emphasize how valuable this concern has been in providing the impetus for including the new data in the revised manuscript and for identifying useful computational approaches to presenting the new findings.

Concern #2: It also seems rather premature to conclude that LEAP-RBP have distinct advantage over the RIC, based on the fact that it identified greater number of annotated ‘mRNA binders’. Such known mRNA interacting RBPs may also interact with the ribosomal RNAs in cell. Actual interaction between LEAP-RBP exclusive ‘mRNA binders’ and mRNA may have been relatively, and thus absent from the RIC defined RBPs. It is thus unclear and rather premature to say that LEAP-RBP can replace, or have distinct advantage over, the RIC methods in the profiling of mRNA interactome.

Response: We feel we have been careful to not conclude that the LEAP-RBP method has an advantage over RIC methods for profiling the mRNA interactome, and we agree that RBPs which interact with mRNA may also interact with rRNA as is frequently reported in CLIP-seq studies [8, 9]. To this point, and noted above, we removed the word “exceptional” from the manuscript to make clear that we are not viewing LEAP-RBP as globally superior to other related RNA-centric methods for study of the mRNA interactome. Our comparison showing the LEAP-RBP method identified a greater number of mRNA binders than RIC/eRIC is not, however, inaccurate and does rely on quantitative signal-based metrics which are very useful in drawing robust conclusions on RNA interaction specificity and dynamics.

Concern #3: Also it is not clear to me that how the authors performed similar analysis, defining S/N, on the RIC and eRIC experiments both of which were not done with the SILAC. Authors may want to elaborate more on how they derived such S/N and comment on whether it is appropriate, and how, to apply the same criteria/analysis platform that authors devised to the none SILAC experiments.

Response: We appreciate the reviewers concern though feel that this was addressed in the previously submitted manuscript and supporting material. For our analyses and conclusions to be of utility to the field, we felt the need to extensively document the work and analyses performed. The unavoidable consequence of this is a very lengthy presentation. To this point all **Supplementary Tables** and **Source Data** containing signal-based metrics estimated using available MS datasets generated by non-SILAC LC-MS/MS approaches contain one or both of following statements: “*S/N and %TP_s were considered overestimated for non-SILAC experiments and for experiments where UV-crosslinked and non-crosslinked samples weren't pooled prior to RNA-centric*”

protein enrichment.”; “Non-SILAC comparisons were expected to result in UV-enrichment of free protein as evidenced by an increase in non-specific %TP_(S) contributions from non-RBPs (Methods).”. Accordingly, the MS data analysis section we include details how signal-based metrics were estimated for non-SILAC LC-MS/MS experiments and supporting evidence for the claims above were provided throughout the **Supplementary Note (Supplementary Note 4b,5a,b,6a,7e,8a,b)**. To estimate signal-based metrics based on non-SILAC comparisons, it was necessary to make the same assumption originally made in the referenced studies themselves that protein recovered from independent non-crosslinked samples reflected the amount of free protein recovered from independent UV-crosslinked samples. Because UV- or signal-dependent recovery of noise is observed for all RNA-centric methods, this underestimates the amount of free protein recovered from UV-crosslinked cells and overestimates the percentage of total protein isolated that is RNA-bound. However, and as a major point of our paper, this phenomenon can be easily detected by an increase in non-specific %TP_(S) contributions from non-RBPs (i.e., those lacking GO-annotated RNA-binding activity).

Concern #4: Here authors claim that LEAP-RBP on four different cell lines, 3 of which are human origin and another of rat origin. As the authors noted in other comments, demonstration of the condition dependent RBP profiling may indeed be out of scope for this study and it does not seem to be necessary for the demonstration of the value of this study. Nonetheless, experiment and data presented in Figure 4 does not seem to constitute a rigorous test for LEAP-RBP’s ability to detect biologically relevant change in RNA interactome.

1. Experimental conditions within three different human cell lines are drastically different, in terms of nucleus to cytoplasm ratio etc. It is thus not surprising that three RBP profile determined via LEAP-RBP are distinct from each other. It is therefore unclear to me how one would properly assess the actual significance of the data to reach the conclusion that LEAP-RBP has the ability to accurately determine the biologically relevant changes in RNA-protein interactions.

2. In line with the above comment, I also appreciate that authors performed the additional experiment on rat insulinoma and their claim that increase in the abundance of distinct ER RBPs can be explained based on the distinct characteristics of the 832/13 cell line. On the other hand, authors compared the human and rat cell lines to reach to that conclusion which does seem to be rather inappropriate. LEAP-RBP abundance may have changed due to the number of other factors apart from the explanation provided by the authors. Strictly speaking, it remains unclear whether the LEAP-RBP detected context dependent change in RNA-protein interactions or simple differences between the human and rat proteins, including but not limited to the RNA binding affinity and protein characteristics.

Response: We have taken the reviewer’s concerns quite seriously and feel we have now addressed them with the new experimental and data additions made in the revised manuscript (new **Fig. 4**). We agree with the reviewer’s distinction between condition- and context-dependent differences in RNA-bound proteomes and are grateful for the warranted criticism.

Comment #2: In terms of the orthogonal validation approach, I was referring to the RNA centric approaches, that have been repeatedly utilized in the RIC studies. Representative example and related information can be found in Beckmann et al. 2015 Nature communications (Figure 1. E). On the other hand, based on the authors’ response I now agree that additional validation may not be necessary and out of scope for this manuscript.

Response: We recognize the utility of the orthogonal validation methods the reviewer is referring to and, as they’ve mention, don’t consider it to be necessary and out of the scope of the study.

Literature Cited:

1. Gerstberger, S., M. Hafner, and T. Tuschl, A census of human RNA-binding proteins. *Nat Rev Genet*, 2014. 15(12): p. 829-45.
2. Ichimura, T., et al., Isolation and some properties of a 34-kDa-membrane protein that may be responsible for ribosome binding in rat liver rough microsomes. *FEBS Lett*, 1992. 296(1): p. 7-10.
3. Hannigan, M.M., et al., Quantitative Proteomics Links the LRRRC59 Interactome to mRNA Translation on the ER Membrane. *Mol Cell Proteomics*, 2020. 19(11): p. 1826-1849.

4. Porter, D.F., et al., easyCLIP analysis of RNA-protein interactions incorporating absolute quantification. *Nat Commun*, 2021. 12(1): p. 1569.
5. Baltz, A.G., et al., The mRNA-bound proteome and its global occupancy profile on protein-coding transcripts. *Mol Cell*, 2012. 46(5): p. 674-90.
6. Castello, A., et al., Insights into RNA biology from an atlas of mammalian mRNA-binding proteins. *Cell*, 2012. 149(6): p. 1393-406.
7. Castello, A., et al., Comprehensive Identification of RNA-Binding Domains in Human Cells. *Mol Cell*, 2016. 63(4): p. 696-710.
8. Zinnall, U., et al., HDLBP binds ER-targeted mRNAs by multivalent interactions to promote protein synthesis of transmembrane and secreted proteins. *Nat Commun*, 2022. 13(1): p. 2727.
9. Baquero-Perez, B., et al., The Tudor SND1 protein is an m(6)A RNA reader essential for replication of Kaposi's sarcoma-associated herpesvirus. *Elife*, 2019. 8.

REVIEWERS' COMMENTS

Reviewer #1 (Remarks to the Author):

I would like to thank the authors for their effort addressing all my comments. The extensive edition and the addition of novel data is much appreciated and somehow partially address my concerns. As I stated in my previous comments, I still think the technique is valuable for RBP profiling studies and is smartly designed. As the other reviewer indicated, seems to be very relevant for OOPS/XRNAX and analogous techniques, although I agree again with reviewer #2 regarding the comparison with RIC technology.

Comment #1:

I appreciate the addition of novel proteins for validation. It is indeed essential to ensure that the developed method does not exhibit bias to proteins that bind different types of RNA, and this has been nicely demonstrated selecting proteins that bind tRNAs, mRNAs, snRNAs and snoRNAs for further validation. However, my question has not been fully addressed. Of course, I am not asking for CLIP of every single candidate, but I insist that it is important to demarcate the possible limitations of the technique. It would be very helpful to know if proteins with a few targets, or proteins whose targets are not highly expressed would be captured by the method, and as it stands right now, I still do not have it clear.

Comment #2:

I thank the authors for their effort addressing the clear 'need for quantitative analyses and common metrics for establishing bona fide vs. biologically irrelevant interactions'. I think that is clearly established throughout the manuscript. Nonetheless, from my perspective, an RBP may be binding just a few targets and be biologically meaningful. What the authors state about GAPDH is opposed to what has been described about this protein since 1977 – I think they should have a look to this review: Garcin E. (2019), *Seminars in Cell & Developmental Biology*. GAPDH as a model non-canonical AU-rich RNA binding protein. 86:162-173. ISSN 1084-9521. <https://doi.org/10.1016/j.semcd.2018.03.013>. As I mentioned there are dozens of articles describing specific targets of this protein and the biological relevance, so this cannot be just omitted, or state that 'due to the low crosslinking efficiency it is unlikely that the observed RNA crosslinking is biologically relevant'. This is just a simple example, but as I mentioned in my previous comments, this hot topic is currently being studied by several groups worldwide and the lack of these type of RBPs using LEAP-RBP cannot be simply justified by the possible irrelevance of these interactions...

Reviewer #2 (Remarks to the Author):

While I already found the manuscript to be in relatively good shape previously, I am grateful for the authors' response and the efforts to address some of my additional comments and the concerns. Newly updated experimental data, presented in Figure 4, provide the manuscript with additional layer of biological findings and broadened the potential applicability of the method. I would just like to clarify the authors' misunderstanding regarding concern #1 that I provided previously. Otherwise, I found the study to be of value to the scientific community and support its publication.

Concern #1 and the author's response:

Regarding the concern #1, "method to the cultured cell lines and model organisms.", I was in fact only referring to the fact that SILAC is largely inapplicable to human and thus "model organisms". In line with that comment authors performed additional experiment based on non-SILAC approach which I appreciated.

I also agree with the point/additional point added by authors to the manuscript regarding the potential limitation of the LEAP-RBP due to the use of UV crosslinking.

In that regard, it also is not clear to me why would the LEAP-RBP be only applicable to the UV cross-linked samples. Authors may want to elaborate more on the fact that same principle/method may as well be utilized with the other protein-RNA crosslinking methods such as PAR and chemical crosslinking methods.

Reviewer #1:

Summary: I would like to thank the authors for their effort addressing all my comments. The extensive edition and the addition of novel data is much appreciated and somehow partially address my concerns. As I stated in my previous comments, I still think the technique is valuable for RBP profiling studies and is smartly designed. As the other reviewer indicated, seems to be very relevant for OOPS/XRNAX and analogous techniques, although I agree again with reviewer #2 regarding the comparison with RIC technology.

We greatly appreciate the reviewers' kind words and scientific criticism. Indeed, we believe it's important to discuss all limitations inherent to RNA-centric methods, LEAP-RBP or others, during their application for studying proteins that bind to only a few targets and/or whose targets are not highly expressed. To this point, we have made the following revisions to the final paragraph of the discussion section in the revised manuscript (edits are in italicized blue font).

Revisions in response to concern (edits are in blue font)

The results presented herein suggest that the number of RNA-binding proteins currently thought to comprise the RNA-interactome (~4925 human RBPs) [1] and/or those with GO RBP-annotations (~1693) is an overestimation. This perspective is consistent with findings in a recent CLIP-based RNA-protein crosslinking frequency analysis, where a crosslink frequency threshold that distinguishes bona fide and low significance protein-RNA interactions was reported [2]. Importantly, LEAP-RBP combined with quantitative proteomic and SRA analysis provides direct experimental evidence of RNA-binding and orthogonal validation of RBP activity [3-5]. The principle behind SRA has been previously reported and performed in combination with silica based RBP purification methods [6, 7]. Yet, biases in RBP-RNA adduct recovery or low sensitivity ($[S]/\mu\text{g RNA}$) and/or low S/N can confound detection of many bona fide RBPs by SRA analysis alone (e.g., pAbPC1 and XRN1) (T; Fig. 9b). To that end, high %TPs and efficient, unbiased recovery of RNA-bound protein is critical to the accurate identification of RNA interactomes and their state change dynamics, and is a goal largely met by LEAP-RBP (Supplementary Note 3). *The inability to validate metabolic enzymes such as GAPDH which are frequently identified as candidate RBPs by other RBP profiling such as RIC and eRIC may indicate a limitation of our methodology for validation of low-frequency RNA interactors (Supplementary Note 7d). In these scenarios, validation of their RNA-binding function in situ using complementary and/or orthogonal methods may be preferable.* The high specificity and selectivity of the LEAP-RBP method for RNA-bound protein allows efficient capture of broad-spectrum RNA-interactors from biological samples. Potential applications beyond those demonstrated here including PAR and chemical crosslinking approaches are reasonable to consider using the provided strategies (Supplementary Fig. 9-42, Supplementary Note 1-8, Methods).

Comment #1: I appreciate the addition of novel proteins for validation. It is indeed essential to ensure that the developed method does not exhibit bias to proteins that bind different types of RNA, and this has been nicely demonstrated selecting proteins that bind tRNAs, mRNAs, snRNAs and snoRNAs for further validation. However, my question has not been fully addressed. Of course, I am not asking for CLIP of every single candidate, but I insist that it is important to demarcate the possible limitations of the technique. It would be very helpful to know if proteins with a few targets, or proteins whose targets are not highly expressed would be captured by the method, and as it stands right now, I still do not have it clear.

We appreciate the reviewers' points and as mentioned in our previous response have made additional efforts to highlight possible limitations regarding the validation of low-frequency interactors in our system. Based on the data we have, it's likely that low-frequency interactors are effectively captured by LEAP-RBP, but given their low RNA-bound abundance, may fall below the limit of detection despite the high enrichment efficiency of our method for RNA-protein adducts.

Comment #2: I thank the authors for their effort addressing the clear 'need for quantitative analyses and common metrics for establishing bona fide vs. biologically irrelevant interactions'. I think that is clearly established throughout the manuscript. Nonetheless, from my perspective, an RBP may be binding just a few targets and be biologically meaningful. What the authors state about GAPDH is opposed to what has been described about this protein since 1977 – I think they should have a look to this review: Garcin E. (2019), Seminars in Cell & Developmental Biology. GAPDH as a model non-canonical AU-rich RNA binding protein. 86:162-173. ISSN 1084-9521. <https://doi.org/10.1016/j.semcdb.2018.03.013>. As I mentioned there are dozens of articles describing specific targets of this protein and the biological relevance, so this cannot be just omitted, or state that 'due to the low crosslinking efficiency it is unlikely that the observed RNA crosslinking is biologically relevant'. This is just a simple example, but as I mentioned in my previous comments, this hot topic is currently being studied by several groups worldwide and the lack of these type of RBPs using LEAP-RBP cannot be simply justified by the possible irrelevance of these interactions...

We appreciate the reviewer's point and note that we have indeed considered experimental evidence, including citation of the review by Garcin (2019)(doi.org/10.1016/j.semcdb.2018.03.013) supporting the role of GAPDH as an RNA-binding protein. As noted in our previous responses, we support the possibility that GAPDH and other metabolic enzymes such as ENO1 and PKM may regulate only a few RNA targets. However, we do note that Garcin states "*GAPDH is a multi-functional enzyme that binds to a wide array of mRNAs in cells and in vitro, in addition to its involvement in glycolysis.*" Given that GAPDH is one of the most abundant proteins in the cell yet remains undetectable in RNP fractions isolated by LEAP-RBP and other RNA-centric methods, we feel that it is reasonable to offer our data-driven interpretation - that our inability to identify it as a candidate RBP is likely due to low UV-crosslinking efficiency. Critically, GAPDH, along with other dehydrogenases, display UV-irradiation dependent recovery in the interphase of an AGPC extract, but our work reveals that this is not due to significant crosslinking to RNA. Yet, these are negative data and despite the likelihood of non-specific UV-crosslinking events between RNA and GAPDH or other abundant cellular protein and the growing evidence that many of these proteins show low RNA-binding preferences during CLIP-seq studies [2, 8], we agree that it is not yet possible to claim that such interaction are physiologically irrelevant. Our hope is that our work will catalyze further rigorous investigation into glycolytic enzyme function in RNA biology.

Reviewer #2:

Summary: While I already found the manuscript to be in relatively good shape previously, I am grateful for the authors' response and the efforts to address some of my additional comments and the concerns. Newly updated experimental data, presented in Figure 4, provide the manuscript with additional layer of biological findings, and broadened the potential applicability of the method. I would just like to clarify the authors' misunderstanding regarding concern #1 that I provided previously. Otherwise, I found the study to be of value to the scientific community and support its publication.

We greatly appreciate the reviewer's for supporting publication of our study and are pleased that the additional data addressed remaining concerns regarding the applicability of our method.

Concern #1 and the author's response: Regarding the concern #1, "method to the cultured cell lines and model organisms.", I was in fact only referring to the fact that SILAC is largely inapplicable to human and thus "model organisms". In line with that comment authors performed additional experiment based on non-SILAC approach which I appreciated.

I also agree with the point/additional point added by authors to the manuscript regarding the potential limitation of the LEAP-RBP due to the use of UV crosslinking.

In that regard, it also is not clear to me why would the LEAP-RBP be only applicable to the UV cross-linked samples. Authors may want to elaborate more on the fact that same principle/method may as well be utilized with the other protein-RNA crosslinking methods such as PAR and chemical crosslinking methods.

We appreciate the reviewer's clarification on these points and agree that the same methodology may be used with other protein-RNA crosslinking approaches. This point has been included in the last paragraph of the discussion in the revised Manuscript.

Revisions in response to concern (edits are in italicized blue font)

The results presented herein suggest that the number of RNA-binding proteins currently thought to comprise the RNA-interactome (~4925 human RBPs) [1] and/or those with GO RBP-annotations (~1693) is an overestimation. This perspective is consistent with findings in a recent CLIP-based RNA-protein crosslinking frequency analysis, where a crosslink frequency threshold that distinguishes bona fide and low significance protein-RNA interactions was reported [2]. Importantly, LEAP-RBP combined with quantitative proteomic and SRA analysis provides direct experimental evidence of RNA-binding and orthogonal validation of RBP activity [3-5]. The principle behind SRA has been previously reported and performed in combination with silica based RBP purification methods [6, 7]. Yet, biases in RBP-RNA adduct recovery or low sensitivity ($|S|/\mu\text{g RNA}$) and/or low S/N can confound detection of many bona fide RBPs by SRA analysis alone (e.g., pAbPC1 and XRN1) (T; Fig. 9b). To that end, high %TPs and efficient, unbiased recovery of RNA-bound protein is critical to the accurate identification of RNA interactomes and their state change dynamics, and is a goal largely met by LEAP-RBP (Supplementary Note 3). The inability to validate metabolic enzymes such as GAPDH which are frequently identified as candidate RBPs by other RBP profiling such as RIC and eRIC may indicate a limitation of our methodology for validation of low-frequency RNA interactors (Supplementary Note 7d). In these scenarios, validation of their RNA-binding function in situ using complementary and/or orthogonal methods may be preferable. The high specificity and selectivity of the LEAP-RBP method for RNA-bound protein allows efficient capture of broad-spectrum RNA-interactors from biological samples. *Potential applications beyond those demonstrated here including PAR and chemical crosslinking approaches are reasonable to consider using the provided strategies* (Supplementary Fig. 9-42, Supplementary Note 1-8, Methods).

Literature Cited:

1. Qin, W., et al., Spatiotemporally-resolved mapping of RNA binding proteins via functional proximity labeling reveals a mitochondrial mRNA anchor promoting stress recovery. *Nat Commun*, 2021. 12(1): p. 4980.
2. Porter, D.F., et al., easyCLIP analysis of RNA-protein interactions incorporating absolute quantification. *Nat Commun*, 2021. 12(1): p. 1569.
3. Hentze, M.W., et al., A brave new world of RNA-binding proteins. *Nat Rev Mol Cell Biol*, 2018. 19(5): p. 327-341.
4. Tawk, C., et al., A systematic analysis of the RNA-targeting potential of secreted bacterial effector proteins. *Sci Rep*, 2017. 7(1): p. 9328.
5. Vaishali, et al., Validation and classification of RNA binding proteins identified by mRNA interactome capture. *RNA*, 2021. 27(10): p. 1173-1185.
6. Asencio, C., A. Chatterjee, and M.W. Hentze, Silica-based solid-phase extraction of cross-linked nucleic acid-bound proteins. *Life Sci Alliance*, 2018. 1(3): p. e201800088.
7. Urdaneta, E.C., et al., Purification of cross-linked RNA-protein complexes by phenol-toluol extraction. *Nat Commun*, 2019. 10(1): p. 990.
8. Ray, D., et al., RNA-binding proteins that lack canonical RNA-binding domains are rarely sequence-specific. *Sci Rep*, 2023. 13(1): p. 5238.